# OmniEarth-Bench: Probing Cognitive Abilities of MLLMs for Earth's Multi-sphere Observation Data

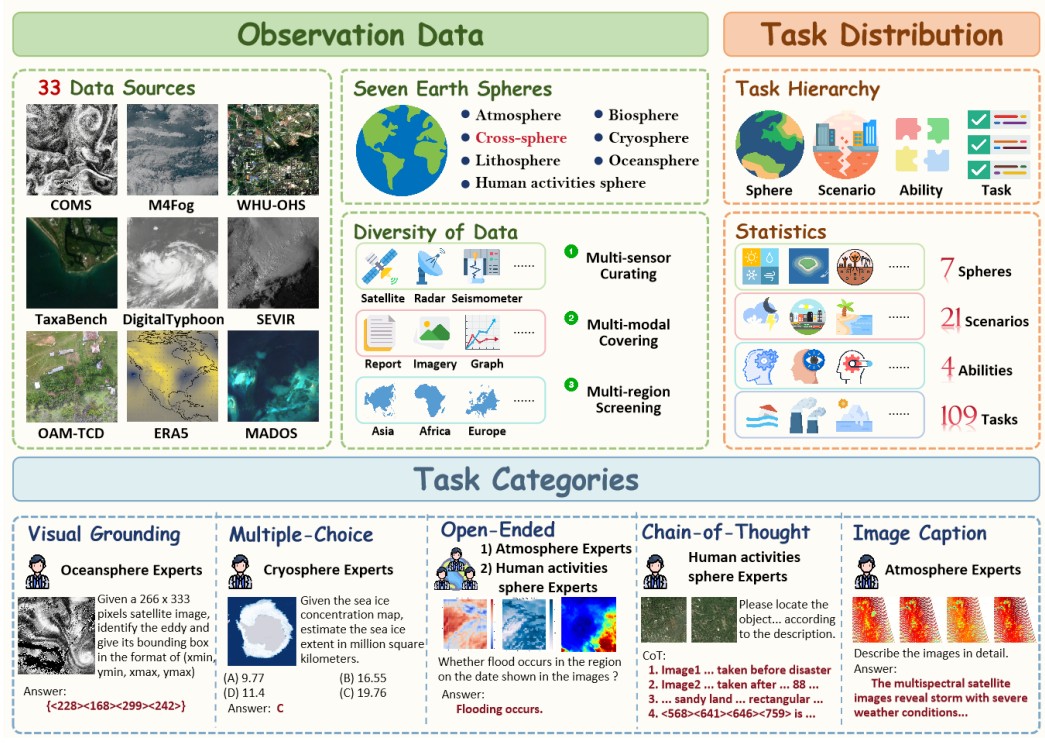

Figure 1: **Overview of OmniEarth-Bench.** Our benchmark spans six Earth science spheres and cross-sphere, encompassing 109 expert-curated tasks derived from 33 data sources. This involved the efforts of 20 experts and 45 crowd-sourcing annotators contributing to the annotations.

## ABSTRACT

Existing benchmarks for multimodal learning in Earth science offer limited, siloed coverage of Earths spheres and their cross-sphere interactions, typically restricting evaluation to the human-activity sphere or atmosphere and to at most 16 tasks. Holistically evaluating MLLMs on observational data across all Earth spheres faces three limitations: *multi-source heterogeneous data*, *unlocking scientific formulation*, and *cross-sphere reasoning*. Therefore, we introduce **OmniEarth-Bench**, the first multimodal benchmark that systematically spans all six spheres: atmosphere, lithosphere, oceansphere, cryosphere, biosphere, and human-activity sphere, and cross-sphere. Built with a scalable, modular pipeline that ingests 33 native Earth-observation sources and expert-in-the-loop curation, OmniEarth-Bench produces 29,855 standardized, expert-curated annotations. All annotations are organized into a four-level hierarchy (Sphere, Scenario, Ability, Task), encompassing 109 expert-curated evaluation tasks. Experiments on 9 state-of-the-art MLLMs reveal that even the most advanced models struggle with our benchmarks, where none of them reach 35% accuracy, revealing systematic gaps in Earth-system cognitive ability. The dataset and evaluation code were released at OmniEarth-Bench.

# 1 INTRODUCTION

Earth scientists study critical environmental and societal problems by modeling Earth's inter-connected subsystems (Bergen et al., 2019), such as the atmosphere, lithosphere, hydrosphere, cryosphere, biosphere, and human-activity sphere (Super, 2023), which together determine plane-tary behavior and risks. Cross-sphere couplings underpin many high-impact discoveries and appli-cations: for example, accurate flood prediction depends on atmospheric precipitation, biospheric soil moisture, and lithospheric runoff (Victor et al., 2024). Earth Observational (EO) data from satel-lites and in-situ networks have expanded dramatically, creating an opportunity to use data-driven methods to reveal process couplings and improve monitoring, forecasting, and decision support in domains such as disaster response, ecosystem management, and climate science (Reichstein et al., 2019; Wang et al., 2023; Ma, 2023; Zhi et al., 2024).

Recently, multimodal large language models (MLLMs) and their benchmarks have advanced core capabilities, including visual perception, long-context modeling, and chain-of-thought (CoT) rea-soning (Liu et al., 2024; Tong et al., 2024; Fu et al., 2023; Li et al., 2023; Saikh et al., 2022; Zhang et al., 2024d; Jiang et al., 2025). In remote sensing, a growing array of multimodal benchmarks has been introduced to tackle large-scale (Wang et al., 2025; Luo et al., 2025), multispectral (Zhang et al., 2024c; Soni et al., 2025), and other applications (Zhou et al., 2025; Li et al., 2025). Atmo-spheric science is now following suit, deploying multimodal models for severe weather events (Ma et al., 2024), meteorological heatmaps (Chen et al., 2024), and climate event forecasting (Li et al., 2024c). Given that the Earth system comprises six major spheres and their intricate couplings, we face an overarching challenge: **how can we holistically evaluate MLLMs cognitive ability under a unified benchmark for observation data across all spheres?** We identify three key attributes of Earth science that give rise to this challenge:

(1) **Multi-source heterogeneous data:** EO data are multi-source and heterogeneous (e.g., multi-spectral satellite imagery, seismic signals, weather reanalysis, microwave sea-ice concentration), demanding careful spatiotemporal co-registration, quality control, and variable/units harmonization across sensors and scales to yield credible labels and supervision for cross-sphere tasks. *Observation Data* in Fig 1 show that heterogeneous EO data cover six earth spheres and cross-sphere.

(2) **Unlocking scientific formulation:** Across Earth science, many tasks demand fine-grained scientific reasoning-for example, multifaceted diagnosis of the El NiñoSouthern Oscillation (ENSO) (Ham et al., 2019) and carbon-flux estimation (Fortier et al., 2024) that quantifies exchange rates between the biosphere and the atmosphere. Creating authoritative evaluation dimensions re-quires domain experts to define scientifically meaningful, sphere-specific tasks and to assess each dimensions suitability as a target for MLLM evaluation. *Task Distribution* and *Task Categories* in Fig. 1 show that each sphere requires coordinated, discipline-specific experts to unlock scientific formulation.

(3) **Cross-sphere reasoning:** Many Earth processes are intrinsically cross-sphere (e.g., precipitationsoil-runoff, oceanatmosphere exchange), models and evaluations must capture interactions rather than iso-lated patterns. This requires domain experts to for-malize the interaction pathway and identify a sci-entific definition set across spheres and to translate these agreements into evaluation criteria that faith-fully reflect inter-sphere dynamics. A cross-sphere example in the *Task Categories* of Fig. 1 shows that cross-sphere tasks require coordinated expertise from experts across different spheres.

In this paper, we introduce **OmniEarth-Bench**, a systematic benchmark to evaluate the cognitive abil-ity of MLLMs across all six Earth science spheres and their couplings. To ensure scientific validity, we engaged 2-5 domain experts per sphere to de-fine evaluation dimensions and select or curate rel-evant observational datasets (e.g., MODIS and other

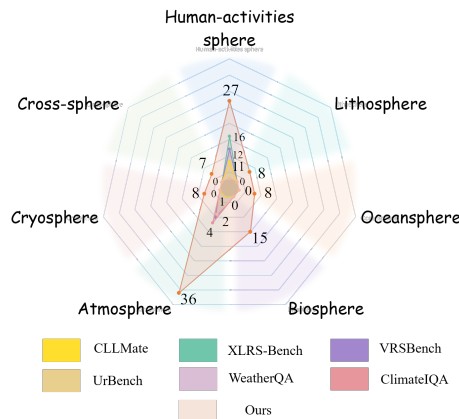

Figure 2: **Comparison of Different Bench-marks.** Our OmniEarth-Bench shows the broadest coverage, with dedicated cross-sphere dimensions.

Table 1: **Comparison between existing vision-language benchmarks and our benchmark.** ✗ represents semi-automated, *i.e.*, machine generation followed by human verification.

| Dataset | Spheres | Cross-Sphere | Observation Data | Data Source Volume | Task | | |
|---|---|---|---|---|---|---|---|
| | | | | | Volume | Dimensions Volume | Expert Annotation |
| ScienceQA (Saikh et al., 2022) | ✗ | ✗ | ✗ | - | 21,000 | 127 | ✗ |
| Seed-Bench (Li et al., 2023) | ✗ | ✗ | ✗ | - | 19,242 | 12 | ✗ |
| MME (Fu et al., 2023) | ✗ | ✗ | ✗ | - | 2,374 | 14 | ✓ |
| MMBench (Liu et al., 2024) | ✗ | ✗ | ✗ | - | 3,217 | 20 | ✓ |
| MME-Realworld (Zhang et al., 2024d) | ✗ | ✗ | ✗ | - | 29,429 | 43 | ✓ |
| ZeroBench (Roberts et al., 2025) | ✗ | ✗ | ✗ | - | 100 | 0 | ✗ |
| MME-CoT (Jiang et al., 2025) | ✗ | ✗ | ✗ | - | 1,130 | 0 | ✗ |
| VRSBench (Li et al., 2024d) | human-activity sphere | ✗ | ✓ | 2 | 175,703 | 12 | ✗ |
| XLRS-Bench (Wang et al., 2025) | Human-activity sphere | ✗ | ✓ | 6 | 45,008 | 16 | ✓ |
| RSIEval (Hu et al., 2023) | Human-activity sphere | ✗ | ✓ | 1 | 933 | 1 | ✗ |
| UrBench (Zhou et al., 2025) | Human-activity sphere | ✗ | ✓ | 6 | 11,600 | 11 | ✗ |
| WeatherQA (Ma et al., 2024) | Atmosphere | ✗ | ✓ | 1 | 8,000 | 2 | ✗ |
| ClimateIQA (Li et al., 2024c) | Atmosphere | ✗ | ✓ | 2 | 254,040 | 4 | ✗ |
| CLLMate (Li et al., 2024c) | Atmosphere | ✗ | ✓ | 2 | 7,747 | 1 | ✗ |
| OmniEarth-Bench | 6 Spheres | ✓ | ✓ | 33 | 29,855 | 109 | ✓ |

satellite/in-situ sources). The annotation team (20 experts plus 45 crowd annotators) produced and quality-checked 29,855 expert-curated annotations organized into a four-level hierarchy (Sphere, Scenario, Ability, Task). The final benchmark contains 109 L-4 Tasks across seven thematic spheres (the six spheres plus an explicit cross-sphere). As shown in Fig. 2 and summarized in Tab. 1, OmniEarth-Bench substantially expands both the breadth and scientific coherence of EO evaluation compared with prior work. Our evaluations across nine state-of-the-art MLLMs reveal large and systematic failure modes (none exceed 35% overall accuracy), demonstrating the pressing need for Earth-system modeling and specialized reasoning mechanisms.

The key contributions are:

- **A unified Earth-observation processing pipeline.** We build a scalable, modular pipeline that ingests 33 native Earth-science data sources and, via expert-in-the-loop curation, produces 29,855 standardized, expert-curated annotations by 20 domain experts and 45 annotators to constitute the OmniEarth-Bench evaluation set.

- **A four-level, sphere-complete evaluation framework.** We provide the first benchmark that systematically covers all six Earth spheres and explicit cross-sphere scenarios, organized into a four-level hierarchy with 109 L-4 sub-dimensions to measure breadth and real-world relevance beyond prior, siloed EO benchmarks.

- **Comprehensive evaluations.** Comprehensive evaluations on nine state-of-the-art MLLMs reveal that even the most advanced models struggle with our benchmarks. Especially, in some cross-sphere tasks, the performance of leading models like GPT-4o drops to 0.0%.

## 2 RELATED WORK

**Earth Multimodal Benchmark.** Recent advancements in large multimodal models (MLLMs) have accelerated progress in Earth sciences (Kuckreja et al., 2024; Muhtar et al., 2024), leading to the development of several evaluation benchmarks (Li et al., 2024d; Wang et al., 2025; Ma et al., 2024; Chen et al., 2024). Current benchmarks primarily target the human-activity sphere and atmosphere. In the human-activity sphere, remote sensing-based benchmarks include RSIEval (Hu et al., 2023), VRSBench (Li et al., 2024d), XLRS-Bench (Wang et al., 2025), and so on. Atmospheric benchmarks include WeatherQA (Ma et al., 2024), ClimateIQA (Chen et al., 2024) and CLLMate (Li et al., 2024c). **However, these benchmarks exhibit notable limitations:** 1) They typically address isolated spheres, neglecting cross-sphere interactions essential to real-world Earth science challenges. 2) They offer limited evaluation dimensions, for example, atmospheric benchmarks assessing fewer than four question types.

**General Multimodal Benchmark.** Large-scale vision-language models (VLMs) have shown great promise in multimodal tasks such as scene understanding and visual sentiment analysis, prompting the development of diverse benchmarks to quantitatively assess their capabilities. However, earlier benchmarks mostly targeted specific domains with limited evaluation tasks (*e.g.*, visual grounding (Sun et al., 2022; Zhan et al., 2023) or visual question answering (VQA) (Hudson & Manning,

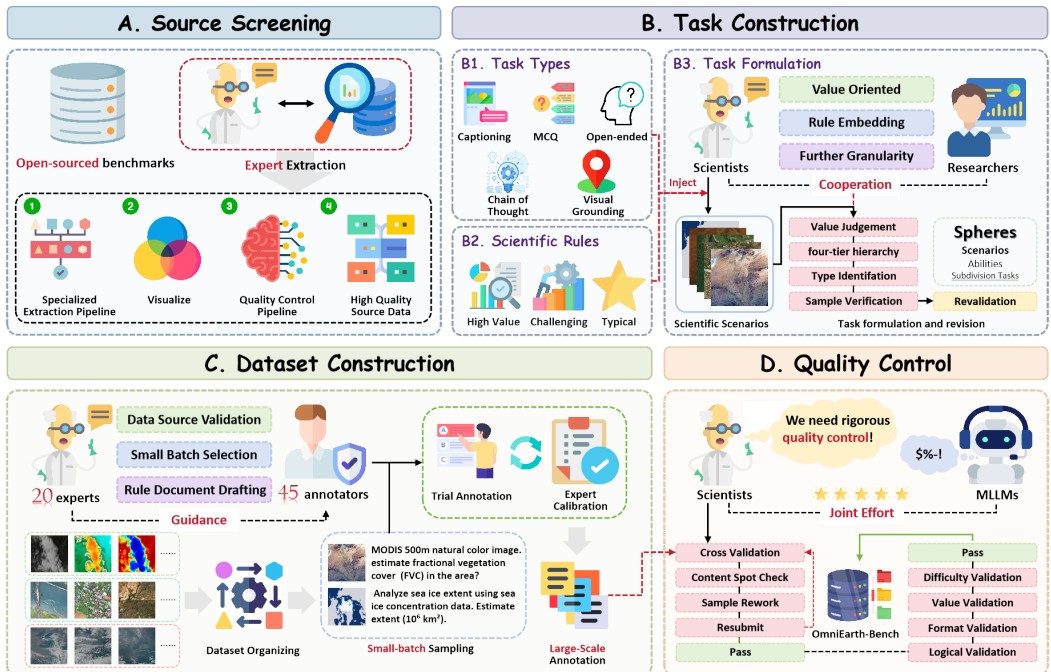

Figure 3: **Pipeline of OmniEarth-Bench.** Our pipeline comprises 4 stages: Source Screening, Task Construction, Dataset Construction, and Quality Control, all led by experts. The first two stages are exclusively conducted by experts, while crowd-sourcing annotators assist in the latter two stages.

2019; Marino et al., 2019; Goyal et al., 2017; Gurari et al., 2018; Singh et al., 2019)). Recent efforts aim for more comprehensive assessments: MME (Fu et al., 2023), MMBench (Liu et al., 2024), Seed-Bench (Li et al., 2023), MMT-Bench (Ying et al., 2024), MME-Realworld (Zhang et al., 2024d), HLE (Phan et al., 2025) and MMMU-Pro (Yue et al., 2024). Multimodal chain-of-thought (CoT) benchmarks were also developed, such as MME-CoT (Jiang et al., 2025) and ZeroBench (Roberts et al., 2025). Despite these advancements, **two critical limitations remain:** 1) Earth sciences have been largely neglected, with only SuperGPQA featuring a minimal number (only 100 annotations) of geophysics-related textual questions. 2) Existing benchmarks overlook the importance of observational data, a distinctive strength of Earth sciences (e.g., climate data grids, seismic signals).

## 3 OMNIEARTH-BENCH

### 3.1 PIPELINE OF BENCHMARK

**Source Screening.** Our Benchmark comprises not only publicly available open-source datasets but also a significant portion of data manually extracted by experts from satellite imagery and raw observational sources. For example, Vegetation Monitoring uses satellite imagery from MODIS and expert-curated data from the Global Land Surface Satellite (GLASS), including Leaf Area Index, Fractional Vegetation Cover, and Peak Vegetation Coverage Area. Moreover, for the Eddy data in oceansphere, the chlorophyll (CHL) data used in this study were obtained by applying the OCI empirical algorithm to Level-2 data acquired by the Geostationary Ocean Color Imager I (GOCI) aboard the Oceanography and Meteorology Satellite (COMS). After careful selection and integration, we compiled a dataset originating from 33 distinct data sources spanning all Earth spheres. It is crucial to clarify that *33 data sources* refers to the raw data origins, not 33 different input modalities (e.g., individual spectral bands or 1D signals). To ensure fair evaluation, domain experts processed all data into MLLM-compatible formats, applying specific strategies tailored to the data characteristics of each sphere. This process prioritized the native data structure; for instance, multi-spectral data was converted into multiple single-channel grayscale images to avoid a dependency on potentially misleading RGB visualizations. Tab.2 is a summary of the data sources used for each Earth sphere, with detailed data organization and construction procedures presented in the Appendix A.3.

Table 2: **Data source of different spheres**, including open-source datasets, satellite websites, and other observation data sources. We only exhibit the L1 and L2 dimensions. Processing pipeline for each data source is shown in the Appendix A.3.

| L1 dimensons | L2 deminesons | Data Source | Annotations Volume |
|---|---|---|---|
| Cross-sphere | Global Flood Forecasting | GFF (Victor et al., 2024) | 873 |
| | Bird Species Prediction | SatBird (Teng et al., 2023) | 2,253 |
| | Carbon Flux Monitoring | CarbonSense (Fortier et al., 2024) | 330 |
| Human-activity sphere | Urban Construction | UBCv1 (Huang et al., 2023) | 3,161 |
| | Land Use | WHU-OHS (Li et al., 2022) | 2,990 |
| | Surface Disaster Assessment | XView (Gupta et al., 2019) | 3,851 |
| Biosphere | Species Distribution Prediction | TreeSatAI (Astruc et al., 2024) OAM-TCD (Veitch-Michaelis et al., 2024) | 2,819 |
| | Vegetation Monitoring | GLASS (Liang et al., 2021) | 900 |
| | Environmental Pollution Monitoring | ROSID (Nurseitov et al., 2024a) | 246 |
| | Human Footprint Assessment | HFP (Sanderson et al., 2002) | 600 |
| | Crop Growth Monitoring | MOPAD (Zheng et al., 2021a) | 1,656 |
| Atmosphere | SEVIR Weather | SEVIR (Veillette et al., 2020) | 893 |
| | Typhoon Events | DigitalTyphoon (Kitamoto et al., 2023) | 5,082 |
| | Short-term meteorological events | ERA5 (Hersbach et al., 2020) | 140 |
| | Medium-term meteorological events | ERA5 (Hersbach et al., 2020) | 160 |
| | Seasonal meteorological events | ERA5 (Hersbach et al., 2020) | 60 |
| | Interannual climate change | ERA5 (Hersbach et al., 2020) | 60 |
| Lithosphere | Earthquake monitoring and prediction | STRAD (Mousavi et al., 2019) | 1,500 |
| | Geological exploration imaging | TGS-Salt (Kainkaryam et al., 2019) | 631 |
| Oceansphere | Marine Debris and Oil Pollution | MADOS (Kikaki et al., 2024) | 221 |
| | Marine Extreme Events | ERASSTv5 (Huang et al., 2017) | 583 |
| | Marine Phenomenon Detection | COMS (Wang et al., 2024b), | 570 |
| Cryosphere | Sea ice forecast | G02202 (SIC) (Meier et al., 2021) PIOMAS (Schweiger et al., 2011) | 200 |
| | Glacier analysis | CryoSat-2 (Helm et al., 2014) IceBridge (Studinger, 2014) | 30 |

**Task Construction**. As shown in Fig.3, OmniEarth-Bench defines tasks across four hierarchical levels (L1-L4): L1 covers the seven domains based on established geophysical spheres: atmosphere, lithosphere, oceansphere, cryosphere, biosphere, human-activity sphere, and cross-sphere. L2 includes expert-approved, representative scenarios within each sphere, selected based on their scientific and practical value (e.g., earthquake prediction). Tab.2 illustrates representative scenarios covered by the L1 and L2 levels. Detailed descriptions of the L3 and L4 dimensions for each sphere are provided in the Appendix A.5 and Appendix A.6. L3 comprises four core abilities: Perception, General Reasoning, Scientific-Knowledge Reasoning, and CoT Reasoning. Perception and General Reasoning align with previous works such as MMBench (Liu et al., 2024) and XLRS-Bench (Wang et al., 2025), where Perception focuses on sensory inputs and Reasoning on inference. Scientific-Knowledge Reasoning addresses complex reasoning tasks requiring deep domain expertise in Earth sciences. CoT Reasoning evaluates the effectiveness of chain-of-thought processes within Earth science scenarios. L4 provides further granularity by subdividing tasks based on the L1-L3 dimensions. Achieving robust general intelligence in Earth sciences requires MLLMs to perform effectively across all hierarchical levels.

**Benchmark Construction.** For each of the six Earth spheres, this involved a dedicated team of 2-5 experts (Ph.D. holders or candidates) and 5-10 annotators (undergraduate and masters students). (1) For each sphere, evaluation dimensions were collaboratively defined by domain experts and MLLM specialists, ensuring high practical value and complexity. Cross-sphere tasks involved experts from multiple domains. This approach addresses the limitations observed when crowd-sourcing annotators proposed overly simplistic tasksfor example, Estimated Maximum Precipitation Level in atmosphere, which GPT-4o solved with 97.7% accuracy. Expert-led design ensures meaningful evaluation. (2) Experts were also responsible for defining data sources. For complex tasks, crowd-sourcing annotators struggled with downloading and aligning data (e.g., MODIS and GLASS from NASA). Thus, experts curated and organized datasets, with annotators assisting.

**Quality Control.** To ensure data integrity and task relevance, the quality control process involved two main steps. Cross-Validation: Annotator outputs were systematically compared against expert-provided annotation examples. Any discrepancies were flagged and reviewed by domain experts to ensure annotation correctness, especially for complex tasks involving multi-source data. Final Quality Assessment: specialists conducted thorough reviews to confirm that annotations adhered to expert standards and maintained consistency across all tasks and spheres. High-quality annotations were approved and incorporated into the dataset, while annotations that did not meet quality

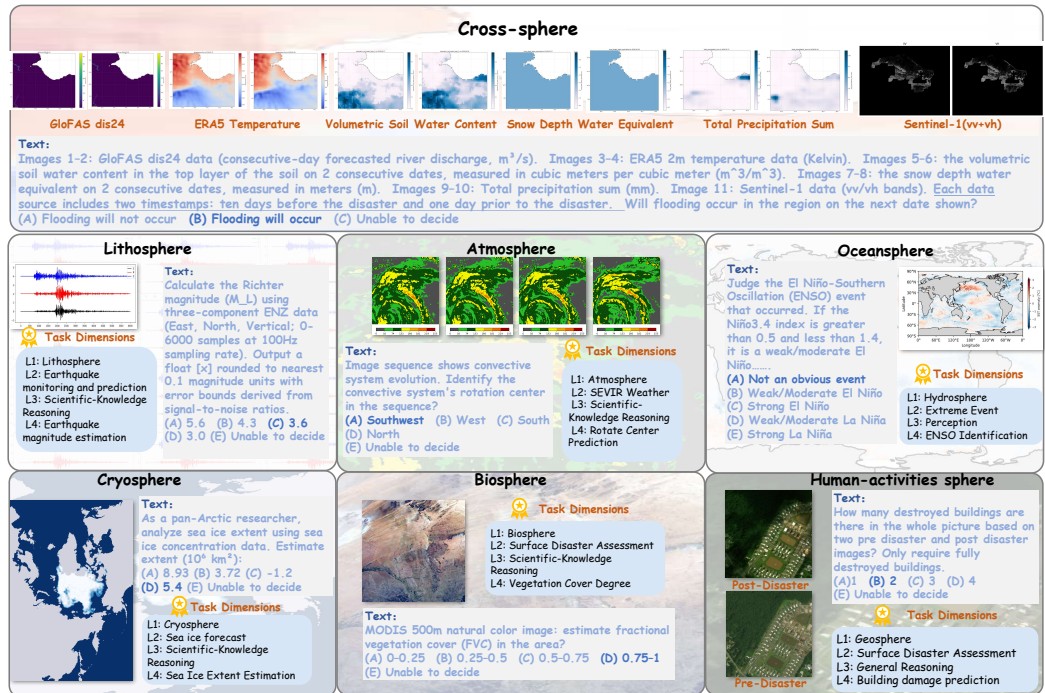

Figure 4: **Examples of OmniEarth-Bench.** OmniEarth-Bench comprises 109 unique L4 tasks, each with distinct questions, answers, and images.

standards underwent iterative refinement through a feedback loop involving annotators and expert supervision. This cyclical process ensured continuous improvement and maintained reliability.

## 3.2 TASK DIMENSIONS

OmniEarth-Bench defines tasks across four hierarchical levels (L1-L4), comprising 7 L1 dimensions, 25 L2 dimensions, 5 L3 dimensions, and 109 expert-defined L4 subtasks with real-world applicability. One representative L4 subtask from each L1 sphere is illustrated in Fig 4. We offer cross-sphere as an example of the L1-L4 divisions, with details for other spheres given in the Appendix A.7.

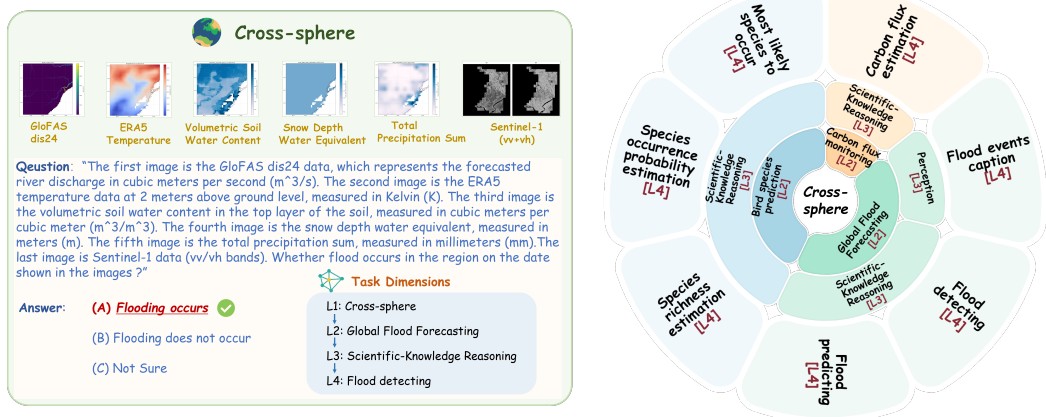

(a) An example in Cross-sphere.

(b) Task hierarchy of cross-sphere (L1-L4 dimension).

Figure 5: **Details of Dimension in Cross-sphere.** Cross-sphere has 3 L-2 dimensions, 2 L-3 dimensions and 7 L-4 dimensions.

Cross-sphere tasks in Earth science carry high practical and societal importance (Di Giuseppe et al., 2025; Dai et al., 2023). As shown in Fig 5, we select three representative L2 scenarios from socially impactful applications, including *Global Flood Forecasting (L2), Bird Species Prediction (L2) and Carbon Flux Monitoring (L2)*. Due to their reliance on expert knowledge and complex reasoning,

all are categorized as Scientific-Knowledge Reasoning (L3). Despite the complexity of cross-sphere scenarios, we successfully collaborated with domain experts to construct **7 high-value subtasks (L4 dimensions).** Further details on full tasks are available in the Appendix A.2. Overall, the L1-L4 divisions provide a coherent evaluation framework across Earth science spheres, with dimensions designed by domain experts to ensure high value.

### 3.3 STATISTICS AND ANALYSES

**Overview Statistics.** OmniEarth-Bench includes 109 expert-defined, high-value evaluation dimensions and 29,855 samples annotated by both experts and crowd-sourced contributors. As shown in Tab. 1, it offers clear advantages over existing benchmarks. Uniquely built on observational Earth science datarather than exam-style datasetsOmniEarth-Bench spans all six spheres and cross-sphere scenarios. Consistency metrics are reported in Tab. 3, with additional details and dimension-specific indicators provided in the Appendix A.2.

**Observational Data vs. Exam-questions Data.** Unlike subject-based benchmarks like ScienceQA (Saikh et al., 2022), which rely on exam questions or online learning problems followed by manual filtering, our approach takes a fundamentally different path. While such methods could theoretically span all six Earth spheres, they face two key limitations: (1) Benchmarks like ScienceQA focus on scientific inquiry rather than practical geoscience applications, limiting their real-world relevance. (2) Their evaluation dimensions are constrained by a bottom-up design: questions are derived from existing image-text pairs in papers, then filtered and categorized. In contrast, OmniEarth-Bench follows a top-down strategy: domain experts first define evaluation dimensions based on real-world geoscience challenges and data availability, then curate corresponding data. This ensures each task is both meaningful and grounded in practical utility.

Table 3: **Main statistics of OmniEarth-Bench**

| Statistic | Number |
|---|---|
| Total questions | 29,855 |
| - Cross-sphere | 3,456 |
| - Human-activity sphere | 9,362 |
| - Biosphere | 6,221 |
| - Atmosphere | 6,395 |
| - Lithosphere | 2,131 |
| - Oceansphere | 1,374 |
| - Cryosphere | 230 |
| Question Formats | |
| - Multiple-choice questions | 27,082 |
| - Open-ended questions | 27,082 |
| - Visual grounding questions | 2,697 |
| - CoT questions | 610 |
| - Images caption | 76 |
| MCQ | |
| - Single-image questions | 24,108 |
| - Multi-image questions | 5,671 |
| - Maximum question length | 213 |
| - Average question length | 48.2 |
| CoT | |
| - Total key step annotation | 3,473 |
| - Average key step annotation | 5.8 |
| - Average key step length | 14.8 |
| - Maximum question length | 101 |
| - Average question length | 50.2 |
| Caption | |
| - Average word length | 133.5 |
| - Average sentence length | 4.5 |

## 4 EXPERIMENT

### 4.1 EXPERIMENTAL SETUP.

The MLLMs evaluated on OmniEarth-Bench are grouped into two categories: (a) open-source VLMs, including Qwen2.5-VL (Wang et al., 2024a), LLava-Onevision (Li et al., 2024b), InterVL3 (Chen et al., 2023) and InternLM-XComposer-2.5 (Zhang et al., 2024b); (b) closed-source VLMs, such as GPT-4o (OpenAI, 2024), Gemini-2.0 (Team et al., 2023) and Claude 3.7 Sonnet (Anthropic, 2023). All models were evaluated using LMMs-Eval (Zhang et al., 2024a; Li et al., 2024a). Details of the evaluation are shown in Appendix A.4 and our code.

### 4.2 MAIN RESULTS

**All MLLMs exhibit suboptimal performance across all 7 spheres.** As illustrated in Tab. 4 and Tab. 7, nearly all MLLMs achieve accuracy rates below 40%, significantly underperforming relative to their success on traditional perception or reasoning benchmarks (Liu et al., 2024; Masry et al., 2022; Singh et al., 2019). Several factors likely contribute to this challenge. First, current multimodal large models are typically trained without domain-specific Earth science data, which impedes their ability to comprehend related queries. Second, many Earth science problems are inherently complex, particularly cross-domain prediction tasks that demand in-depth, specialized knowledge, which existing LLMs or MLLMs may not possess. Finally, OmniEarth-Bench provides

Table 4: **Experimental results on each sphere of VQA tasks, with models ranked by average performance.** '*Avg*' represents the average accuracy across sub-tasks. '*Experts*' means evaluation results of 100 examples in each sphere by experts. We mark the highest score of each metric in red , and the second highest underlined. Here, Cross., Atmo., Litho., Ocean., Cryo., Bio., and Human. stand for Cross-sphere, Atmosphere, Lithosphere, Oceansphere, Cryosphere, Biosphere, and Human-activity sphere.

| Method | Spheres (L1 dimensions) | | | | | | | Avg. |
| --- | --- | --- | --- | --- | --- | --- | --- | --- |
| | Cross. | Atmo. | Litho. | Ocean. | Cryo. | Bio. | Human. | |
| Experts | 90 | 96 | 91 | 95 | 93 | 97 | 95 | 93.4 |
| **Multiple choice question** | | | | | | | | |
| *Open-source MLLMs* | | | | | | | | |
| Claude-3.7-Sonnet (Anthropic, 2023) | 30.68 | 24.72 | 28.15 | 23.12 | 54.46 | 31.21 | 11.18 | 29.07 |
| Gemini-2.0 (Team et al., 2023) | 16.93 | 20.83 | 38.94 | 16.94 | 58.52 | 20.83 | 23.74 | 28.10 |
| GPT-4o (OpenAI, 2024) | 0.04 | 9.64 | 12.80 | 13.35 | 37.48 | 1.97 | 2.76 | 11.15 |
| *Open-source MLLMs* | | | | | | | | |
| InternVL3-72B (Zhu et al., 2025) | 19.19 | 33.98 | 23.39 | 20.22 | 74.56 | 31.99 | 29.46 | 33.26 |
| InternVL3-7B (Zhu et al., 2025) | 42.85 | 30.10 | 37.47 | 20.28 | 49.27 | 28.74 | 23.18 | 33.13 |
| LLaVA-Onevision-7B (Li et al., 2024b) | 19.26 | 33.69 | 28.72 | 24.54 | 46.40 | 37.31 | 30.62 | 31.51 |
| InternLM-XComposer-2.5-7B (Dong et al., 2024) | 19.78 | 17.45 | 28.88 | 21.06 | 40.04 | 30.67 | 24.76 | 26.09 |
| Qwen2.5-VL-7B (Bai et al., 2025) | 9.85 | 9.25 | 18.65 | 13.95 | 17.85 | 10.94 | 6.23 | 12.39 |
| Qwen2.5-VL-72B (Bai et al., 2025) | 3.92 | 4.82 | 22.43 | 16.27 | 5.88 | 14.91 | 8.63 | 10.98 |
| **Open-ended question** | | | | | | | | |
| *Closed-source MLLMs* | | | | | | | | |
| Gemini-2.0 (Team et al., 2023) | 31.48 | 38.10 | 41.67 | 24.97 | 61.49 | 27.33 | 31.85 | 36.70 |
| GPT-4o (OpenAI, 2024) | 25.76 | 23.21 | 33.13 | 25.17 | 46.46 | 13.65 | 17.17 | 26.36 |
| *Open-source MLLMs* | | | | | | | | |
| InternVL3-72B (Zhu et al., 2025) | 29.51 | 39.14 | 27.51 | 32.45 | 53.87 | 38.29 | 34.67 | 36.49 |
| Qwen2.5-VL-72B (Bai et al., 2025) | 24.78 | 22.08 | 38.62 | 31.17 | 15.23 | 20.22 | 16.87 | 24.14 |

high-resolution, intricate imagery, and the task of interpreting such complex visuals presents unique obstacles for MLLMs. This underscores the pressing need for specialized models or advanced post-training techniques to effectively address these challenges.

**CoT Performance.** Following the MME-CoT (Jiang et al., 2025), we leverage two interpretable metrics to evaluate the CoT correctness: recall and precision. The two metrics respectively attend to the two aspects of the CoT correctness: informativeness and accuracy. As shown in Tab. 5, InternVL3 outperformed Qwen2.5-VL and LLaVA-OneVision with the highest F1 score. Larger open-source variants showed superior performance, underscoring the scalability of model size.

Table 5: **CoT performance on OmniEarth-Bench**

| Models | LLaVA-OneVision-7B | Qwen2.5-VL-7B | InternVL3-8B | InternVL3-78B |
| --- | --- | --- | --- | --- |
| Precision | 89.83 | 92.72 | 94.02 | 94.74 |
| Recall | 23.41 | 29.12 | 34.47 | 35.5 |
| F1 | 37.14 | 44.32 | 50.45 | 51.65 |

Table 6: **Images caption performance on OmniEarth-Bench**

| Method | BLEU-1 | BLEU-2 | BLEU-3 | BLEU-4 | METEOR | ROUGE_L | CIDEr |
| --- | --- | --- | --- | --- | --- | --- | --- |
| *Closed-source MLLMs* | | | | | | | |
| Claude-3.7-Sonnet | 5.45 | 2.46 | 1.04 | 0.36 | 5.54 | 6.13 | 1.34 |
| GPT-4o | 5.05 | 2.61 | 1.43 | 0.86 | 6.57 | 6.98 | 0.00 |
| *Open-source MLLMs* | | | | | | | |
| Qwen2.5-VL-7B | 3.59 | 1.39 | 0.63 | 0.30 | 4.06 | 3.77 | 0.44 |
| Qwen2.5-VL-72B | 5.22 | 2.21 | 0.95 | 0.41 | 5.99 | 6.15 | 0.00 |

**Caption performance.** As shown in Tab. 6, the closed-source models, particularly GPT-4o, demonstrated the strongest overall performance. However, the large open-source model Qwen2.5-VL-72B achieved highly competitive results, significantly narrowing the gap. The substantial performance leap from the 7B to the 72B version of Qwen2.5-VL strongly underscores the effective scalability of model size for improving caption generation capabilities.

Table 7: **Visual grounding performance on OmniEarth-Bench.**

| Spheres | Metrics | GPT-4o | Gemini-2.0 | Claude-3.7-Sonnet | Qwen2.5-VL | LLaVA-OneVision | InternVL3 8B | InternVL3 78B |
| --- | --- | --- | --- | --- | --- | --- | --- | --- |
| Human-activity sphere | Acc@0.5 | 0.02 | 0.03 | 0 | 0.59 | 0.2 | 0 | 2.36 |
| | Acc@0.7 | 0 | 0 | 0 | 0 | 0 | 0 | 0.2 |
| Lithosphere | Acc@0.5 | 0.08 | 0.13 | 0.02 | 5.3 | 0 | 8.94 | 4.3 |
| | Acc@0.7 | 0 | 0.04 | 0 | 0.33 | 0 | 1.66 | 0.33 |
| Oceansphere | Acc@0.5 | 0.12 | 0.34 | 0.2 | 1.81 | 1.51 | 6.63 | 13.86 |
| | Acc@0.7 | 0.01 | 0.06 | 0.07 | 0 | 0 | 0.6 | 3.61 |

**Visual grounding performance.** As shown in Tab.7, visual grounding on OmniEarth-Bench is low across spheres and IoU thresholds: only InternVL3-78B leads in Oceansphere (Acc@0.5=13.86), and human-activity sphere is hardest (peak 2.36), with general-purpose MLLMs near zero. The weakness stems from domain shift to earth observation data (large-scale variation, clutter, small/elongated targets) and insufficient multi-scale alignment, which inflate errors at higher IoU.

### 4.3 Further Analysis

**MCQ-style vs. Open-ended evaluation.** Results in Tab. 4 show that open-ended evaluation often yields higher scores, as models are free to respond in natural language without being constrained by pre-defined choices. This setting better reflects real-world usage, where models are expected to articulate answers directly. In contrast, MCQ-style evaluation offers a more standardized and objective framework. By providing fixed answer choices, it reduces ambiguity in scoring and ensures comparability across models. The inclusion of an Unable to decide option further mitigates the risk of models producing answers that appear reasonable but are logically inconsistent. As a result, while open-ended evaluation captures the flexibility of natural language reasoning, MCQ-style benchmarks remain valuable for delivering fair and rigorous assessments.

**Time-sensitive task.** The Earth's seven spheres encompass numerous temporally correlated tasks. ENSO, a key climate mode influencing global weather extremes via teleconnections (Timmermann et al., 2018), has seen improved forecasts through domain-specific AI models (Ham et al., 2019; Guo et al., 2024). As shown in Fig. 6, prediction accuracy declines with longer lead times, echoing the limitations of specialized models. However, the performance of MLLMs still lags behind tailored models. Performance drops further for Indian Ocean Dipole (IOD) predictions, aligning with challenges faced by existing methods (Liu et al., 2021).

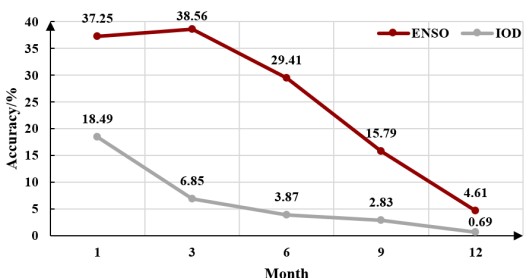

Figure 6: **GPT-4o performance on ENSO and IOD prediction with different lead months (previous).**

**Limited Gains from Scaling Model Size on Earth-Science Tasks.** In Table 4, we evaluate two sizes of the InterVL3 model and find that the 72B InterVL3 does not provide a significant advantage over the 7B model in our benchmarks, with performance even declining in some evaluation metrics. This contrasts with the substantial improvements observed in general-domain tasks. This performance bottleneck likely stems from the lack of Earth-science-specific knowledge, rather than a limitation in model capacity. Even large MLLMs struggle to reason about unfamiliar scientific concepts without targeted training on domain-specific data. These findings highlight the importance of prioritizing the integration of domain-specific knowledge in future Earth-science MLLM development, rather than merely increasing model size.

**Impact of Model Safety on Results.** In Tab. 4, we observe that Qwen2.5-VL and GPT-4o perform very poorly, even falling below the level of random guessing. However, this does not mean that these two models have the worst perceptual and science-related abilities. We observe that these models tend to refuse to answer when they are uncertain, whereas InternVL3 and LLaVA-Onevision randomly guess an answer. For instance, Qwen2.5-VL-72B refused to answer 18,258 questions. This safety mechanism in the models leads to the poor performance of Qwen2.5-VL and GPT-4o.

## 5 Conclusion

We have introduced OmniEarth-Bench, a foundational multimodal benchmark and the first to establish a systematic evaluation across all six spheres of the Earth system (atmosphere, lithosphere, Oceansphere, cryosphere, biosphere, and human-activity sphere) along with their cross-sphere interactions. This benchmark introduces 109 expert-curated evaluation dimensions and four hierarchical levels of reasoning (perception, general reasoning, expert-knowledge reasoning, and chain-of-thought reasoning), representing a novel and rigorous evaluation design for geoscientific MLLMs. Our results show that even state-of-the-art MLLMs (e.g., Claude) struggle with OmniEarth-Bench; none of the tested models surpassed 35% accuracy. This stark performance gap underscores the benchmarks difficulty and exposes fundamental limitations in current models geoscientific understanding. We anticipate that OmniEarth-Bench will serve as a catalyst for future research in geoscientific AI, guiding the development of models capable of expert-level analysis across Earths spheres and enabling advanced applications in environmental monitoring and climate science.

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

## A APPENDIX

### A.1 OVERVIEW OF THE APPENDIX

This appendix supplements the proposed **OmniEarth-Bench** with details excluded from the main paper due to space constraints. The appendix is organized as follows:

- Sec. A.2: More details of OmniEarth-Bench.
- Sec. A.3: Data Processing Strategies for each Spheres.
- Sec. A.4: Construction Details of Different Question Formats.
- Sec. A.5: Detailed results of specific sub-tasks (L-4 dimension).
- Sec. A.6: Detailed results of specific sub-tasks (L-3 dimension).
- Sec. A.7: Visualizations of samples and challenging cases.
- Sec. A.8: Datasheets for the OmniEarth-Bench dataset.
- Sec. A.9: Discussion on limitations and societal impact.
- Sec. A.9: Usage of LLM.

### A.2 MORE DETAILS OF OMNIEARTH-BENCH

This section provides additional details about the dataset. We begin with a visual illustration (Fig.4) that highlights the structure and real-world applicability of OmniEarth-Bench, presenting a curated example from each scientific sphere to show how the benchmark spans from high-level domains (L1) down to specific, expert-defined tasks (L4). To complement this overview, Table 8 and Table 9 present detailed statistics for each L4 dimension, along with their relationships to the L3 and L2 dimensions, fully clarifying the datasets structure and composition.

**Experiments setup.** Following MMBench (Liu et al., 2024) and MME-Realworld (Zhang et al., 2024d) methods, in the MCQ format of the VQA task, we manually created 5 options for each question: one correct answer, three distractors, and one special answer (unable to answer). We evaluated the accuracy and reported the L-1 dimension for the VQA task, with L-3 and L-4 results available in the Appendix A.5 and Appendix A.6. All scores in Tab. 4 are reported as percentages (%). For the Grounding task, we used precision, assessing accuracy based on the intersection between predicted and ground truth bounding boxes, with predictions deemed correct if IoU exceeds a threshold (0.5 and 0.7). For open-ended evaluation, we use an external LLM as a judge to automatically assess the correctness of the generated answers. Following the XLRS-Bench (Wang et al., 2025), to evaluate the quality of the generated captions, we employed a comprehensive suite of standard metrics, including BLEU, METEOR, ROUGE_L, and CIDEr.

**Human Annotations vs. GPT Annotations.** All annotations are finished by experts and crowd-sourcing annotators. Unlike MMBench Liu et al. (2024), we did not use tools like GPT-4o OpenAI (2024). It was driven by two key reasons: (1) GPT-4o cannot generate VQA data requiring deep domain expertise. Tasks under the Scientific-Knowledge Reasoning (L3) demand substantial background knowledge and must be constructed collaboratively by experts. (2) Although GPT-4o can generate samples for general perception or simple reasoning tasks, expert evaluation found the data to be low quality and insufficiently challenging. For example, in the visual grounding task, GPT-4o only detects highly salient structures, failing to support our goal of testing MLLMs on locating diverse buildings across complex scenes. As a result, all OmniEarth-Bench data was exclusively created by experts and annotators.

### A.2.1 CROSS-SPHERE

- L2-Global Flood Forecasting:
    - **Flood Detecting**: Predicts whether a flood event will occur in the near future based on ground and atmospheric variables, including river discharge, 2-meter air temperature, top-layer volumetric soil water content, snow depth water equivalent, total precipitation, along with Sentinel VV / VH data from the preceding two days.

- **Flood Predicting**: Predicts whether a flood event will occur in the near future based on the same variables used in Flood Detecting, along with Sentinel VV / VH data from the preceding two days.

- L2-Carbon flux monitoring:

  - **Carbon flux estimation:** Refers to the process of quantifying the rate and direction of carbon exchange (e.g., carbon dioxide) between the biosphere (vegetation and microorganisms, etc.) and the atmosphere, which tests the LLM's ability to interpret biogeochemical cycles, integrate multi-dimensional environmental data (e.g., satellite, sensor networks), and apply physics-based or statistical models for climate change analysis.

- L2-Bird species prediction:

  - **Most likely species to occur:** Predicting the species with the highest likelihood of presence in a specific habitat based on environmental variables (e.g., climate, habitat type), testing the LLM's ability to analyze spatial-environmental correlations and prioritize species under data-driven constraints.

  - **Species occurrence probability estimation:** Quantifying the probability of a species being present in a given geographic area, evaluating the LLM's grasp of probabilistic reasoning and ecological variable weighting.

  - **Species richness estimation:** Calculating the total number of distinct species within a defined ecosystem or region, testing the LLM's capacity to integrate multi-modal data to predict biodiversity.

### A.2.2 HUMAN-ACTIVITY SPHERE

The human-activity sphere leverages remote sensing and mapping technologies across three key scenarios: *Urban Construction (L2), Land Use (L2), and Surface Disaster Assessment (L2)*. Urban construction supports planning and socio-economic analysis; land use classification underpins environmental monitoring and resource management; and disaster assessment enables rapid post-event response and risk mitigation. OmniEarth-Bench spans all four L3 capability dimensions in the human-activity spherePerception, General Reasoning, Scientific-Knowledge Reasoning, and CoT Reasoningwith **27 subtasks (L4 dimensions)**, surpassing all existing benchmarks in this domain Li et al. (2024d); Wang et al. (2025). The complete task details are as follows.

- L2-Surface Disaster Assessment:

  - **Change detection counting of post-disaster completely destroyed building:** Compares pre- and post-disaster images to count fully destroyed buildings, evaluating temporal change detection.

  - **Counting of post-disaster partially damaged building:** Detects and counts lightly or moderately damaged structures in post-disaster imagery.

  - **Building damage prediction:** Estimates potential damage severity from pre-disaster views, testing risk assessment without ground truth.

  - **Disaster prediction:** Predicts future disaster types using current imagery, evaluating temporal modeling capabilities.

  - **Disaster type classification:** Identifies disaster types (e.g., flood, earthquake) from satellite images, testing visual pattern recognition.

  - **Geolocation estimation of disaster-affected regions from imagery:** Predicts the geographic location of affected areas based on visual cues, assessing spatial referencing.

  - **Individual building damage assessment:** Compares pre- and post-event imagery to evaluate building-level structural changes.

  - **Multi-image individual visual localization task:** Uses multi-temporal or multi-view images to locate specific buildings, assessing multi-view reasoning.

  - **Spatial relationships under complex conditions:** Infers spatial relations (e.g., relative position, containment) between objects in imagery, testing 3D reasoning.

  - **Visual grounding of damaged individual buildings:** Locates damaged structures in post-disaster imagery, evaluating anomaly detection and spatial precision.

- L2-Urban Development:

  - **Fine-grained object type recognition:** Classifies specified buildings in high-resolution imagery, testing the models ability to distinguish visually similar structures.
  - **Overall counting:** Counts all buildings or urban facilities in an image, evaluating object detection and counting under complex conditions.
  - **Counting under complex conditions:** Counts objects that meet given conditions (e.g., attributes or constraints), testing constrained multimodal reasoning.
  - **Overall building height estimation:** Estimates structural vertical dimensions using multi-source geospatial inputs, assessing 3D reconstruction accuracy and cross-sensor measurement consistency.
  - **Individual building height estimation:** Estimates the height of an individual building from satellite views, testing 2D-to-3D inference.

- L2-Land Use:

  - **Overall land type classification:** Identifies macro land cover types (e.g., urban, farmland, water), evaluating scene-level understanding.
  - **Fine-grained land type classification:** Classifies specific land use (e.g., crop types) at finer scales, testing detailed semantic discrimination.
  - **Visual localization of land use types:** Locates specific land types within an image, evaluating spatial perception.
  - **Counting of land types under complex conditions:** Counts land use regions meeting complex conditions, assessing constrained visual reasoning.
  - **Visual groudning of land types:** Locates specific land types to evaluate the model's visual localization capability and land type classification ability.

### A.2.3 BIOSPHERE

We present a biosphere-focused MLLM benchmark built on observational data and retrieval products, featuring **15 practical L4 subtasks**. It includes four representative L2 scenarios: Vegetation Monitoring (L2), Human Footprint Assessment (L2), Environmental Pollution Monitoring (L2), Species Distribution Prediction (L2), and Crop Growth Monitoring (L2). Vegetation Monitoring Crowther et al. (2015) evaluates plant and ecosystem health to support function assessment, carbon accounting, and climate response. Human Footprint Assessment Venter et al. (2016) quantifies human impact on nature, informing sustainability and biodiversity strategies. Environmental Pollution Monitoring Nurseitov et al. (2024b) identifies pollution events and their extent, guiding environmental policy and mitigation. Species Distribution is a key concern in the biosphere, as it guides biodiversity conservation and supports modeling species range shifts under climate and land-use change. Crop Growth Monitoring Zheng et al. (2021b) assesses crop health for precision agriculture and sustainable farming. The complete task details are as follows.

- L2-Crop growth monitoring:

  - **Dead oil palm identification:** Identifies dead trees in unmanned aerial vehicle (UAV) imagery, testing the models domain knowledge in crop growth.
  - **Dead oil palm counting:** Counting the number of dead trees in an image, testing the models object counting capability.

- L2-Environmental pollution monitoring:

  - **Terrestrial oil spill counting:** Counting oil spill points in satellite imagery, testing the models ability in environmental pollution recognition and object counting.
  - **Terrestrial oil spill area calculation:** Calculating the total area of oil spills in the image, evaluating the model's applicability in pollution events.

- L2-Human footprint assessment:

  - **Human footprint assessment:** Assessing the impact of human activities in the region based on imagery, testing the models ability to recognize and reason about human activity features
  - **Human footprint index estimation:** Calculating the human footprint index of a region, testing the models understanding of human activity patterns.

- L2-Species Distribution Prediction:

  - **Tree species prediction:** Identifying the type of tree that occupies the largest proportion, testing the models ability to recognize features of different tree species.
  - **Tree species proportion prediction:** Identifying the proportion of specific tree species, testing the models ability in species recognition and statistical reasoning.
  - **Animal classification:** Identifying animal species within the bounding box, testing the models ability to extract local information and distinguish between different animal species.
  - **Geographical location inference of plant species:** Inferring the geographic coordinates from the image and the given tree species, testing the models domain knowledge of tree species distribution.
  - **Global animal counting:** Counting the number of animals in the image, testing the models ability in animal instance extraction and counting.
  - **Species distribution prediction:** Predicting the likely animal species in a region based on the image and geographic coordinates, testing the models ability to extract ecological features and its knowledge of species distribution.

- L2-Vegetation monitoring:

  - **Fractional vegetation cover estimation:** Calculating the fractional vegetation cover in the image, testing the models ability to recognize vegetation features.
  - **Leaf area index estimation:** Calculating the leaf area index from multi-band imagery, testing the models ability to comprehensively understand and utilize multi-source information.
  - **Peak vegetation coverage area grounding:** Locating peak vegetation coverage areas in the image using multi-band imagery, testing the models ability to localize vegetation features.

### A.2.4 ATMOSPHERE

The atmosphere is a key domain in Earth sciences with high practical value and extensive research interest Gong et al. (2024); Stock et al. (2024). While existing benchmarks target specific atmospheric sub-scenarios Ma et al. (2024); Chen et al. (2024); Li et al. (2024c), they lack comprehensive domain-wide coverage. OmniEarth-Bench addresses this gap by defining evaluation dimensions across six representative scenarios using data from ERA5 Hersbach et al. (2020), SEVIR Veillette et al. (2020), and Typhoon Kitamoto et al. (2023) datasets: *Short-term Weather Events (L2), Medium-term Weather Events (L2), Seasonal Weather Events (L2), Interannual Climate Variation (L2), Typhoon Event (L2), and SEVIR Weather (L2).* For example, the *Typhoon Event* dimension serves as a flagship benchmark for atmospheric machine learning, supporting operational hazard forecasting and advancing research on tropical cyclone intensity and structure. These six scenarios (L2) span **36 expert-designed subtasks (L4 dimensions)** with strong real-world relevance, substantially surpassing existing atmospheric benchmarks. The complete task details are as follows.

- L2-Short-term weather events:

  - **Event intensity identification:** Determine extreme intensity or variable value at given position or region.
  - **Event localization:** Localize event center or moving direction of event.
  - **Event trend analysis:** Determine varying trend or speed of variable.
  - **Event type identification:** Determine type of current event.
  - **Dynamic feature identification:** Determine dynamic structure via multi-variable analysis.
  - **Event evolution analysis:** Determine stage of event via multi-variable analysis.
  - **Thermodynamic feature identification:** Determine thermodynamic features or structure via multi-variable analysis.

- L2-Medium-term weather events:

  - **Cyclone movement identification:** Determine moving direction of cyclone.
  - **Cyclone phase identification:** Determine the different phase of cyclone

- **Event intensity identification:** Determine extreme intensity or variable value at given position or region.
- **Event localization:** Localize event center or moving direction of event.
- **Event trend analysis:** Determine varying speed or trend of current event.
- **Geopotential pattern identification:** Determine pattern / structure of given geopotential.
- **Moisture flux analysis:** Determine intensity of moisture flux transformation.
- **System identification:** Determine dynamic structure via multi-variable analysis.
- **System evolution trend analysis:** Determine the evolution stage of the current system via multi-variable analysis.

- L2-Typhoon:

    - **Pressure estimation:** Using the same image stacks, the task outputs the minimum sealevel pressure (hPa) at the cyclone eye; this complements wind speed and enables pressurewind relationship validation.
    - **Radius of major gale axis estimation:** Using scatterometerderived peakgust layers, the model regresses the semimajor radius (km) of 50kt gusts, characterising the reach of damaging winds for early warning.
    - **Radius of major storm axis estimation:** From the segmented windfield map, the model estimates the semimajor radius (km) of 34kt galeforce winds, quantifying the storms main spatial extent and directly supporting surgerisk assessment.
    - **Radius of minor gale axis estimation:** Outputs the corresponding semiminor radius, enabling a complete 2D description of the gust envelope.
    - **Radius of minor storm axis estimation:** Analogous to the above, but for the semiminor radius, capturing asymmetric size features critical to trackshift sensitivity analysis.
    - **Wind estimation:** Given timesynchronised multispectral satellite images, models must regress the stormcentre 1min sustained surface wind speed in kt, providing a physicsconsistent proxy for SaffirSimpson intensity classification.

- L2-Seasonal weather events:

    - **Precipitation anomaly identification:** Determine precipitation anomaly value at given timestamp or region.
    - **Seasonal comparison:** Analysis of temperature/precipitation anomaly within a year.
    - **Temperature anomaly identification:** Determine temperature anomaly value at given timestamp or region.

- L2-Interannual climate variation:

    - **ENSO feature analysis:** Determine status or features of ENSO.
    - **Long-term Precipitation trend analysis:** Determine trend of precipitation anomaly among years.
    - **Long-term Temperature trend identification:** Determine trend of temperature anomaly among years.

- L2-SEVIR Weather:

    - **Event type prediction:** Identifies storm event types based on visible and infrared channels from satellite, along with Vertical Integrated Liquid (VIL) data from weather radar.
    - **Miss alarm estimation:** Estimates the miss rate by comparing forecasted outputs against SEVIR ground truth.
    - **Movement prediction:** Given a sequence of VIL data, MLLMs are required to identify the move direction of the convective system.
    - **Rotate center prediction**

### A.2.5   LITHOSPHERE

We firstly construct an MLLM benchmark for the lithosphere based on observational data, comprising **8 practical subtasks (L4 dimensions) .** We define two representative L2 scenarios within

the lithosphere: *Seismic Monitoring and Prediction (L2) and Geophysical Exploration (L2).* Seismic monitoring and prediction Allen & Melgar (2019), a critical domain in geosciences, aims to uncover Earths internal dynamics and earthquake nucleation mechanisms, forming a theoretical basis for early warning and disaster mitigation. Geophysical exploration imaging Yu & Ma (2021), by analyzing subsurface responses to physical fields such as seismic waves, electromagnetic fields, and gravity/magnetic anomalies, enables high-resolution geological modeling essential for understanding subsurface structures, hydrocarbon exploration, and geological hazard assessment. The complete task details are as follows.

- L2-Earthquake monitoring and prediction:
    - **P-wave phase picking:** Taking three-component observed seismic waveforms as input, output the arrival times of the P-wave characteristic seismic signals.
    - **S-wave phase picking:** Taking three-component observed seismic waveforms as input, output the arrival times of the S-wave characteristic seismic signals.
    - **Earthquake or noise classification:** Distinguishing seismic signals from natural earthquakes versus artificial noise or vibrations, testing the LLM's understanding of geophysical signal patterns, noise discrimination, and time-series data analysis.
    - **Earthquake magnitude estimation:** Determine the earthquake magnitude based on the seismic amplitude at the location of the S-wave seismic phase.
    - **Earthquake source-receiver distance inference:** Single-station earthquake location is simplified to determining the distance from the earthquake hypocenter to the geophone through the distance between the P-wave and S-wave seismic phases.

- L2-Geophysics imaging:
    - **Salt body detection:** Identifying subsurface salt dome structures in geological or seismic data, testing the LLM's domain knowledge in geophysics, spatial pattern recognition, and geological feature interpretation.
    - **salt body location:** Precisely determining the spatial coordinates or depth of salt bodies within geological formations, evaluating the LLM's capability in spatial reasoning, multi-dimensional data integration, and quantitative analysis accuracy.

A.2.6 OCEANSPHERE

We build a multi-layer MLLM benchmark for the oceansphere based primarily on satellite and analysis data products, featuring **8 practical L4 subtasks**. This domain includes three representative L2 scenarios: *Marine Oil Spills and Debris Monitoring (L2), Extreme Oceanic Events Warning (L2), and Ocean Phenomena Detection (L2).* The Marine Oil Spills and Debris Monitoring Al-Ruzouq et al. (2020) scenario uses multi-source remote sensing and in situ water quality data to track the spatial distribution and temporal dynamics of oil contamination and floating debris, supporting environmental management and emergency response. The Extreme Oceanic Events Warning Ham et al. (2019); Ling et al. (2022) scenario targets the detection and prediction of major climate modes such as El NiñoSouthern Oscillation (ENSO) and the Indian Ocean Dipole (IOD), aiming to mitigate their societal and economic impacts. The Ocean Phenomena Detection scenario Duo et al. (2019); Koračin et al. (2014) involves identifying ocean features like eddies and marine fog, which are key for maritime safety and ecological studies.

- L2-Extreme Events:
    - **Enso identification:** Critical for mitigating global climate extremes, this task classifies ENSO events (e.g., El Niño and La Niña) by analyzing the Pacific SST anomaly maps.
    - **Iod identification:** Essential for monsoon forecasting and reducing compound risks in Indian Ocean nations, this task identifies Indian Ocean Dipole phases (positive/negative) from SST anomalies, similar to ENSO identification.
    - **Enso forecast:** As a complement to ENSO identification, this task predicts whether or what type of ENSO event will occur several months ahead using global SST anomaly maps of the past six months, which requires the model to capture the temporal evolution process.

– **Iod forecast:**As a complement to IOD identification, this task predicts IOD event occurrence and type, similar to ENSO forecasting.

• L2-Phenomenon Detection:

– **Eddy identification:** Fundamental for marine ecosystem management, this task identifies eddy types (cyclonic/anticyclonic) from the chlorophyll grayscale satellite image.

– **Marine fog detection:**Critical for maritime safety and intelligent navigation, this task identifies fog presence via satellite imagery.

– **Eddy Localization :** As a complement to eddy localization, this task detects the location of eddies, enhancing search-and-rescue operations and pollution mitigation.

• L2-Marine Debris and Oil Pollution:

– **Marine Pollution Type Classification:** Marine Pollution Type Classification refers to the scientific method of systematically categorizing marine pollutants based on their sources or characteristics (e.g., oil spills, plastic waste, chemical discharges)," which can test an LLM's domain knowledge in environmental science, multi-category semantic comprehension, and fine-grained classification capabilities.

### A.2.7 CRYOSPHERE

We conduct an MLLM benchmark for the cryosphere primarily based on sea ice reanalysis data, glacial imagery, and graphical plots, incorporating **8 practical L4 subtasks**. We identify two representative L2 scenarios within the cryosphere: *Sea Ice Forecasting (L2) and Glacier Analysis (L2)*. Sea ice forecasting focuses on predicting the dynamic changes of sea ice in polar regions. Arctic sea ice is crucial for understanding global climate change Budikova (2009); Zhou et al. (2024). Its continuous decline over the last few decades has made sea ice forecasting significant for navigating through the Arctic Ocean during melting seasons. Moreover, the loss of the Antarctic sea ice would greatly impact the global sea level. Glacier analysis Khan et al. (2022); Chudley et al. (2025), aims to study the glacial movements and changes of glaciers over time. The complete task details are as follows.

• L2-Glacier analysis:

– **Glacial Lake Recognition:** A melting glacier could result in multiple glacial lakes. Identifying them could provide valuable information for analyzing the variation trend of glaciers. We provide the model with images of glacial lakes, glaciers, and regular lakes. The model is asked to output the quantity of glacial lakes. This L4 task not only assesses the model's ability to identify glacial lakes from the others, it also assesses whether the model is capable of accurately reasoning about the overall quantities.

– **Glacier Melting Estimation:** To evaluate the model's ability to analyze glacier data, we first present the model with the observation of glacier melting rate data and a sample chart showing the correlation between the melting rate and displayed color. Then, we provide two predictive charts from different models, and the model is required to identify which one better matches the provided real-world observation. Additionally, we show the model images of different glaciers at various times to see if it can determine which glacier is more likely to be in a melting state.

– **Slide Recognition:** This task is designed to assess the model's ability to determine glacier landslide risks. First, we show it images of different glaciers and ask it to judge which one is more likely to experience a landslide based on their melting conditions. Then, we provide traverse and longitudinal melting profiles showing the melting rates and thickness of glaciers at different locations in Greenland, and ask the model to determine which glacier is more prone to landslides.

• L2-Sea ice forecast:

– **SIC Estimate SIT:** To further test the model's ability to analyze sea ice concentration data, we provide it with a sea ice thickness variation trend chart, and sea ice concentration data from a date following the last point on that chart. The model is instructed to forecast the sea ice thickness of the following day based on inputs.

- **SIC Estimate SIV:** To explore the model's ability to analyze sea ice concentration data, we provide it with a sea ice volume variation trend chart, and sea ice concentration data from a date following the last point on that chart. We then assess whether the model can accurately forecast the sea ice volume for the following day based on those two inputs.

- **SIT Trend Prediction:** In this L4 task, we further evaluate the model's ability to directly analyze the trend data and make reasonable forecasts. Similarly, we provide the model with the previous year's sea ice thickness variation curve and the trend up to a certain point in the following year. Then, the model is required to predict subsequent sea ice thickness according to input data.

- **SIV Trend Prediction:** In this L4 task, we provide the model with the previous year's sea ice volume variation curve and the trend up to a certain point in the following year, to test the model's ability to make short-term forecasts of sea ice volume based on given data.

- **Sea Ice Extent Estimation:** Questions in this L4 task are designed to assess the model's ability to distinguish between Antarctic and Arctic sea ice, determine the melting season, i.e., evaluate the sea ice extent changes over time, and estimate the sea ice extent area from a given sea ice concentration map.

Table 8: **Characteristics of each task (L4 dimension) in human-activity sphere, Biosphere, Cross-sphere and Biosphere.** Human-activity sphere has 27 subtasks, Biosphere has 15 subtasks, Cross-sphere has 7 subtasks, Lithosphere has 8 subtasks.

| L1 | L2 | L3 | L4 Task Description | Format | Samples | Answer Type |
|---|---|---|---|---|---|---|
| Human-activity sphere | Surface Disaster | Perception | Change detection counting of post-disaster completely destroyedbuilding | VQA | 502 | Multiple Choice/Open-ended |
| | | | Counting of post-disaster partially damaged building | VQA | 498 | Multiple Choice/Open-ended |
| | | | Landslide events caption | Images Caption | 4 | Caption |
| | | General Reasoning | Building damage prediction | VQA | 499 | Multiple Choice/Open-ended |
| | | | Disaster prediction | VQA | 500 | Multiple Choice/Open-ended |
| | | | Dissaster type classification | VQA | 500 | Multiple Choice/Open-ended |
| | | | Geolocation estimation of disaster-affected regions from imagery | VQA | 502 | Multiple Choice/Open-ended |
| | | | Individual building damage assessment | VQA | 107 | Multiple Choice/Open-ended |
| | | | Multi-image individual visual localization task | VQA | 102 | Multiple Choice/Open-ended |
| | | | Spatial relationships under complex conditions | VQA | 99 | Multiple Choice/Open-ended |
| | | | Visual grounding of damaged individual buildings | VQA | 102 | Multiple Choice/Open-ended |
| | | CoT | Individual building damage assessment | VQA/CoT | 107 | Multiple Choice/Open-ended |
| | | | Multi-image individual visual localization task | VQA/CoT | 102 | Multiple Choice/Open-ended |
| | | | Spatial relationships under complex conditions | VQA/CoT | 99 | Multiple Choice/Open-ended |
| | | | Visual grounding of damaged individual buildings | VQA/CoT | 102 | Multiple Choice/Open-ended |
| | Urban Development | Perception | Fine-grained object recognition | VQA | 514 | Multiple Choice/Open-ended |
| | | | Overall counting | VQA | 502 | Multiple Choice/Open-ended |
| | | General Reasoning | Counting under complex conditions | VQA | 754 | Multiple Choice/Open-ended |
| | | | Individual building height estimation | VQA | 101 | Multiple Choice/Open-ended |
| | | | Overall building height estimation | VQA | 100 | Multiple Choice/Open-ended |
| | | CoT | Individual building height estimation | VQA/CoT | 101 | Multiple Choice/Open-ended |
| | | | Overall building height estimation | VQA/CoT | 100 | Multiple Choice/Open-ended |
| | Land Use | Perception | Overall land type classification | VQA | 509 | Multiple Choice/Open-ended |
| | | | Fine-grained land type classification | VQA | 509 | Multiple Choice/Open-ended |
| | | | Visual groudning of land types | Visual Grounding | 508 | Bounding Box |
| | | | Visual localization of land use types | VQA | 509 | Multiple Choice/Open-ended |
| | | General Reasoning | Complex land counting | VQA | 449 | Multiple Choice/Open-ended |
| Biosphere | Crop growth monitoring | Perception | Dead oil palm counting | VQA | 828 | Multiple Choice/Open-ended |
| | | | Dead oil palm identification | VQA | 828 | Multiple Choice/Open-ended |
| | Environmental pollution monitoring | Perception | Terrestrial oil spill area calculation | VQA | 123 | Multiple Choice/Open-ended |
| | | | Terrestrial oil spill counting | VQA | 123 | Multiple Choice/Open-ended |
| | Human footprint assessment | Scientific-Knowledge Reasoning | Human footprint assessment | VQA | 300 | Multiple Choice/Open-ended |
| | | | Human footprint index estimation | VQA | 300 | Multiple Choice/Open-ended |
| | Species Distribution Prediction | Perception | Tree species prediction | VQA | 500 | Multiple Choice/Open-ended |
| | | | Tree species proportion prediction | VQA | 500 | Multiple Choice/Open-ended |
| | | Expert- knowledge Deductive Reasoning | Animal classification | VQA | 108 | Multiple Choice/Open-ended |
| | | | Geographical location inference of plant species | VQA | 500 | Multiple Choice/Open-ended |
| | | | Global animal counting | VQA | 110 | Multiple Choice/Open-ended |
| | | | Species distribution prediction | VQA | 1000 | Multiple Choice/Open-ended |
| | Vegetation monitoring | Scientific-Knowledge Reasoning | Fractional vegetation cover estimation | VQA | 300 | Multiple Choice/Open-ended |
| | | | Leaf area index estimation | VQA | 300 | Multiple Choice/Open-ended |
| | | | Peak vegetation coverage area grounding | VQA | 300 | Multiple Choice/Open-ended |
| Cross-sphere | Bird species prediction | Scientific-Knowledge Reasoning | Most likely species to occur | VQA | 900 | Multiple Choice/Open-ended |
| | | | Species occurrence probability estimation | VQA | 453 | Multiple Choice/Open-ended |
| | | | Species richness estimation | VQA | 900 | Multiple Choice/Open-ended |
| | Carbon flux monitoring | Scientific-Knowledge Reasoning | Carbon flux estimation | VQA | 330 | Multiple Choice/Open-ended |
| | Global Flood Forecasting | Scientific-Knowledge Reasoning | Flood detecting | VQA | 596 | Multiple Choice/Open-ended |
| | | | Flood predicting | VQA | 277 | Multiple Choice/Open-ended |
| | | | Flood events caption | Images Caption | 28 | Caption |
| Lithosphere | Earthquake monitoring and prediction | Perception | P-wave phase picking | VQA | 300 | Multiple Choice/Open-ended |
| | | | S-wave phase picking | VQA | 300 | Multiple Choice/Open-ended |
| | | | Earthquake or noise classification | VQA | 300 | Multiple Choice/Open-ended |
| | | Scientific-Knowledge Reasoning | Earthquake magnitude estimation | VQA | 300 | Multiple Choice/Open-ended |
| | | | Earthquake source-receiver distance inference | VQA | 300 | Multiple Choice/Open-ended |
| | Geophysics imaging | Perception | Salt body location | Visual Grounding | 302 | Bounding Box |
| | | | Salt body detection | VQA | 329 | Multiple Choice/Open-ended |
| | Volcanic activity | Perception | Volcanic activity caption | Images Caption | 2 | Caption |

Table 9: **Characteristics of each task (L4 dimension) in Atmosphere, Oceansphere and Cryosphere.** Atmosphere has 36 subtasks, Oceansphere has 8 subtasks, Cryosphere has 8 subtasks.

| L1 | L2 | L3 | L4 Task Description | Format | Samples | Answer Type |
|---|---|---|---|---|---|---|
| Atmosphere | Interannual climate variation | Perception | ENSO feature analysis | VQA | 86 | Multiple Choice/Open-ended |
| | | | Long-term Precipitation trend analysis | VQA | 50 | Multiple Choice/Open-ended |
| | | | Long-term Temperature trend identification | VQA | 51 | Multiple Choice/Open-ended |
| | Medium-term weather events | Perception | Cyclone movement identification | VQA | 185 | Multiple Choice/Open-ended |
| | | | Cyclone phase identification | VQA | 90 | Multiple Choice/Open-ended |
| | | | C-WAVE Caption | Images Caption | 17 | Caption |
| | | | Event localization | VQA | 93 | Multiple Choice/Open-ended |
| | | | Event intensity identification | VQA | 594 | Multiple Choice/Open-ended |
| | | | Event onset identification | VQA | 42 | Multiple Choice/Open-ended |
| | | | Event trend analysis | VQA | 575 | Multiple Choice/Open-ended |
| | | | H-WAVE Caption | Images Caption | 17 | Caption |
| | | | Geopotential pattern identification | VQA | 33 | Multiple Choice/Open-ended |
| | | | Moisture flux analysis | VQA | 150 | Multiple Choice/Open-ended |
| | | | System identification | VQA | 231 | Multiple Choice/Open-ended |
| | | | Storm Caption | Images Caption | 8 | Caption |
| | | Scientific-Knowledge Reasoning | System evolution trend analysis | VQA | 91 | Multiple Choice/Open-ended |
| | SEVIR Weather | Scientific-Knowledge Reasoning | Event type prediction | VQA | 300 | Multiple Choice/Open-ended |
| | | | Miss alarm estimation | VQA | 300 | Multiple Choice/Open-ended |
| | | | Movement prediction | VQA | 200 | Multiple Choice/Open-ended |
| | | | Rotate center prediction | VQA | 93 | Multiple Choice/Open-ended |
| | Seasonal weather events | Perception | Precipitation anomaly identification | VQA | 75 | Multiple Choice/Open-ended |
| | | | Seasonal comparison | VQA | 101 | Multiple Choice/Open-ended |
| | | | Temperature anomaly identification | VQA | 75 | Multiple Choice/Open-ended |
| | Short-term weather events | Perception | Event intensity identification | VQA | 323 | Multiple Choice/Open-ended |
| | | | Event localization | VQA | 133 | Multiple Choice/Open-ended |
| | | | Event trend analysis | VQA | 297 | Multiple Choice/Open-ended |
| | | | Event type identification | VQA | 139 | Multiple Choice/Open-ended |
| | | Scientific-Knowledge Reasoning | Dynamic feature identification | VQA | 40 | Multiple Choice/Open-ended |
| | | | Event evolution analysis | VQA | 90 | Multiple Choice/Open-ended |
| | | | Thermodynamic feature identification | VQA | 40 | Multiple Choice/Open-ended |
| | Typhoon | Scientific-Knowledge Reasoning | Pressure estimation | VQA | 847 | Multiple Choice/Open-ended |
| | | | Radius of major gale axis estimation | VQA | 847 | Multiple Choice/Open-ended |
| | | | Radius of minor gale axis estimation | VQA | 847 | Multiple Choice/Open-ended |
| | | | Radius of major storm axis estimation | VQA | 847 | Multiple Choice/Open-ended |
| | | | Radius of minor storm axis estimation | VQA | 847 | Multiple Choice/Open-ended |
| | | | Wind estimation | VQA | 847 | Multiple Choice/Open-ended |
| Cryosphere | Glacier analysis | Perception | Glacial Lake Recognition | VQA | 12 | Multiple Choice/Open-ended |
| | | Scientific-Knowledge Reasoning | Glacier Melting Estimation | VQA | 10 | Multiple Choice/Open-ended |
| | | | Slide Recognition | VQA | 8 | Multiple Choice/Open-ended |
| | Sea ice forecast | Scientific-Knowledge Reasoning | SIC Estimate SIT | VQA | 20 | Multiple Choice/Open-ended |
| | | | SIC Estimate SIV | VQA | 20 | Multiple Choice/Open-ended |
| | | | SIT Trend Prediction | VQA | 30 | Multiple Choice/Open-ended |
| | | | SIV Trend Prediction | VQA | 30 | Multiple Choice/Open-ended |
| | | | Sea Ice Extent Estimation | VQA | 100 | Multiple Choice/Open-ended |
| Oceansphere | Extreme Events | Perception | Enso identification | VQA | 146 | Multiple Choice/Open-ended |
| | | | Iod identification | VQA | 140 | Multiple Choice/Open-ended |
| | | Scientific-Knowledge Reasoning | Enso forecast | VQA | 152 | Multiple Choice/Open-ended |
| | | | Iod forecast | VQA | 145 | Multiple Choice/Open-ended |
| | Marine Debris and Oil Pollution | Perception | Marine Pollution Type Classification | VQA | 110 | Multiple Choice/Open-ended |
| | Phenomenon Detection | Perception | Eddy identification | VQA | 204 | Multiple Choice/Open-ended |
| | | | Marine fog detection | VQA | 200 | Multiple Choice/Open-ended |
| | | | Eddy identificaEddy Localizationtion | Visual Grounding | 166 | Multiple Choice/Open-ended |

## A.3 DATA PROCESSING STRATEGIES FOR THE SIX SPHERES.

A key point of inquiry is how Multimodal Large Models (MLLMs) process or adapt to the full range of input modalities provided in our dataset. This section elaborates on the data processing strategies employed for the 33 diverse sources integrated into OmniEarth-Bench.

Our general workflow involves domain experts collaborating with annotators to manually transform raw inputs into formats suitable for MLLMs, such as RGB images paired with questions. The specific approach depends on the nature of the original data:

- **When RGB images are available:** Experts select scientifically relevant images for Visual Question Answering (VQA) tasks and design corresponding questions, with annotators assisting in formulating the responses and distractors.
- **When only raw data is provided:** Experts visualize key variables as RGB images based on predefined dimensions and scientific conventions. Some detailed examples using meteorological data are included below.

The complete processing code and pipeline documentation will be released on our project website in a future update for community use. The following provides a detailed breakdown for several of the Earth's spheres.

**Cryosphere** To create a test-ready QA benchmark for Level-4 sea-ice analyticscovering Sea-Ice Extent (SIE) estimation, Sea-Ice Volume (SIV) trend prediction, SIV-from-SIC estimation, Sea-Ice Thickness (SIT) trend prediction, and SIT-from-SIC estimationwe (1) acquire daily sea-ice concentration fields from NSIDCs G02202-v4 archive and Arctic SIV time-series from PIOMAS, (2) decode the NetCDF products with the open-source `netCDF4` package, (3) derive daily SIE and basin-averaged SIC, (4) plot SIE, SIC, and SIV evolutions as authoritative references, and (5) assemble questionanswer pairs that juxtapose data-driven ground-truth with intentionally misleading distractors. Each question is purpose-built to probe a specific facet of sea-ice reasoning in multimodal large-language models.

**Biosphere** For multi-scenario ecological Q&A tasks, we built a high-resolution dataset combining remote sensing imagery, eco-meteorological data, and multi-source annotations. It includes MODIS 7-band, Landsat, and UAV images; COCO-format dead oil palm labels; multi-band oil spill masks; surface vegetation and human footprint indices; daily meteorological sequences (temperature, humidity, radiation, wind, precipitation); 19 SatBird bioclimatic variables; carbon flux (NEE) observations; and structured metadata (CSV/JSON with means, species probabilities, and patch IDs). Preprocessing involved three main stages:

1. **Metric Computation and Normalization**
   - **GLASS:** Mean FVC $\times$ 0.004 and LAI $\times$ 0.1 values written to CSV.
   - **HFP:** Aggregated raster means.
   - **CarbonSense:** Removed missing rows, converted meteorological variables to text, and aligned them with MODIS images.
   - **SatBird:** Stratified by species richness and extracted species probabilities from hotspot JSON.
   - **ROSID:** Binarized masks, counted connected domains, and calculated oil spill area from pixel count.
   - **MOPAD:** Filtered `category_id == 1` to count dead oil palms.
2. **Image Standardization**
   - Converted SatBird `.tif` to `.jpg`, CarbonSense `.pkl` to `.png`.
   - Linked GLASS/HFP patches to full 7-band MODIS imagery.
   - Preserved original resolution for UAV and Landsat images.
3. **Thresholding and Answer Binning**
   - Mapped continuous variables (FVC, LAI, HFP) to preset intervals.
   - Generated distractors using $\pm 2040\%$ (ROSID) or $\pm 210$ trees (MOPAD) around ground truth.

- Binned species occurrence probabilities into [0, 0.3], [0.3, 0.6], and [0.6, 1].
- Converted oil spill pixel count $\times$ 900 m$^2$ to km$^2$ and rounded.

**Atmosphere**  For atmospheric weather events, we primarily use reanalysis datasets such as ERA5, with meteorological variables visualized as RGB images that include variable names, legends, latitude-longitude grids, and national boundaries.

1. **Short- and Medium-Term Events:** We compiled global event records from 19792020, retrieved corresponding timestamps from ERA5, and used 1-hour and 6-hour intervals for short- and medium-term events, respectively. For multivariable or long-duration cases, the interval was increased fourfold to limit image volume.

2. **Seasonal and Interannual Events:** These longer-range events require post-processing of raw model data. We use NOAAs Global Reports (2010.012025.03) featuring visualized anomalies and regional summaries. Following LLaVA's methodology, we use ChatGPT-4o to generate QA pairs. Seasonal inputs include all 12 monthly reports from a given year; interannual tasks use annual reports from 20102024.

**Lithosphere**  For seismic waveform analysis, we leverage the large-scale Stanford Earthquake Dataset (STEAD). The process involves several key stages: (1) we select high-quality, three-component seismic waveforms and filter out samples with missing components or low signal-to-noise ratios; (2) all waveforms are standardized to a uniform 100 Hz sampling rate, normalized, and formatted into 60-second slices; and (3) these raw waveforms are converted into standardized RGB image representations for visual analysis. Based on these images, we construct VQA tasks covering five core seismic challenges: earthquake/noise classification, P-wave and S-wave arrival picking, epicentral distance inference, and magnitude estimation. For each task, the ground-truth answer (True Value) is derived from STEADs expert-annotated labels, while distractor answers (False Values) are systematically generated by applying controlled perturbations to the true values to test model robustness.

**Oceansphere**  For the oceansphere, our data processing varies based on the source format. For analyzing climate phenomena like the El Niño-Southern Oscillation (ENSO), we use the ERSSTv5 Sea Surface Temperature (SST) dataset. The VQA construction involves: (1) calculating 30-year centered climatology and 3-month averaged anomalies; (2) computing the Niño3.4 index to classify ENSO events; (3) visualizing the SST anomalies as RGB maps; and (4) selecting relevant maps to construct question-answer pairs. This supports two primary tasks: identifying the current ENSO state using Pacific anomaly maps from the peak DJF season, and forecasting future ENSO events using a sequence of six consecutive global anomaly maps as input. For datasets that are already in visible light RGB format, such as the MADOS dataset for marine pollution monitoring from Sentinel-2 imagery, the process is more direct. It primarily involves selecting scientifically relevant images and constructing VQA tasks around visually identifiable features, such as classifying pollution types (e.g., oil spills, marine debris, floating algae) and visually localizing targets like oil platforms or specific patches of contamination.

**Human-activity sphere**  For the human-activity sphere, the data is predominantly in visible light RGB format, simplifying the preprocessing pipeline. We utilize a range of specialized datasets such as xView2, UBCv1, and WHU-OHS to construct VQA tasks focused on disaster impact assessment and fine-grained urban and land-use analysis. These tasks include identifying disaster types (e.g., floods, wildfires), counting damaged or destroyed buildings by comparing pre- and post-event imagery, assessing the damage level of individual structures, and performing fine-grained urban analysis, such as object counting (vehicles, buildings), land-use classification (farmland, urban built-up), and visual localization of specific man-made features.

## A.4 Construction Details of Different Question Formats.

### A.4.1 Chain-of-Thought (CoT) Annotation Details.

This section provides a detailed account of the motivation, scope, and rigorous workflow used to construct the Chain-of-Thought (CoT) annotations for the Level-4 (L4) evaluation tasks in OmniEarth-Bench.

**Motivation and Scope** During the developmental phase of our benchmark, we identified a subset of tasks that were exceptionally challenging for current Multimodal Large Models (MLLMs). These tasks, such as the visual localization of a single damaged building across pre- and post-disaster imagery, could not be solved through single-step perception or retrieval. They inherently require a complex, multi-step reasoning process that involves identifying context, comparing features, eliminating distractors, and finally pinpointing the target.

Therefore, the primary motivation for introducing CoT annotations was to create a mechanism to explicitly evaluate the capacity of MLLMs to perform and follow logical, multi-step scientific reasoning. These annotations are designed as a gold standard for evaluation, not for model training.

This annotation effort was strategically focused on 6 of our most demanding sub-tasks, where such complex reasoning is indispensable. In total, this resulted in a collection of 610 samples meticulously annotated with detailed reasoning chains.

**Annotation Paradigm and Workflow** To ensure the quality, objectivity, and logical consistency of our CoT annotations, we designed and implemented a rigorous, multi-stage, human-in-the-loop workflow. This process was structured to leverage the capabilities of advanced AI while relying on human expertise for creation and final validation. The workflow is detailed below:

1. **AI-Generated Reference Chain.** For each question, a high-quality reference reasoning chain was first generated using the GPT-4o model. Crucially, to ensure the reference was accurate and complete, the model was provided with both the question and the ground-truth answer. This initial chain served as a comprehensive, machine-generated example of a possible logical pathway.

2. **Human-Authored Key Steps.** Certified annotators, all holding at least a bachelor's degree, were then tasked with creating a new and independent set of crucial reasoning steps based on their own analysis of the problem, using the AI-generated chain only as a reference.

3. **Ensuring Logical Independence and Diversity.** To prevent mere paraphrasing of the AI's output and to encourage diverse, human-centric logic, a strict quality constraint was enforced: the semantic and logical overlap between the final human-authored steps and the initial AI-generated reference chain had to be less than 20%. This ensured that our CoT annotations represent genuine human reasoning patterns.

4. **Domain Expert Verification.** The newly created CoT annotations were subsequently passed to our team of domain experts (Ph.D. students in relevant Earth science fields) for a final round of verification. They meticulously checked each reasoning chain for:

   • *Logical Soundness:* Ensuring the steps flow logically from one to the next.

   • *Scientific Accuracy:* Validating any domain-specific knowledge or claims.

   • *Completeness:* Confirming that all necessary steps to reach the correct conclusion are present.

5. **Refinement and Condensation.** Lastly, all expert-validated steps were refined to be as concise as possible. This involved removing redundant phrases and retaining only the core logic and essential visual or data-driven cues. The goal was to produce an information-dense yet easy-to-follow reasoning pathway that represents the most efficient and accurate thought process to solve the task.

This structured paradigm guarantees that our CoT annotations are not only correct and logical but also a robust and reliable tool for evaluating the sophisticated reasoning capabilities of advanced AI models in the Earth sciences.

### A.4.2 OPEN-ENDED QUESTIONS ANNOTATION DETAILS.

**Motivation and Scope** To more rigorously assess the true generative and reasoning capabilities of Multimodal Large Language Models (MLLMs), we introduced an open-ended question format for the OmniEarth-Bench. The standard Multiple-Choice Question (MCQ) format, while efficient for broad-scale evaluation, provides explicit options that can act as "scaffolds" or hints for the model. This mitigates the risk of models guessing the correct answer from a limited set of choices without genuine understanding. The open-ended format removes these scaffolds, compelling the model to generate answers from scratch and providing a more challenging and realistic measure of its knowledge and reasoning abilities in the Earth science domain.

The open-ended questions in OmniEarth-Bench are systematically transformed from the existing L4 subtasks originally designed in the MCQ format. This transformation process is not uniform but is instead tailored to the intrinsic nature of the answer for each task. We categorized all L4 subtasks into four distinct types to guide this conversion:

1. **Classification with a small, fixed label set:** Tasks where the answer is one of a few predefined categories (e.g., ENSO phases).

2. **Classification with a large label set:** Tasks where the answer is a choice from a larger, but still finite, set of options (e.g., direction of cyclone movement).

3. **Exact regression:** Tasks that require a precise numerical answer (e.g., counting objects).

4. **Interval regression:** Tasks where the answer is a numerical or temporal range (e.g., duration of an event).

**Workflow** The construction and validation of open-ended question-answer pairs followed a meticulous, expert-driven workflow:

1. **Task Classification:** Domain experts first categorized each of the 103 L4 subtasks into one of the four types defined in the Scope. This initial step determined the specific transformation strategy to be applied.

2. **Prompt Transformation:** Based on its category, the prompt for each task was re-engineered:
   - For tasks with **small, fixed label sets**, the multiple-choice options were removed, and the question was rephrased to directly ask for the answer (e.g., "What is the ENSO phase shown in the image?").
   - For tasks with **large label sets**, the candidate labels were explicitly listed within the prompt itself, instructing the model to choose and generate the answer from the provided list (e.g., "From the following list [North, Northeast, East...], what is the direction of movement...?").
   - For **exact regression** tasks, the question was rephrased to directly ask for the numerical value (e.g., "How many deceased oil palm trees are in the image?").
   - For **interval regression** tasks, the prompt was modified to specify the required answer format (e.g., "Provide the time range in the format 'Start Month - End Month'.").

3. **Golden Answer Formulation:** The correct option from the original MCQ (e.g., option 'C') was converted into its full-text or numerical "golden answer" (e.g., the text "Strong El Niño" or the number "15").

4. **Expert Review and Refinement:** Finally, all newly generated open-ended questions and their corresponding golden answers were rigorously reviewed by the domain experts. This step ensured that the rephrased questions were unambiguous, scientifically accurate, and that the golden answers remained correct and appropriately formatted.

**Evaluation Method** The evaluation of open-ended questions poses a unique challenge, as traditional exact string matching is often too rigid to account for semantically correct but varied natural language responses. Therefore, we employ a **Large Language Model (LLM) as an automated judge** for assessing the correctness of generated answers. Specifically, a powerful, state-of-the-art LLM (e.g., GPT-4o) is provided with the original question, the MLLM's generated answer, and the expert-defined golden answer. The LLM's task is to determine whether the MLLM's answer is semantically equivalent to or sufficiently captures the essence of the golden answer, thereby marking

it as correct or incorrect. This approach leverages the advanced semantic understanding capabilities of LLMs to provide a more nuanced and fair assessment of open-ended generative performance, overcoming the limitations of keyword-based or fixed-choice evaluations.

### A.4.3 IMAGES CAPTION ANNOTATION DETAILS.

**Motivation and Scope** Standard image captioning benchmarks typically focus on describing the literal content of an image, which is insufficient for the complex analytical needs of Earth science. In this domain, a deep understanding requires not only recognizing visual patterns in satellite imagery but also integrating them with external, contextual scientific data. To address this gap, we designed a specialized captioning task for OmniEarth-Bench. The motivation is twofold: first, to create a benchmark that evaluates an MLLM's ability to perform **multimodal fusion**, synthesizing information from both visual satellite data and textual historical disaster data; second, to test a model's capacity for domain-specific reasoning by requiring it to act as a meteorological expert, thereby moving beyond simple object description to nuanced scientific interpretation.

The scope of this task is centered on significant climate-related disasters, leveraging a combination of satellite and historical data sources.

1. **Image Data Source:** The visual data consists of multispectral images from the **Microwave Humidity Sounder (MHS)** satellite. Each event is represented by a set of images corresponding to MHS's five distinct spectral bands, capturing 12 hours of observational data.

2. **Contextual Data Source:** The textual data is sourced from the **EM-DAT International Disaster Database**, which provides detailed historical records of climate events.

3. **Event Types:** The task covers a range of severe weather and climate phenomena, including **storms, floods, landslides, wildfires, cold waves, and heat waves**.

4. **Output:** The deliverable is a collection of image sets paired with single-paragraph, detailed captions. Each caption cohesively integrates the visual evidence from the MHS imagery with the factual context from the corresponding EM-DAT record.

**Workflow** The creation of the image-caption pairs followed a systematic, multi-stage workflow, ensuring both scientific rigor and data quality.

1. **Raw Data Acquisition and Preprocessing:**
   - Level 1B data from the MHS satellite, which is initially sparse and irregular, was collected.
   - This raw data was processed using a **remapping algorithm** to project the sparse observation points onto a regular, spatially coherent global grid, creating a dense image-like representation for each spectral channel.

2. **Event Identification and Cropping:**
   - Using the EM-DAT database, significant historical climate disasters were identified, and their specific geographic locations and timelines were extracted.
   - An **event cropping algorithm** was then applied to the global satellite data grids. For each identified disaster, the corresponding 12-hour, 5-channel image sequence was precisely cropped from the global map based on the event's location and time.

3. **Automated Caption Generation with Multimodal Input:**
   - The cropped set of multispectral satellite images and the associated textual disaster report from EM-DAT were provided as combined input to a state-of-the-art LLM (GPT-4o).
   - A specialized prompt was engineered to instruct the model to act as a "meteorological analysis expert." The prompt explicitly required the model to generate a single, cohesive paragraph that analyzes and describes the event by synthesizing relevant details from *both* the images and the historical data.

4. **Expert Review and Validation:**
   - Finally, every automatically generated caption underwent a rigorous review by human domain experts. This validation step was crucial for verifying the scientific accuracy

of the interpretation, the factual correctness of the fused information, and the overall linguistic quality and coherence of the caption. Any captions that were inaccurate or failed to properly integrate the data sources were either refined by experts or discarded to maintain the high standard of the benchmark.

## A.5 DETAILED RESULTS OF SPECIFIC SUB-TASKS (L-4 DIMENSION).

**Worse performance of Qwen2.5-VL.** Qwen2.5-VL lagged behind contemporary open-source models like InterVL3 on Earth-related tasks, with both its 7B and 72B versions rarely ranking first in any L4 subtask. Despite its larger parameter size, the 72B model often scored zero on multiple tasks. However, this low accuracy shouldn't be seen as a lack of capability.Qwen2.5-VL often responds with E (unable to answer) when lacking domain knowledgean honest approach. In contrast, some models tend to guess when uncertain, which is less desirable.

**Many models scored zero on various sub-tasks.** Even the top-performing InternVL3-78B failed on several. GPT-4o, a widely used closed-source model, recorded zero accuracy on nearly half the tasks. These results underscore the effectiveness and domain specificity of our sub-task design.

**No significant gap between closed-source and open-source models.** Although Gemini and Claude3 slightly lag behind LLaVA-OneVision and InternVL3 on sub-tasks, the gap is minimal. This indicates that open-source multimodal models are still well-suited for advancing Earth science research and agent development, without relying exclusively on closed-source alternatives.

**Poor performance of some subtasks in Cross-sphere.** In the cross-sphere species richness prediction task, no model surpassed 10% accuracy on 900 test samples, which aligns with expectations given the task's complexity. Integrating climate variables, satellite imagery, and vegetation factors creates a highly intricate prediction challenge beyond the capabilities of current models.

Table 10: **CoT performance of each L4 subtasks.** We mark the highest score of each metric in red , and second highest underlined.

| Task | Num. | Qwen2.5-VL-7B | | | LLaVA-OneVision-7b | | | InternVL3-8B | | | InternVL3-78B | | |
|---|---|---|---|---|---|---|---|---|---|---|---|---|---|
| | | Percision | Recall | F1 | Percision | Recall | F1 | Percision | Recall | F1 | Percision | Recall | F1 |
| Multi-image Visual Localization | 102 | 95.87 | 31.21 | 47.09 | 93.28 | 40.36 | 56.34 | 93.89 | 41.34 | 57.40 | 97.49 | 43.74 | 60.39 |
| Visual grounding of damaged individual buildings | 102 | 95.97 | 25.98 | 40.89 | 92.38 | 22.92 | 36.73 | 95.86 | 33.47 | 49.62 | 91.99 | 42.58 | 58.21 |
| Overall building height estimation | 100 | 83.97 | 14.50 | 24.73 | 83.73 | 18.17 | 29.86 | 93.82 | 20.00 | 32.97 | 92.59 | 19.50 | 32.22 |
| Spatial relationships under complex conditions | 99 | 94.51 | 35.98 | 52.12 | 91.75 | 22.27 | 35.84 | 93.79 | 36.60 | 52.65 | 96.25 | 32.55 | 48.65 |
| Individual building damage assessment | 107 | 93.21 | 37.54 | 53.52 | 87.93 | 12.96 | 22.59 | 92.79 | 40.74 | 56.62 | 95.58 | 39.06 | 55.46 |
| Individual building height estimation | 101 | 92.49 | 12.71 | 22.34 | 87.69 | 13.2 | 22.95 | 91.77 | 13.37 | 23.34 | 90.46 | 14.36 | 24.78 |

**InternVL3 outperforms other MLLMs in CoT tasks.** The InterVL3 series performed strongly on CoT tasks, with both the 7B and 78B models achieving top results across all subtasks, showcasing their strength in geoscience chain-of-thought reasoning. Future geoscience reasoning tasks could benefit from further training and application of this series.

Table 11: **Visual Grounding performance one each L4 subtasks.**

| L4 | Num | Qwen2.5-VL-7B | | LLaVA-OneVision-7b | | InternVL3-8B | | InternVL3-78B | | GPT-4o | | Gemini-2.0 | | Claude-3-7 | |
|---|---|---|---|---|---|---|---|---|---|---|---|---|---|---|---|
| | | acc@0.5 | acc@0.7 | acc@0.5 | acc@0.7 | acc@0.5 | acc@0.7 | acc@0.5 | acc@0.7 | acc@0.5 | acc@0.7 | acc@0.5 | acc@0.7 | acc@0.5 | acc@0.7 |
| Salt Body Location | 302 | 0 | 0 | 5.3 | 0.33 | 8.94 | 1.66 | 4.3 | 0.33 | 0.08 | 0 | 0.13 | 0.04 | 0.02 | 0 |
| Eddy Localization | 166 | 3.01 | 0 | 1.81 | 0.6 | 6.63 | 0.6 | 13.86 | 3.61 | 0.12 | 0.01 | 0.34 | 0.06 | 0.2 | 0.07 |
| Visual Grounding of Land Types | 508 | 0.2 | 0.00 | 0.59 | 0.20 | 2.56 | 0.59 | 2.36 | 0.20 | 0.02 | 0.00 | 0.03 | 0.00 | 0.00 | 0.00 |

**Visual grounding performance is notably poor across all models** It exposes two main shortcomings: limited geoscientific knowledge and weak visual localization capabilities. Both open- and closed-source models fall short in these aspects.

Table 12: **VQA Performance in L4 dimension.** We mark the highest score of each metric in red , and second highest underlined.

| Domain | L4-task | Num | Qwen2.5-VL 7B | Qwen2.5-VL 72B | InternVL3 8B | InternVL3 78B | LLaVA-OV 7B | GPT-4o | Gemini | Claude3 |
|---|---|---|---|---|---|---|---|---|---|---|
| Human-activity | Change detection... | 502 | 6.97 | 1.39 | 43.03 | 72.51 | 85.06 | 0 | 74.1 | 10.36 |
| | Partially damaged... | 498 | 1.41 | 0 | 4.02 | 45.98 | 50.6 | 0 | 11.24 | 0.8 |
| | Building damage... | 499 | 0.2 | 4.81 | 5.81 | 7.62 | 33.87 | 0 | 8.22 | 4 |
| | Disaster prediction | 500 | 0.8 | 5.6 | 4 | 0.6 | 0.6 | 0 | 0.2 | 2.8 |
| | Disaster type... | 500 | 1 | 6.2 | 6.6 | 8.2 | 1.6 | 0.4 | 4.2 | 11.2 |
| | Geolocation... | 502 | 12.15 | 9.76 | 36.25 | 44.82 | 31.67 | 2.59 | 36.45 | 33.47 |
| | Fine-grained object... | 514 | 10.89 | 19.46 | 21.98 | 30.35 | 48.05 | 0 | 47.67 | 16 |
| | Overall counting | 502 | 1.99 | 2.59 | 18.13 | 17.13 | 21.12 | 0.2 | 31.67 | 0 |
| | Counting complex... | 754 | 0.4 | 0 | 31.75 | 19.05 | 18.25 | 0.27 | 9.28 | 0.93 |
| | Overall land... | 509 | 0 | 0.2 | 1.96 | 1.38 | 1.38 | 0 | 1.57 | 1.18 |
| | Fine-grained land... | 509 | 3.93 | 6.68 | 26.13 | 6.48 | 39.88 | 39.29 | 30.26 | 32 |
| | Visual localization... | 509 | 1.38 | 3.54 | 5.11 | 4.52 | 1.77 | 0.39 | 6.68 | 1.57 |
| | Land types ... | 449 | 3.12 | 3.34 | 24.5 | 16.04 | 14.7 | 0 | 5.35 | 0 |
| | Individual building ... | 107 | 0.93 | 7.48 | 28.97 | 24.3 | 47.66 | 0 | 35.51 | 14.02 |
| | Multi individual ... | 102 | 10.78 | 13.73 | 28.43 | 53.92 | 67.65 | 8.82 | 41.81 | 43.14 |
| | Spatial relation ... | 99 | 22.22 | 21.21 | 33.33 | 36.36 | 23.23 | 2.02 | 39.39 | 16.16 |
| | Visual grounding ... | 102 | 35.29 | 37.25 | 52.94 | 52.94 | 65.69 | 28.43 | 56.86 | 37.25 |
| | Overall height ... | 509 | 0 | 0.2 | 1.96 | 1.38 | 1.38 | 0 | 1.57 | 1.18 |
| | Individual height ... | 101 | 0 | 0 | 45.54 | 0 | 38.61 | 0 | 0 | 0 |
| Biosphere | Dead oil .. counting | 828 | 52.9 | 47.95 | 48.67 | 73.67 | 73.67 | 0 | 56.04 | 60.51 |
| | oil ..identification | 828 | 85.51 | 70.65 | 81.52 | 83.33 | 85.63 | 0 | 51.81 | 77.29 |
| | oil spill area ... | 123 | 0 | 0 | 5.69 | 5.69 | 0 | 0 | 0 | 0 |
| | oil spill counting ... | 123 | 0 | 0.81 | 41.46 | 53.66 | 22.76 | 0 | 7.32 | 0 |
| | footprint assess .. | 300 | 0.33 | 0.67 | 36 | 26.33 | 22.67 | 0.67 | 7.67 | 26.67 |
| | footprint index .. | 300 | 0 | 0 | 3.33 | 3.33 | 23.33 | 0 | 0 | 20 |
| | species prediction .. | 500 | 5 | 15.2 | 35.6 | 46.4 | 46.8 | 0.8 | 15.8 | 13.2 |
| | species proportion.. | 500 | 0 | 0.2 | 26.2 | 1.8 | 18.4 | 0 | 5.2 | 2.4 |
| | Animal classification | 108 | 4.63 | 1.85 | 15.74 | 27.78 | 43.52 | 0 | 9.26 | 2.78 |
| | Geographical ... | 500 | 4.6 | 13.4 | 18.2 | 26 | 50 | 10.4 | 38.6 | 75.8 |
| | Species distribution... | 1000 | 2.1 | 68.6 | 73.3 | 78.1 | 40.1 | 16.8 | 91.5 | 90.2 |
| | Fractional ... | 300 | 0 | 0 | 13 | 25 | 56 | 0 | 3.67 | 46 |
| | Leaf area index... | 300 | 0 | 0 | 7.33 | 14 | 50 | 0 | 0 | 29.33 |
| | animal counting... | 110 | 0 | 3.64 | 0 | 6.36 | 2.73 | 0.91 | 7.27 | 0 |
| | Peak vegetation... | 300 | 9 | 0.67 | 25 | 8.33 | 24 | 0 | 18.3 | 24 |
| Cross-sphere | Most likely species... | 900 | 0.11 | 18.89 | 20.89 | 40.67 | 21.89 | 0.22 | 36.67 | 59.22 |
| | Species occurrence... | 453 | 2.65 | 4.64 | 58.5 | 13.69 | 8.17 | 0 | 3.31 | 29.8 |
| | Species richness ... | 900 | 0 | 0 | 9 | 0.33 | 7.67 | 0 | 0 | 0 |
| | Carbon flux .. | 330 | 11.52 | 0 | 24.85 | 0.61 | 25.45 | 0 | 0.61 | 0 |
| | Flood detecting | 596 | 44.8 | 0 | 52.18 | 51.51 | 52.35 | 0 | 28.52 | 51 |
| | Flood predicting | 277 | 0 | 0 | 91.7 | 8.3 | 0 | 0 | 32.49 | 44.04 |
| Cryosphere | Glacial Lake .. | 12 | 25 | 25 | 75 | 75 | 75 | 50 | 66.67 | 66.67 |
| | Glacier Melting... | 10 | 30 | 10 | 40 | 50 | 30 | 10 | 30 | 40 |
| | Slide Recognition... | 8 | 65.5 | 0 | 37.5 | 62.5 | 62.5 | 12.5 | 62.5 | 50 |
| | SIC Estimate SIT | 20 | 0 | 0 | 55 | 100 | 45 | 85 | 80 | 90 |
| | SIC Estimate SIV | 20 | 0 | 0 | 45 | 80 | 40 | 50 | 70 | 80 |
| | SIT Trend Prediction | 30 | 3.33 | 0 | 50 | 90 | 50 | 40 | 46.7 | 60 |
| | SIV Trend Prediction | 30 | 0 | 0 | 36.67 | 80 | 36.67 | 13.33 | 53.33 | 20 |
| | Sea Ice Extent... | 100 | 19 | 12 | 55 | 59 | 32 | 39 | 59 | 29 |
| Lithosphere | P-wave phase picking | 300 | 8 | 11.33 | 10.67 | 16 | 8 | 6 | 22.33 | 11 |
| | S-wave phase picking | 300 | 36.67 | 35.33 | 61.67 | 41 | 32 | 16.67 | 49.33 | 28.33 |
| | Earthquake or noise .. | 300 | 44.67 | 86.33 | 63 | 59.33 | 52.33 | 50 | 93.33 | 95 |
| | magnitude estim... | 300 | 1.33 | 0 | 38.33 | 0 | 32.67 | 0 | 26.67 | 1.67 |
| | source-receiver ... | 300 | 20.33 | 0.67 | 35.33 | 6.67 | 33 | 1.67 | 14 | 21.67 |
| | Salt body detection | 329 | 0.91 | 0.93 | 15.81 | 17.33 | 14.29 | 2.43 | 27.96 | 11.25 |

Table 13: **VQA Performance in L4 dimension.** We mark the highest score of each metric in `red`, and second highest underlined.

| Sphere | L4-task | Num | Qwen2.5-VL | | InternVL3 | | LLaVA-OV | GPT-4o | Gemini | Claude3 |
|---|---|---|---|---|---|---|---|---|---|---|
| | | | 7B | 72B | 8B | 78B | 7B | | | |
| | Cyclone move... | 185 | 8.65 | 2.16 | 37.84 | 47.03 | 34.59 | 6.49 | 38.38 | 44.32 |
| | Cyclone phase... | 90 | 0 | 0 | 48.89 | 0 | 25.56 | 1.11 | 25.56 | 5.75 |
| | Event intensity... | 594 | 17.34 | 44.61 | 38.72 | 0 | 30.14 | 13.21 | 53.62 | 33.28 |
| | Event localization | 93 | 18.28 | 5.38 | 46.24 | 41.94 | 33.33 | 18.98 | 51.82 | 43.07 |
| | Event onset... | 42 | 0 | 0 | 26.19 | 0 | 35.71 | 16.67 | 30.95 | 38.1 |
| | Event trend... | 575 | 0.18 | 0 | 48.52 | 0 | 36.87 | 2.96 | 39.82 | 14.7 |
| | Geopotential... | 33 | 18.18 | 12.12 | 18.18 | 15.15 | 12.12 | 18.18 | 9.09 | 30.3 |
| | Moisture flux... | 150 | 0 | 0 | 66.00 | 0 | 53.34 | 0 | 0 | 6.12 |
| | System... | 231 | 4.33 | 17.75 | 33.77 | 0 | 22.94 | 18.18 | 40.69 | 52 |
| | System evolution... | 91 | 2.2 | 0 | 40.66 | 0 | 56.04 | 17.58 | 57.14 | 70 |
| | Event intensity... | 323 | 18.89 | 13.62 | 50.77 | 0 | 34.37 | 30.64 | 59.54 | 18.31 |
| | Event localization | 133 | 1.5 | 0.75 | 47.37 | 0 | 13.53 | 25.85 | 50.68 | 21.05 |
| | Event trend... | 297 | 3.03 | 0 | 31.65 | 0 | 35.69 | 4.71 | 41.08 | 19.57 |
| | Event type... | 139 | 23.74 | 17.27 | 36.69 | 0 | 25.9 | 23.74 | 27.34 | 0 |
| | Dynamic feature... | 40 | 45 | 67.5 | 37.5 | 0 | 2.5 | 15 | 27.5 | 30.77 |
| | Event evolution... | 90 | 0 | 0 | 24.44 | 0 | 2.22 | 0 | 21.11 | 8 |
| Atmosphere | Thermodynamic... | 40 | 0 | 0 | 15 | 0 | 0 | 7.5 | 20 | 15.79 |
| | Pressure... | 847 | 0 | 0 | 0.83 | 0 | 30.34 | 0 | 0 | 0 |
| | Radius (gale)... | 847 | 0 | 0 | 21.49 | 0.59 | 24.91 | 0 | 0 | 35.42 |
| | Radius (storm)... | 847 | 0 | 0 | 23.02 | 0.71 | 14.76 | 0 | 0 | 47.23 |
| | minor gale ... | 847 | 0 | 0 | 0 | 5.79 | 15.47 | 0 | 0 | 0 |
| | minor storm ... | 847 | 0 | 0 | 1.89 | 0 | 4.84 | 0 | 0 | 0 |
| | Wind estimation | 847 | 0 | 0 | 0 | 5.79 | 30.46 | 0 | 0 | 0 |
| | Precipitation ... | 75 | 0 | 1.33 | 21.33 | 28 | 22.67 | 6.67 | 28 | 16 |
| | Seasonal ... | 101 | 8.91 | 9.9 | 46.53 | 0 | 45.54 | 27.52 | 40.5 | 40 |
| | Temperature ... | 75 | 16 | 18.67 | 45.33 | 57.33 | 48 | 29.33 | 49.33 | 48 |
| | ENSO feature... | 86 | 41.86 | 63.95 | 75.58 | 0 | 77.91 | 90.7 | 75.58 | 73.26 |
| | Long.. Precipitation | 50 | 0 | 0 | 34 | 48 | 26 | 20 | 26 | 32 |
| | Long.. Temperature | 51 | 0 | 0 | 49.02 | 60.78 | 27.45 | 39.22 | 25.49 | 27.45 |
| | Event type .. | 300 | 15 | 0 | 36 | 19 | 42.67 | 20.67 | 1 | 16 |
| | Miss alarm .. | 300 | 0 | 0 | 19 | 22 | 32 | 0 | 0.67 | 8.33 |
| | Movement .. | 200 | 45 | 21 | 76.5 | 60.5 | 81 | 2 | 69 | 64.5 |
| | Rotate center... | 93 | 3.23 | 2.15 | 62.37 | 75.27 | 46.24 | 0 | 6.45 | 58.06 |
| | ENSO .. | 146 | 21.92 | 34.93 | 33.56 | 17.81 | 23.97 | 31.51 | 26.03 | 51.37 |
| | IOD Identification | 140 | 4.29 | 19.29 | 12.86 | 4.29 | 12.14 | 50 | 5.71 | 12.86 |
| | ENSO Forecast | 152 | 0 | 3.29 | 23.03 | 36.18 | 15.79 | 6.58 | 23.03 | 19.74 |
| Oceansphere | IOD Forecast | 145 | 0 | 0 | 4.83 | 18.62 | 18.62 | 0 | 0.69 | 0 |
| | Eddy Identification | 204 | 3.93 | 0.49 | 3.92 | 4.9 | 44.1 | 1.47 | 4.9 | 14.22 |
| | Marine Fog .. | 200 | 67.5 | 55 | 61 | 57 | 53.5 | 3 | 55.5 | 55.5 |
| | Marine Pollution... | 110 | 0 | 0.91 | 2.73 | 2.73 | 3.64 | 0.91 | 2.73 | 8.18 |

## A.6 DETAILED RESULTS OF SPECIFIC SUB-TASKS (L-3 CAPABILITY).

This section highlights the performance of MLLMs across all L-3 capabilities. The VQA task is split into perception, General reasoning, and Scientific-Knowledge Reasoning dimensions, with results shown in Tables 14. Model performance varies across L3 dimensions: InterVL3 excels in perception,

Table 14: **VQA Performance in L3 dimension.** We mark the highest score of each metric in red, and second highest underlined.

| L4-task | Qwen2.5-VL | | InternVL3 | | LLaVA-OV | GPT-4o | Gemini | Claude3 |
|---------|------------|------|-----------|------|----------|--------|--------|---------|
| | 7B | 72B | 8B | 78B | 7B | | | |
| Perception | 13.42 | 15.37 | 35.77 | 24.68 | 34.66 | 15.40 | 33.33 | 26.97 |
| General Reasoning | 7.32 | 10.86 | 28.67 | 27.48 | 30.71 | 3.62 | 21.22 | 13.74 |
| Scientific-Knowledge Reasoning | 8.2 | 5.72 | 30.89 | 27.49 | 30.18 | 8.74 | 23.66 | 31.62 |

LLaVA-OneVision in general reasoning, and Claude3 in scientific knowledge reasoning. Overall, InterVL3 shows consistently strong results, while Qwen2.5VL and GPT-4o fall notably behind.

## A.7 SAMPLES AND CHALLENGING CASES OF OMNIEARTH-BENCH

In this section, we construct a detailed table (Tab. 15) analyzing model performance and error causes for the L-4 subtask. We then use examples to thoroughly illustrate the errors for typical subtasks. In this section, we present a case study analysis of the error types made by Gemini-2.0-Flash Team et al. (2023), Qwen2.5-VL Bai et al. (2025), and InterVL3 Zhu et al. (2025) on various sub-tasks in OmniEarth-Bench. We classify the errors into the following 6 categories:

**Spatio-temporal Frame Confusion** : The model misinterprets either the time sequence or the spatial orientation / coordinate frame of the data, leading to a reversed trend, direction, or geographic reference. See examples in Fig. 7, Fig. 13, etc.

**Threshold / Severity Mis-estimation** : Numeric values are read incorrectly or wrong thresholds are applied, so strength or severity categories are wrong. See examples in Fig. 12, Fig. 15, etc.

**Image-feature Misinterpretation** : Visual cues (texture, color, shape) are misread; key features are missed or artefacts are mistaken for real features. See examples in Fig. 16, Fig. 17, etc.

**Domain-knowledge / Semantic Mis-match** : Mis-application of non-visual expertise  ecology, climate thresholds, hazard mechanics  so the scene is matched to an incorrect knowledge template. See examples in Fig. 9.

**Over-cautious / Refusal** : Adequate information is available, but the model answers Unable to decide (or hedges) to avoid committing. See examples in Fig. 8, Fig. 10, etc.

**Target Mis-location** : The object or area specified in the prompt is not correctly identified, so all subsequent reasoning is off target. See examples in Fig. 14.

Table 15: Table index of case study figures by sub-tasks (L-3 capability) with associated (error) categories for each MLLM.

| Case | L-1 task | L-4 task | Gemini | Qwen2-VL | Internvl3 |
|------|----------|----------|--------|----------|-----------|
| Fig. 7 | Atmosphere | Movement Prediction | Spatio-temporal Frame Confusion | Spatio-temporal Frame Confusion | Correct |
| Fig. 8 | Biosphere | Dead Oil Palm counting | Correct | Over-cautious / Refusal | Over-cautious / Refusal |
| Fig. 9 | Biosphere | Species Distribution Prediction | Correct | Domain-knowledge / Semantic Mis-match | Correct |
| Fig. 10 | Cross-sphere | Most likely species to occur | Correct | Over-cautious / Refusal / Domain-knowledge / Semantic Mis-match | Over-cautious / Refusal |
| Fig. 11 | Cross-sphere | Global Flood Forecasting | Correct | Image-feature Misinterpretation / Threshold / Severity Mis-estimation | Image-feature Misinterpretation / Over-cautious / Refusal |
| Fig. 12 | Cryosphere | SIC Estimate SIT | Correct | Threshold / Severity Mis-estimation / Image-feature Misinterpretation | Threshold / Severity Mis-estimation / Image-feature Misinterpretation |
| Fig. 13 | Cryosphere | Sea Ice Extent Estimation | Spatio-temporal Frame Confusion | Spatio-temporal Frame Confusion | Correct |
| Fig. 14 | Lithosphere | P-wave phase picking | Target Mis-location / Domain-knowledge / Semantic Mis-match | Target Mis-location / Domain-knowledge / Semantic Mis-match | Target Mis-location / Domain-knowledge / Semantic Mis-match |
| Fig. 15 | Oceansphere | ENSO Identification | Threshold / Severity Mis-estimation | Spatio-temporal Frame Confusion / Image-feature Misinterpretation | Spatio-temporal Frame Confusion / Image-feature Misinterpretation |
| Fig. 16 | Oceansphere | Marine Fog Detection | Image-feature Misinterpretation | Image-feature Misinterpretation | Image-feature Misinterpretation |
| Fig. 17 | Human-activity sphere | Fine-grained object type recognition | Image-feature Misinterpretation | Image-feature Misinterpretation | Image-feature Misinterpretation |
| Fig. 18 | Lithosphere | earthquake source-receiver distance inference | Correct | Target Mis-location | Over-cautious / Refusal |

Figure 7: A sample case of Movement Prediction subtask.

## Biosphere

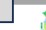

"Question": "This is an unmanned aerial vehicle (UAV) image capturing oil palm trees. How many dead trees are in this image?"
(A) 3
(B) 8
(C) 11
(D) 1
(E) Unable to decide

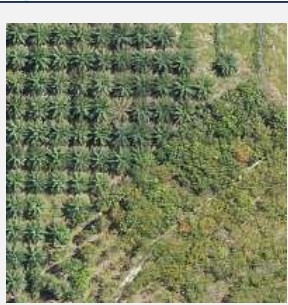

"Correct Answer": "D"

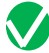

**Gemini: D**

"Reasoning": "Based on the image, I can only identify one tree that appears to be dead or severely damaged. It is located in the upper-middle section of the image, and it has a significantly different color and structure compared to the healthy green oil palm trees."

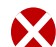 **Over-cautious / Refusal**

**Qwen:E**

"Reasoning": "The image shows a dense plantation of oil palm trees with no clear signs of dead trees. The trees appear uniformly green and healthy, with no visible brown or withered foliage that would indicate dead trees. The image resolution and angle may not be sufficient to detect subtle signs of tree death. Without more detailed information or a closer view, it is not possible to accurately count dead trees. Therefore, the best answer is 'E' - Unable to decide."

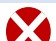 **Over-cautious / Refusal**

**Internvl3:E**

"Reasoning": "The image shows a large number of oil palm trees, but it is difficult to determine the exact number of dead trees without a closer inspection or additional information. The trees appear healthy and green, but there are no clear indicators of dead trees visible from this aerial view."

Figure 8: A sample case of Dead Oil Palm counting subtask.

## Biosphere

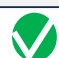

"Question": "Which species is most likely to live in the region shown at the image? The latitude is 60.1805513699° N, and the longitude is 24.9494441843° E."

(A) Fringilla coelebs gengleri
(B) Acridotheres tristis
(C) Passer domesticus
(D) Anthornis melanura
(E) Unable to decide

"Correct Answer": "C"

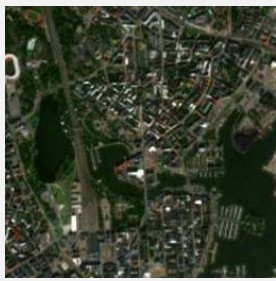

**Gemini: D**

"Reasoning": "The coordinates 60.1805513699° N, 24.9494441843° E indicate Helsinki, Finland. The image shows an urban environment. Passer domesticus (House Sparrow) is a common bird species found in urban areas worldwide, including Europe. The other options are less likely: Fringilla coelebs gengleri is a subspecies of Chaffinch found in the British Isles, Acridotheres tristis (Indian Myna) is native to Asia, and Anthornis melanura (New Zealand Bellbird) is native to New Zealand."

### ❌ Domain-knowledge / Semantic Mis-match

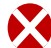

**Qwen: E**

"Reasoning": "The image shows a region with a mix of urban and green spaces, likely in a temperate climate zone. The coordinates place this region in Helsinki, Finland. Fringilla coelebs gengleri, a subspecies of the common chaffinch, is native to northern Europe and well-adapted to such environments. Acridotheres tristis, the common myna, is native to South Asia and not typically found in this region. Passer domesticus, the house sparrow, is common in urban areas but is not specific to this location. Anthornis melanura melanura, the tui, is native to New Zealand and not found in Europe. Therefore, Fringilla coelebs gengleri is the most likely species to live in this region."

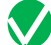

**Internvl3: E**

"Reasoning": "The image shows a densely populated urban area with water bodies, which is characteristic of many cities in temperate regions. The latitude and longitude provided suggest a location in the northern hemisphere, likely in a region with a temperate climate. The species listed are birds, and among them, Passer domesticus, commonly known as the house sparrow, is known to inhabit urban environments. The other species listed are either not typically found in urban settings or are more specific to different regions. Therefore, Passer domesticus is the most likely species to live in the region shown in the image."

Figure 9: A sample case of Species Distribution Prediction subtask.

## Cross-sphere

"Question": "This image shows the satellite view of a bird hotspot, which is located at longitude 34.356469 and latitude -0.738459 in the state of Nyanza, Kenya. …Which of the following bird species is most likely to occur in this hotspot?"
(A)Pitta angolensis
(B)Amadina fasciata
(C)Streptopelia semitorquata
(D)Spatula querquedula
(E)Unable to decide

"Correct Answer": "D"

Gemini: C

"Reasoning": "The image shows a landscape with a mix of vegetation and some built-up areas. The climate data indicates a relatively warm and wet environment. Streptopelia semitorquata (Red-eyed Dove) is a common bird in African woodlands and cultivated areas, which aligns with the habitat and climate suggested by the image and data. The other options are less likely: Pitta angolensis prefers dense forests, Spatula querquedula is a migratory duck and Amadina fasciata prefers drier areas."

### Over-cautious / Refusal

Qwen:B

"Reasoning": "The satellite image shows a region with a mix of green vegetation and some human settlements, indicating a semi-arid to sub-humid environment. The bioclimatic variables suggest a warm climate with moderate precipitation, typical of savanna or woodland habitats. Amadina fasciata, commonly known as the black-rumped shama, is well-adapted to such environments. It prefers open woodlands and savannas, which align with the visual cues from the image. The other options are less likely: Pitta angolensis prefers dense forests, Streptopelia semitorquata is more common in arid regions, Spatula querquedula is a waterfowl requiring wetlands, and selecting 'E' would be too uncertain given the clear habitat match for Amadina fasciata."

### Over-cautious / Refusal

Internvl3:A

"Reasoning": "The satellite image shows a bird hotspot in Nyanza, Kenya, with specific bioclimatic variables. Pitta angolensis, also known as the Angola Pitta, is known to inhabit areas with dense vegetation and a range of temperatures and precipitation levels similar to those described. The image shows a mix of agricultural fields and forested areas, which aligns with the habitat preferences of the Angola Pitta. The other options, such as Amadina fasciata, Streptopelia semitorquata, and Spatula querquedula, have different habitat requirements and are less likely to be found in this type of environment."

Figure 10: A sample case of Most likely species to occur subtask.

## Cross-sphere

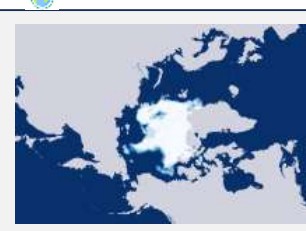

**"Question":** "As a pan-Arctic researcher, you are given one sea ice concentration map (the first image) and one plot for daily trend in sea ice thickness (the second image). Considering the date of SIC map is the following day of the end of SIT trend plot, which choice would most likely to be the averaged SIT of the given SIC map?"

(A)0.946
(B)1.775
(C)1.369
(D)1.052
(E)Unable to decide"

**"Correct Answer": "D"**

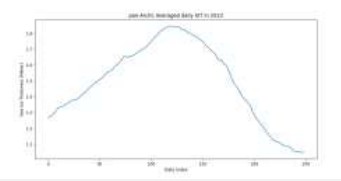

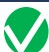

**Gemini: B**

"Reasoning": "The SIC map is for the day after the SIT trend plot ends. The SIT trend plot ends at a date index of approximately 250, where the SIT is around 1.05 meters. Therefore, the most likely average SIT for the SIC map is 1.052."

❌ **Image-feature Misinterpretation**

**Qwen:C**

"Reasoning": "The SIT plot shows a clear downward trend, ending at approximately 1.1 meters. The SIC map indicates a significant amount of sea ice coverage, suggesting that the SIT is not at its lowest point. Given the options, 1.369 meters is the most reasonable choice as it is higher than the endpoint of the SIT plot but lower than the peak, aligning with the expected SIT for the next day."

❌ **Image-feature Misinterpretation**

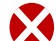

**Internvl3:A**

"Reasoning": "The SIT trend plot shows a decreasing trend in sea ice thickness over time. The date of the SIC map is the following day after the end of the SIT trend plot. By examining the plot, the SIT value at the end of the trend (around day 250) is approximately 1.1 meters. Given the decreasing trend, the SIT on the following day would likely be lower than 1.1 meters. Among the provided choices, 0.946 is the only value lower than 1.1 meters, making it the most likely averaged SIT for the given SIC map."

Figure 11: A sample case of Global Flood Forecasting subtask.

## Cryosphere

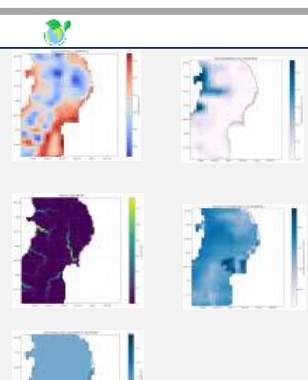

"Question": "The first image is the GloFAS dis24 data, which represents the forecasted river discharge in cubic meters per second (m^3/s). The second image is the ERA5 temperature data at 2 meters above ground level, measured in Kelvin (K). The third image is the volumetric soil water content in the top layer of the soil, measured in cubic meters per cubic meter (m^3/m^3). The fourth image is the snow depth water equivalent, measured in meters (m). The fifth image is the total precipitation sum, measured in millimeters (mm). Whether flood occurs in the region on the date shown in the images ?"

(A)Flooding occurs
(B)Flooding does not occur
(C)Unable to decide"

"Correct Answer": "B"

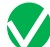

**Gemini: B**

"Reasoning": "The river discharge (dis24) shows moderate levels in the rivers. The total precipitation sum is very low, indicating no significant rainfall. The soil water content is also not particularly high. The snow depth water equivalent is negligible. Therefore, based on the available data, flooding is unlikely to occur."

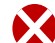 **Threshold / Severity Mis-estimation**

**Qwen:C**

"Reasoning": "The GloFAS dis24 data shows relatively low river discharge values, with no significant areas indicating high discharge that would typically suggest flooding. The temperature data does not show any extreme conditions that would contribute to flooding, such as rapid snowmelt. The soil water content is moderate, with no signs of saturation that could lead to runoff. The snow depth water equivalent is very low, indicating minimal snowmelt contribution. The total precipitation sum is also low, with no heavy rainfall that could cause flooding. However, without additional context or data, such as real-time river levels or local topography, it is not possible to definitively conclude whether flooding occurs or not."

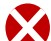 **Domain-knowledge / Semantic Mis-match**

**Internvl3:A**

"Reasoning": "The first image shows high river discharge values, indicating potential flooding. The volumetric soil water content is high, suggesting saturated soil conditions. The snow depth water equivalent is also significant, which could contribute to runoff. The temperature is relatively low, which might slow down evaporation, further increasing the risk of flooding. These factors combined suggest that flooding is likely to occur in the region on the date shown."

Figure 12: A sample case of SIC Estimate SIT subtask.

## Cryosphere

"Question": "As a geoscientist, you are given two sea ice concentration maps, which one describes the sea ice extent in Antarctic region?"

(A) The first sea ice concentration map
(B) The second sea ice concentration map
(C) Both sea ice concentration maps
(D) Neither
(E) Unable to decide

"Correct Answer": "A"

### ❌ Spatio-temporal Frame Confusion

**Gemini: B**

"Reasoning": "The second image shows a significantly reduced sea ice extent compared to the first image. During the melting season, sea ice extent decreases substantially due to warmer temperatures. Therefore, the second image is more likely to represent sea ice extent during the melting season."

### ❌ Spatio-temporal Frame Confusion

**Qwen:B**

"Reasoning": "The second sea ice concentration map shows a significantly reduced area of sea ice compared to the first map. This is indicative of the melting season, where sea ice extent typically decreases. The first map displays a larger and more continuous ice coverage, suggesting it represents a time of year with less melting, such as winter or early spring. The second map's fragmented and reduced ice coverage aligns with the characteristics of the melting season, where ice retreats and breaks up. Therefore, the second map is more likely to represent the sea ice extent during the melting season."

### ✅

**Internvl3:A**

"Reasoning": "The first sea ice concentration map shows a large ice-covered area in the southern hemisphere, which is characteristic of the Antarctic region. The second map shows ice in the northern hemisphere, which is characteristic of the Arctic region. Therefore, the first map describes the sea ice extent in the Antarctic region."

Figure 13: A sample case of Sea Ice Extent Estimation subtask.

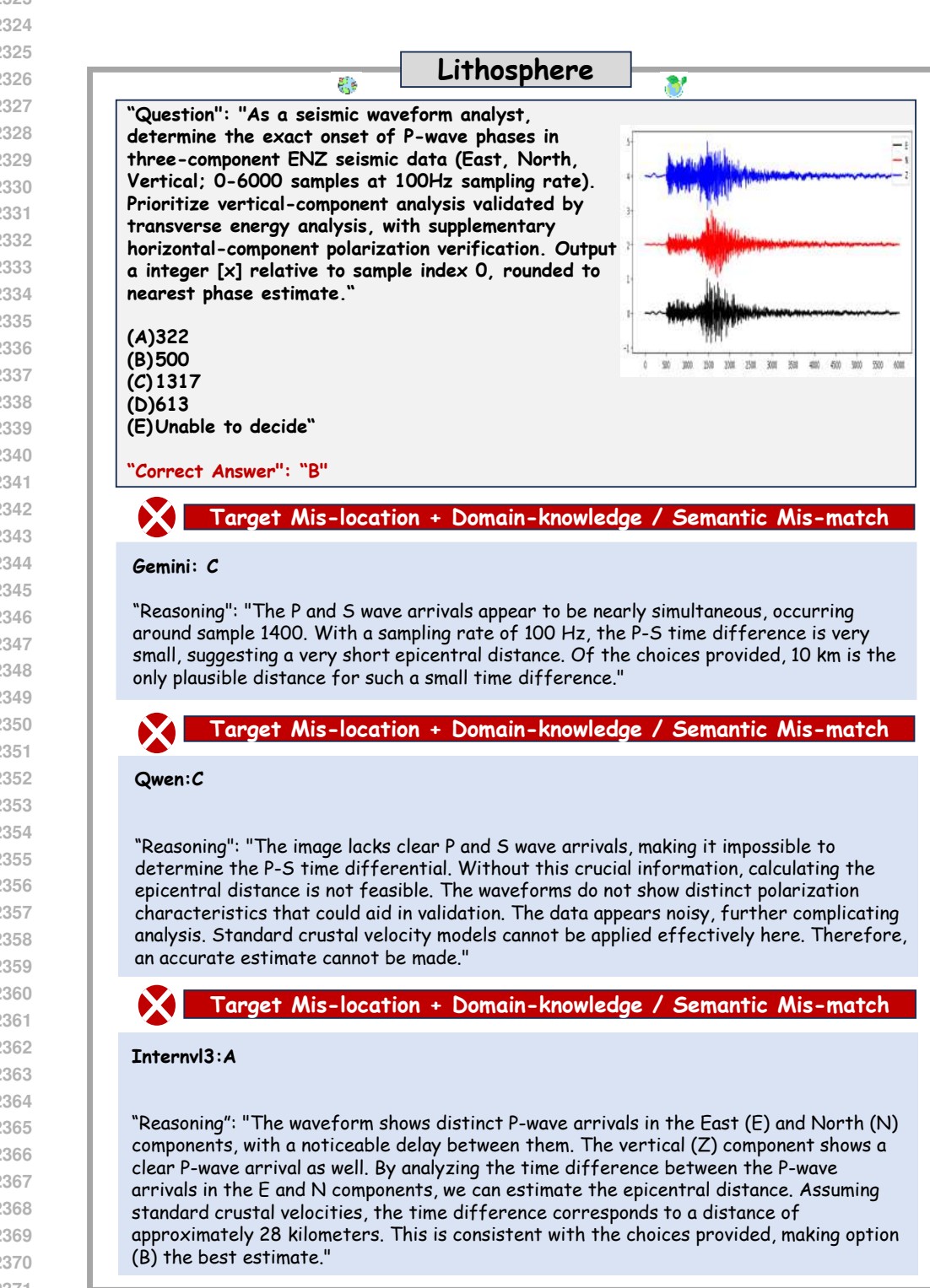

Figure 14: A sample case of P-wave phase picking subtask.

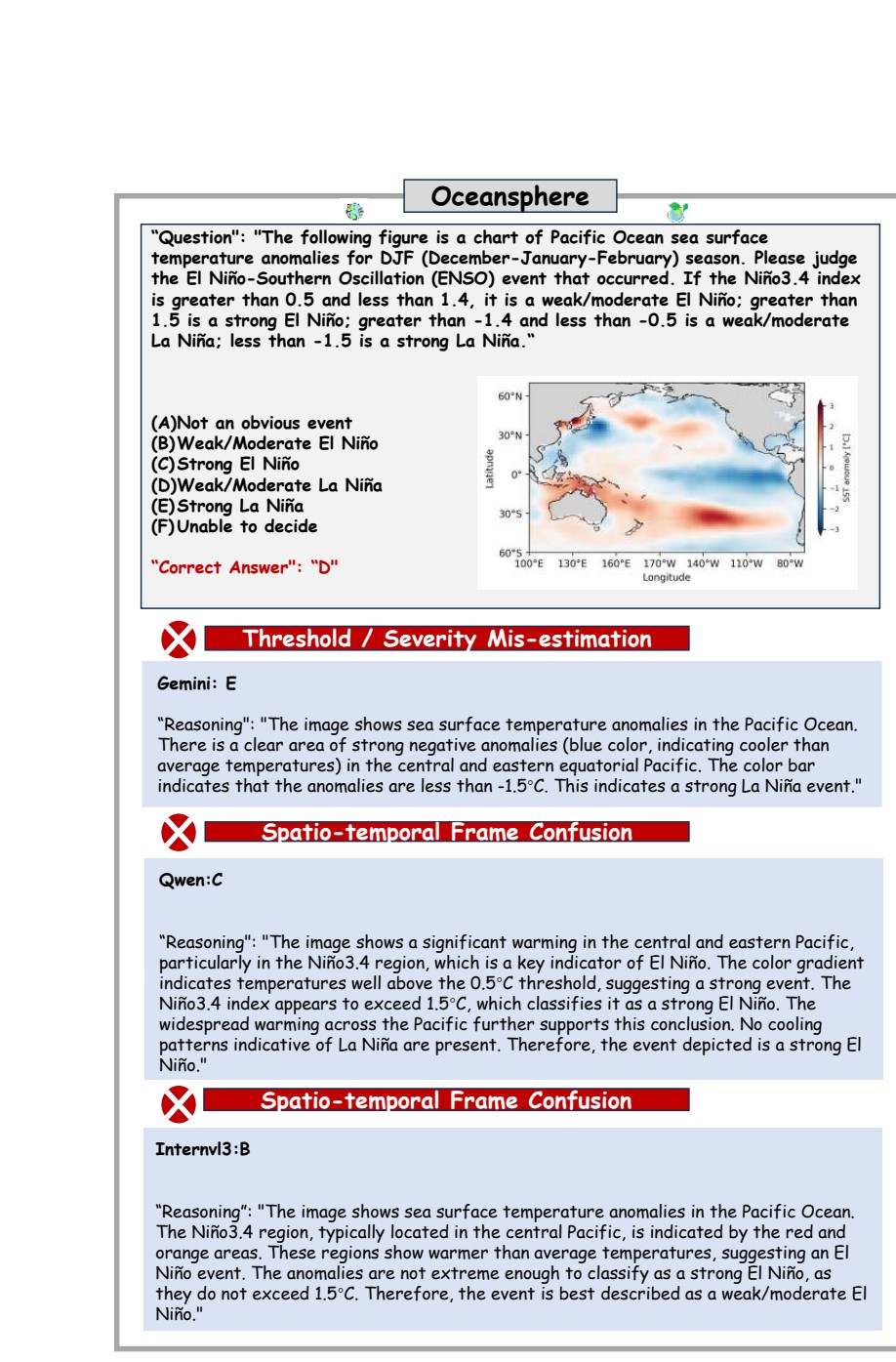

Figure 15: A sample case of ENSO Identification subtask.

## Oceansphere

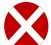 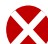

**"Question": "The following image is an RGB image derived using the visible bands of satellite data. Please determine whether there is marine fog in the area shown in the image."**

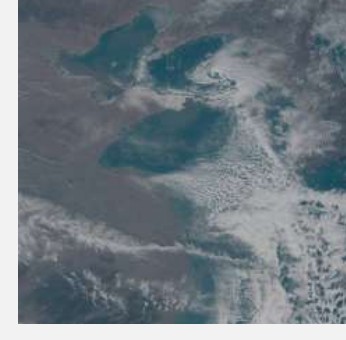

**(A)Yes, there is marine fog**
**(B)No, there is no marine fog**
**(C)Unable to decide**

**"Correct Answer": "B"**

### ❌ Image-feature Misinterpretation

**Gemini: A**

"Reasoning": "The image shows extensive areas of low-lying, featureless white or greyish features over the water. These features are consistent with marine fog or low stratus clouds. The texture and spatial distribution suggest a widespread phenomenon, likely fog."

### ❌ Image-feature Misinterpretation

**Qwen: A**

"Reasoning": "The image shows a large body of water with a significant area covered by a uniform layer of white clouds, which is characteristic of marine fog. The fog appears to be hugging the coastline and extending over the water, a typical pattern for marine fog. The surrounding landmasses are visible with clear distinctions, indicating that the fog is localized over the water. The texture of the cloud layer is smooth and consistent, further suggesting the presence of marine fog. The absence of sharp cloud edges and the gradual transition from the fog to clearer areas support this conclusion."

### ❌ Image-feature Misinterpretation

**Internvl3:A**

"Reasoning": "The image shows a satellite view with visible bands that highlight different features. The presence of marine fog can be inferred from the areas where the water appears lighter and more diffuse compared to the surrounding darker waters. This lighter appearance is characteristic of fog, which scatters light and reduces visibility. Additionally, the fog appears to be concentrated along the coastline, which is typical for marine fog formation."

Figure 16: A sample case of Marine Fog Detection subtask.

## Human-activities sphere

"Question": "Recognize the category of objects from satellite and aerial images given thebounding boxes for referring objects. Bounding box in the format (xmin, ymin,xmax, ymax), where the top left corner is (x_min, y_min) and the bottom-right corner is (x_max, y_max). The resolution of satellite image is 600 × 600. Bounding box:[<132><142><186><187>]"

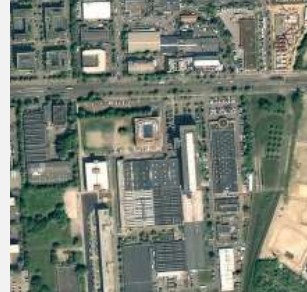

(A)flat_roof
(B)gable_roof
(C)row roof shed
(D)Other
(E)Unable to decide

"Correct Answer": "D"

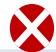 **Image-feature Misinterpretation**

Gemini: A

"Reasoning": "The bounding box [132, 142, 186, 187] refers to a building with a flat roof. The roof appears to be uniformly flat without any visible gables or other roof structures."

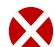 **Image-feature Misinterpretation**

Qwen: A

"Reasoning": "The image shows a large industrial or commercial area with numerous buildings. The roofs of these buildings are predominantly flat, which is a common feature in industrial and commercial architecture. The flat roofs are visible as large, smooth surfaces with no significant slope or gable structures. The presence of parking lots and the layout of the buildings further suggest that these are likely warehouses or factories, which typically have flat roofs for practical reasons such as ease of construction and maintenance. Therefore, the category of objects in the bounding box is best described as 'flat_roof'."

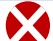 **Image-feature Misinterpretation**

Internvl3:A

"Reasoning": "The bounding box coordinates provided correspond to a flat-roofed structure in the image. The area within the bounding box shows a rectangular shape with a flat surface, typical of flat roofs commonly found in industrial or commercial buildings. The lack of visible sloping edges or peaks further supports this classification."

Figure 17: A sample case of Fine-grained object type recognition subtask.

Figure 18: A sample case of earthquake source-receiver distance inference subtask.

## A.8 DATASHEETS

In this section, we document essential details about the proposed datasets and benchmarks following the CVPR Dataset and Benchmark guidelines and the template provided by Gebru *et al.* Gebru et al. (2021).

### A.8.1 MOTIVATION

The questions in this section are primarily intended to encourage dataset creators to clearly articulate their reasons for creating the dataset and to promote transparency about funding interests. The latter may be particularly relevant for datasets created for research purposes.

1. *"For what purpose was the dataset created?"*

    **A:** Existing benchmarks for Earth science multimodal learning exhibit critical limitations in systematic coverage of geosystem components and cross-sphere interactions, often constrained to isolated subsystems (only in human-activity sphere or atmosphere) with limited evaluation dimensions ($\leq 16$ tasks). To address these gaps, we introduce **OmniEarth-Bench**, the first comprehensive multimodal benchmark spanning all six Earth science spheres (atmosphere, lithosphere, Oceansphere, cryosphere, biosphere and human-activity sphere) and cross-spheres with one hundred expert-curated evaluation dimensions.

2. *"Who funded the creation of the dataset?"*

    **A:** The dataset creation was funded by the affiliations of the authors involved in this work.

### A.8.2 COMPOSITION

Most of the questions in this section are intended to provide dataset consumers with the information they need to make informed decisions about using the dataset for their chosen tasks. Some of the questions are designed to elicit information about compliance with the EUs General Data Protection Regulation (GDPR) or comparable regulations in other jurisdictions. Questions that apply only to datasets that relate to people are grouped together at the end of the section. We recommend taking a broad interpretation of whether a dataset relates to people. For example, any dataset containing text that was written by people relates to people.

1. *"What do the instances that comprise our datasets represent (e.g., documents, photos, people, countries)?"*

    **A:** Our Benchmark comprises not only publicly available open-source datasets but also a significant portion of data manually extracted by experts from satellite imagery and raw observational sources. For example, Vegetation Monitoring uses satellite imagery from MODIS and expert-curated data from the Global Land Surface Satellite (GLASS), including Leaf Area Index, Fractional Vegetation Cover, and Peak Vegetation Coverage Area. All datasets utilized in OmniEarth-Bench are publicly accessible and nonprofit.

2. *"How many instances are there in total (of each type, if appropriate)?"*

    **A:** OmniEarth-Bench consists of seven spheres, with a total of 29,855 annotated data instances.

3. *"Does the dataset contain all possible instances or is it a sample (not necessarily random) of instances from a larger set?"*

    **A:** The images in OmniEarth-Bench are sourced from existing Victor et al. (2024), Teng et al. (2023), Fortier et al. (2024), Huang et al. (2023), Cao & Weng (2024), (Li et al., 2022), Gupta et al. (2019), Astruc et al. (2024), Qian et al. (2023), Veitch-Michaelis et al. (2024), Sastry et al. (2025), Liang et al. (2021), Vermote (2015), Nurseitov et al. (2024a), Sanderson et al. (2002), Zheng et al. (2021a), Veillette et al. (2020), Kitamoto et al. (2023), Hersbach et al. (2020), Mousavi et al. (2019), Kainkaryam et al. (2019), Kikaki et al. (2024), Huang et al. (2017), Wang et al. (2024b), Meier et al. (2021), Comiso (2023), Schweiger et al. (2011), Zhang & Rothrock (2003), Helm et al. (2014), Studinger (2014), Smith et al. (2023) datasets, but all textual annotations were independently created by us.

4. *"Is there a label or target associated with each instance?"*

    **A:** Yes, each instance has been annotated and quality-checked by specialized experts.

5. *"Is any information missing from individual instances?"*

   **A:** No, each individual instance is complete.

6. *"Are relationships between individual instances made explicit (e.g., users movie ratings, social network links)?"*

   **A:** Yes, the relationship between individual instances is explicit.

7. *"Are there recommended data splits (e.g., training, development/validation, testing)?"*

   **A:** The dataset is designed to evaluate the performance of MLLMs across various Earth spheres, so we recommend using it in its entirety as a test set.

8. *"Is the dataset self-contained, or does it link to or otherwise rely on external resources (e.g., websites, tweets, other datasets)?"*

   **A:** OmniEarth-Bench is self-contained and will be open-sourced on platforms like Hugging Face, integrated into evaluation tools such as LLMs-Eval Zhang et al. (2024a); Li et al. (2024a) for easy use.

9. *"Does the dataset contain data that might be considered confidential (e.g., data that is protected by legal privilege or by doctorpatient confidentiality, data that includes the content of individuals non-public communications)?"*

   **A:** No, all data are clearly licensed.

10. *"Does the dataset contain data that, if viewed directly, might be offensive, insulting, threatening, or might otherwise cause anxiety?"*

    **A:** No, OmniEarth-Bench does not contain any data with negative information.

### A.8.3 COLLECTION PROCESS

In addition to the goals outlined in the previous section, the questions in this section are designed to elicit information that may help researchers and practitioners create alternative datasets with similar characteristics. Again, questions that apply only to datasets that relate to people are grouped together at the end of the section.

1. *"How was the data associated with each instance acquired?"*

   **A:** The images in OmniEarth-Bench are sourced from existing Victor et al. (2024), Teng et al. (2023), Fortier et al. (2024), Huang et al. (2023), Cao & Weng (2024), (Li et al., 2022), Gupta et al. (2019), Astruc et al. (2024), Qian et al. (2023), Veitch-Michaelis et al. (2024), Sastry et al. (2025), Liang et al. (2021), Vermote (2015), Nurseitov et al. (2024a), Sanderson et al. (2002), Zheng et al. (2021a), Veillette et al. (2020), Kitamoto et al. (2023), Hersbach et al. (2020), Mousavi et al. (2019), Kainkaryam et al. (2019), Kikaki et al. (2024), Huang et al. (2017), Wang et al. (2024b), Meier et al. (2021), Comiso (2023), Schweiger et al. (2011), Zhang & Rothrock (2003), Helm et al. (2014), Studinger (2014), Smith et al. (2023) datasets, all textual annotations were independently created by experts.

2. *"What mechanisms or procedures were used to collect the data (e.g., hardware apparatuses or sensors, manual human curation, software programs, software APIs)?"*

   **A:** Our Benchmark comprises not only publicly available open-source datasets but also a significant portion of data manually extracted by experts from satellite imagery and raw observational sources. For example, Vegetation Monitoring uses satellite imagery from MODIS and expert-curated data from the Global Land Surface Satellite (GLASS), including Leaf Area Index, Fractional Vegetation Cover and Peak Vegetation Coverage Area. Moreover, for the Eddy data in oceansphere, the chlorophyll (CHL) data used in this study were obtained by applying the OCI empirical algorithm to Level-2 data acquired by the Geostationary Ocean Color Imager I (GOCI) aboard the Oceanography and Meteorology Satellite (COMS). After careful selection and integration, we compiled a comprehensive dataset covering 33 different data sources across all Earth spheres. Tab.2 is a summary of the data sources used for each Earth sphere.

3. *"If the dataset is a sample from a larger set, what was the sampling strategy (e.g., deterministic, probabilistic with specific sampling probabilities)?"*

   **A:** Please refer to the details listed in the main text Section 3.1.

### A.8.4 Preprocessing, Cleaning, and Labeling

The questions in this section are intended to provide dataset consumers with the information they need to determine whether the raw data has been processed in ways that are compatible with their chosen tasks. For example, text that has been converted into a "bag-of-words" is not suitable for tasks involving word order.

1. *"Was any preprocessing/cleaning/labeling of the data done (e.g., discretization or bucketing, tokenization, part-of-speech tagging, SIFT feature extraction, removal of instances, processing of missing values)?"*

   **A:** Yes. During image collection, we prioritized selecting valuable satellite images for annotation. For linguistic annotation, three Level-3 subtasksRegional Land Use Classification, Regional Counting, and Regional Counting with Change Detectionwere marked with red circles. This method, mimicking human interaction, was essential for providing clear, fine-grained region-level analysis on ultra-high-resolution images.

2. *"Was the 'raw' data saved in addition to the preprocessed/cleaned/labeled data (e.g., to support unanticipated future uses)?"*

   **A:** Yes, raw data is accessible.

3. *"Is the software that was used to preprocess/clean/label the data available?"*

   **A:** Yes, the necessary software used to preprocess and clean the data is publicly available.

### A.8.5 Uses

The questions in this section are intended to encourage dataset creators to reflect on tasks for which the dataset should and should not be used. By explicitly highlighting these tasks, dataset creators can help dataset consumers make informed decisions, thereby avoiding potential risks or harms.

1. *"Has the dataset been used for any tasks already?"*

   **A:** No.

2. *"Is there a repository that links to any or all papers or systems that use the dataset?"*

   **A:** Yes, we will provide such links in the GitHub and the Huggingface repository.

3. *"What (other) tasks could the dataset be used for?"*

   **A:** OmniEarth-Bench is suitable for various tasks across other Earth spheres. It covers 103 subtasks spanning six major Earth spheres plus Cross-sphere, and is capable of handling various other tasks.

4. *"Is there anything about the composition of the dataset or the way it was collected and preprocessed/cleaned/labeled that might impact future uses?"*

   **A:** No.

5. *"Are there tasks for which the dataset should not be used?"*

   **A:** N/A.

### A.8.6 Distribution

Dataset creators should provide answers to these questions prior to distributing the dataset either internally within the entity on behalf of which the dataset was created or externally to third parties.

1. *"Will the dataset be distributed to third parties outside of the entity (e.g., company, institution, organization) on behalf of which the dataset was created?"*

   **A:** No. The datasets will be made publicly accessible to the research community.

2. *"How will the dataset be distributed (e.g., tarball on website, API, GitHub)?"*

   **A:** We will provide OmniEarth-Bench in the GitHub and the Huggingface repository.

3. *"When will the dataset be distributed?"*

   **A:** We will create a repository to release the data once the paper is officially published, ensuring compliance with the anonymity principle.

4. *"Will the dataset be distributed under a copyright or other intellectual property (IP) license, and/or under applicable terms of use (ToU)?"*

    **A:** Yes, the dataset will be released under the Creative Commons Attribution-NonCommercial-ShareAlike 4.0 International License.

5. *"Have any third parties imposed IP-based or other restrictions on the data associated with the instances?"*

    **A:** No.

6. *"Do any export controls or other regulatory restrictions apply to the dataset or to individual instances?"*

    **A:** No.

### A.8.7 MAINTENANCE

As with the questions in the previous section, dataset creators should provide answers to these questions prior to distributing the dataset. The questions in this section are intended to encourage dataset creators to plan for dataset maintenance and communicate this plan to dataset consumers.

1. *"Who will be supporting/hosting/maintaining the dataset?"*

    **A:** The authors of this work serve to support, host, and maintain the datasets.

2. *"How can the owner/curator/manager of the dataset be contacted (e.g., email address)?"*

    **A:** The curators can be contacted via the email addresses listed on our paper or webpage.

3. *"Is there an erratum?"*

    **A:** There is no explicit erratum; updates and known errors will be specified in future versions.

4. *"Will the dataset be updated (e.g., to correct labeling errors, add new instances, delete instances)?"*

    **A:** Future updates (if any) will be posted on the dataset website.

5. *"Will older versions of the dataset continue to be supported/hosted/maintained?"*

    **A:** Yes. This initial release will be updated in the future, with older versions replaced as new updates are posted.

6. *"If others want to extend/augment/build on/contribute to the dataset, is there a mechanism for them to do so?"*

    **A:** Yes, we will provide detailed instructions for future extensions.

## A.9 LIMITATION AND POTENTIAL SOCIETAL IMPACT

In this section, we discuss the limitations and potential societal impact of this work.

### A.9.1 POTENTIAL LIMITATIONS

While **OmniEarth-Bench** provides a comprehensive benchmark for evaluating the perception and reasoning capabilities of MLLMs, there are several limitations to consider:

- **Scope of Sensors:** Although our benchmark includes 29,855 annotations and 109 subtasks, it may not cover all possible real-world scenarios. There could be additional sensor data, like multispectral data that were not included in this study, potentially limiting the generalizability of our findings.

- **Model and Dataset Diversity:** In this paper, we extensively evaluated general-purpose MLLMs. As new models emerge, their evaluation results will be added to our open-source leaderboard. Additionally, OmniEarth-Bench will also be expanded in dataset size and task diversity.

### A.9.2 POTENTIAL NEGATIVE SOCIETAL IMPACT

- **Safety Risks:** OmniEarth-Bench is designed to evaluate the performance of vision-language multimodal models in six spheres and cross-sphere scenarios. However, excessive reliance on evaluation datasets may lead to overconfidence in autonomous systems, such as multimodal large models. It is crucial to implement adequate safety measures and human supervision when deploying these MLLMs to ensure public safety.

- **Environmental Impact:** Training MLLMs on large datasets and evaluating them using OmniEarth-Bench requires a certain amount of computational resources. To facilitate future research, we will maintain a leaderboard of MLLMs, removing the need for repeated evaluations of existing models.

### A.10 USAGE OF LLM

**Writing Assistance** LLMs were utilized as an auxiliary tool in the preparation and refinement of this manuscript. This involved tasks such as proofreading for grammatical consistency, improving sentence structure for better flow, and rephrasing complex technical descriptions to enhance clarity for a broader audience. The authors conducted a thorough review and editing of all AI-suggested text to ensure the scientific accuracy and integrity of the final content. The authors retain full responsibility for all statements, claims, and conclusions presented in this work.

**Code and Script Development** During the development phase, LLMs were employed to accelerate the creation of various scripts. This included generating boilerplate code for data processing pipelines (e.g., remapping and cropping algorithms), developing utility functions for data handling, and assisting in debugging evaluation protocols. All code generated or modified with the assistance of LLMs was manually verified, tested, and optimized by the authors to ensure its correctness, efficiency, and adherence to the project's requirements.

**Benchmark Content Generation and Evaluation** Beyond supporting tasks, LLMs were integral to the methodology for generating and evaluating parts of the OmniEarth-Bench dataset itself.

- **Initial Caption Generation:** For the image captioning task, a state-of-the-art LLM was used to generate preliminary scientific captions by synthesizing information from multi-spectral satellite imagery and corresponding textual disaster reports.

- **Automated Evaluation:** In the open-ended question format, an LLM served a critical role as an automated judge to assess the semantic correctness of model-generated answers against the ground-truth answers.

It is essential to note that all content generated by LLMs for the benchmark (i.e., captions) underwent a meticulous review and validation process by domain experts to guarantee scientific accuracy and factual consistency.

