# OpenReview forum: "OmniEarth-Bench: Probing Cognitive Abilities of MLLMs for Earth's Multi-sphere Observation Data"
_ICLR.cc/2026/Conference — ICLR 2026 Conference Withdrawn Submission_

### Official Review · Reviewer_srAo · 2025-10-30

**Soundness:** 3
**Presentation:** 3
**Contribution:** 3
**Rating:** 6
**Confidence:** 4

**Summary:**

This work introduces a new benchmark for MLLMs on earth observation. Most existing earth observation benchmarks only focus on a single "sphere", e.g., human activities sphere or atmosphere (weather), whereas OmniEarth-Bench includes data from 6 spheres as well as cross-sphere interactions. Evaluations are also performed on a number of modern MLLMs and demonstrate relatively poor performance on OmniEarth-Bench, motivating future research on developing more scientifically capable MLLMs.

**Strengths:**

When released, OmniEarth-Bench will certainly be a great contribution to the community. Gathering all the data and expert annotations is an expensive endeavor and will hopefully accelerate future research in this area.

**Weaknesses:**

- Just to clarify, all the questions and categorizations (L1-L4) were created by domain experts and not LLMs?
- The related works section could be clearer about which of the data sources in OmniEarth-Bench (Table 2) are not already incorporated in a major ML benchmark? e.g., ERA5 is definitely a well-established benchmark for weather prediction models. Additionally, it sounds like the main contribution of OmniEarth-Bench is the sourcing of questions to go with the data in all the datasets, not just aggregating the data in a ML-ready package. Is this true? For non-earth science experts, these details will make your work more clearly distinguishable.
- Similarly, the text strongly emphasizes including 'cross-sphere interactions', but it's not clear to me what exactly the impact is. e.g., line 154: "[other benchmarks are] neglecting cross-sphere interactions essential to real-world Earth science challenges." What challenges exactly couldn't be captured in previous benchmarks that is captured now?
- On open-ended scoring via LLM-as-a-judge: What evals were done on this scoring mechanism? For example, I could see LLMs struggling with numerical tasks like comparing very similar numbers (within eps-tolerance) that are not exactly the same.
- how are visual grounding and captioning performance scored? this didn’t seem to be covered in A.4
- What is the role of the ENSO example in Sec 4.3? It comes a bit out of the blue and only has GPT-4o prediction. There are no details on the prompting setup or any other baselines. I would recommend either removing this or making it a proper task evaluated with the rigor of everything else in Sec 4.2.

**Questions:**

See 'weaknesses' for major questions.

Minor questions:
- Maybe L3 (the type of analysis needed to solve the problem) should actually be “L1”? since L1 L2 and L4 seem to be specific to the problem while “L3” describe general skills that MLLM needs to solve the problems. No need to change the paper for rebuttals, just a thought for future work.
- Table 3 says there are 27k MCQ and 27k open-ended when the total is 29k?
- Table 4: why are open-ended questions only evaluated on a small subset of models used for MCQ?

---

> ### Author Response · Authors · 2025-11-20
>
> ### Weakness1: All the questions and categorizations (L1-L4) were created by domain experts and not LLMs?
> Yes. As shown in Figure 3 of the main text, our pipeline comprises 4 stages: Source Screening, Task Construction, Dataset Construction, and Quality Control, all led by experts. The first two stages are exclusively conducted by experts, while crowd-sourcing annotators assist in the latter two stages. MLLMs are involved only in Stage D and solely as part of quality control.
>
> Although fully automating the creation of all questions and the L1–L4 taxonomy with an MLLM may appear promising, it is not feasible for constructing reliable evaluation items in a highly specialized domain such as Earth science. We discuss this in **Appendix A.2 (“More Details of OmniEarth-Bench,” lines 902–910)**.
>
> We appreciate your constructive comment; we believe the appendix discussion may also inform data-annotation practices in other scientific settings.
>
> > Human Annotations vs. GPT Annotations. All annotations are finished by experts and crowdsourcing annotators. Unlike MMBench Liu et al.(2024), we did not use tools like GPT-4o OpenAI(2024). It was driven by two key reasons: (1) GPT-4o cannot generate VQA data requiring deep
> > domain expertise. Tasks under the Scientific-Knowledge Reasoning (L3) demand substantial background knowledge and must be constructed collaboratively by experts. (2) Although GPT-4o can generate samples for general perception or simple reasoning tasks, expert evaluation found the data to be low quality and insufficiently challenging. For example, in the visual grounding task, GPT4o only detects highly salient structures, failing to support our goal of testing MLLMs on locating diverse buildings across complex scenes. As a result, all OmniEarth-Bench data was exclusively created by experts and annotators.

---

> ### Author Response · Authors · 2025-11-20
>
> ### Weakness2: The related works section could be clearer about which of the data sources in OmniEarth-Bench (Table 2) are not already incorporated in a major ML benchmark?  OmniEarth-Bench is the sourcing of questions to go with the data in all the datasets, not just aggregating the data in a ML-ready package?
>
> Yes. Our contribution is not a conventional machine-learning benchmark; it is a vision–language reasoning benchmark for multimodal LLMs. While there has been preliminary exploration in the atmospheric domain (WeatherQA, ClimateIQA, CLLMate) and in the remote-sensing domain (AgroMind, XLRS-Bench, DynamicVL, DisasterM3), these efforts do not provide a system-level evaluation of Earth-science understanding and scientific reasoning across spheres. OmniEarth-Bench is proposed to fill precisely this gap—offering a comprehensive, capability-aligned benchmark that tests perception, reasoning, and domain-grounded inference over Earth-system data rather than isolated, single-scene tasks.
>
> To our knowledge, most data sources used by OmniEarth-Bench have appeared primarily as single-task, single-modality datasets (classification, detection, segmentation, or numeric forecasting) and are rarely included in general-purpose MLLM evaluations. OmniEarth-Bench consolidates these under-represented Earth-system sources into a capability-aligned, cross-sphere vision-language benchmark, enabling evaluation of perception, generic reasoning, and scientific-knowledge reasoning grounded in real observation products—rather than isolated single-scene tasks. We will add a short subsection in Related Work explicitly listing these sources and noting their (non)coverage in mainstream ML/MLLM benchmarks.
>
> > **Earth-observation sources under-represented in MLLM benchmarks.**
> > While several EO datasets are widely used in single-task machine learning (e.g., detection on xView; forecasting on SEVIR/ERA5; seismic segmentation on TGS-Salt), they are rarely incorporated into general-purpose multimodal LLM benchmarks. In particular, cross-sphere products such as GFF/GloFAS (flood), SatBird (species distribution), and CarbonSense (NEE) typically appear in domain-specific pipelines, not as vision-language tasks. Likewise, biosphere (TreeSatAI, OAM-TCD, GLASS, ROSID, HFP, MOPAD), human-activity (UBCv1, WHU-OHS), oceansphere (MADOS, ERASSTv5, COMS), and cryosphere (G02202/PIOMAS/CryoSat-2/IceBridge) datasets are seldom evaluated in MLLM settings. OmniEarth-Bench reframes these sources as unified V-L items aligned to abilities (Perception / General Reasoning / Scientific-Knowledge Reasoning) and cross-sphere scenarios, thereby moving from single-scene ML tasks to a system-level Earth-science evaluation for MLLMs.
>
>
> Indeed. The entries in Table 2 enumerate only the raw data sources and correspond to the first step of our data-production pipeline (Fig. 4). We provide a detailed account of how these sources are utilized—including sphere-specific processing choices and harmonization protocols—in **Appendix A.3, “DATA PROCESSING STRATEGIES FOR THE SIX SPHERES” (lines 1404–1506)**.
>
> > For atmospheric weather events, we primarily use reanalysis datasets such as ERA5, with meteorological variables visualized as RGB images that include variable names, legends, latitude-longitude grids, and national boundaries.
> > Short- and Medium-Term Events: We compiled global event records from 1979-2020, retrieved corresponding timestamps from ERA5, and used 1-hour and 6-hour intervals for shortand medium-term events, respectively. For multivariable or long-duration cases, the interval was
> > increased fourfold to limit image volume.

---

> ### Author Response · Authors · 2025-11-20
>
> ### Weakness3: 'cross-sphere interactions'. What challenges exactly couldn't be captured in previous benchmarks that is captured now?
> Thank you for this thoughtful question.
>
> **(1) Recent works.**
>
> To clarify why cross-sphere tasks inherently require multi-sphere reasoning rather than reasoning from a single sphere, we refer to recent findings published in **Nature Geoscience**, a leading journal in the Earth sciences.
>
> **i) Stratospheric aerosol perturbation by tropospheric biomass burning and deep convection**
>
> This study demonstrates that stratospheric aerosol disturbances are strongly modulated by biomass-burning emissions and deep convective transport. Although biomass burning and land-surface ecology belong to the lithosphere–biosphere, their downstream impacts manifest in the atmospheric domain. This is a canonical example of a physical mechanism that cannot be inferred from atmospheric variables alone.
>
> **ii) Enhanced West Antarctic ice loss triggered by polynya response to meridional winds**
>
> This work shows that persistent northerly wind anomalies—an atmospheric variable—close coastal polynyas, leading to anomalous ocean warming and freshening near ice shelves. Here, atmospheric forcing triggers an oceanic response, which then affects the cryosphere.
>
> **Overall**, earth scientists routinely combine information from multiple spheres to explain such mechanisms. Our goal is to test whether MLLMs can perform analogous cross-sphere reasoning, rather than relying on isolated single-sphere cues. This is precisely why OmniEarth-Bench includes a dedicated cross-sphere dimension.
>
> ---
>
> **(2)OmniEarth-Bench.**
>
> Our cross-sphere tasks are not created by simply concatenating several independent single-sphere questions. Instead, they are drawn from Earth-science problems whose ground-truth labels are *defined by coupled processes across spheres*. We clarify this from both a task-design and an experimental perspective.
>
> **i) our cross-sphere tasks were constructed from intrinsically coupled processes**
>
> The three L2 scenarios under the Cross-sphere dimension—Global Flood Forecasting, Bird Species Prediction, and Carbon Flux Monitoring—all stem from settings where domain experts agree that multiple spheres must be considered jointly:
>
> * **Global Flood Forecasting.** Whether a given location will experience flooding is determined by the **basin-scale water balance**, which inherently couples several spheres:
>   * **Atmosphere:** antecedent and ongoing precipitation, temperature (e.g., snowmelt), and extreme rainfall events;
>   * **Hydrosphere / Lithosphere:** soil moisture state, surface runoff response, river-network routing; snowpack / snow water equivalent (SWE) and its melt contribution.
>
> * **Bird Species Prediction.** Bird species distributions are controlled by:
>   * **Atmospheric / climatic factors:** climate zone, temperature and precipitation regimes;
>   * **Anthroposphere:** land use and human footprint.
>
>     The original SatBird work models occurrence using multi-source environmental covariates precisely because many locations share similar values in any single dimension (e.g., land cover *or* temperature alone) yet host different species.
>
> * **Carbon Flux Monitoring.** Net ecosystem carbon flux (NEE) necessarily depends on both:
>   * **Biosphere:** vegetation state (e.g., LAI, greenness, phenology), ecosystem type;
>   * **Atmosphere:** radiation, temperature, humidity, wind and related meteorological drivers.
>
> **ii) Empirical evidence from single-sphere ablations**
>
> To complement the above design-based arguments, we have added a lightweight ablation study on representative cross-sphere subtasks for a strong open-source MLLM GPT-5 (the best-performing model in our main results).
> 1. Full multi-sphere (original) – all spheres’ inputs are provided as in OmniEarth-Bench.
> 2. Single-sphere only – we retain only a subset of inputs from a *single* sphere (e.g., only atmospheric fields such as temperature and precipitation), while all other images or texts are blanked out.
>
> |Global Flood Forecasting(GFF)|Full|Atmosphere-only|Hydro/Litho-only|
> |-|-|-|-|
> |Accuracy(%)|22.46|17.29|20.37|
> |ΔvsFull(pp)|—|-5.17|-2.09|
>
> |Bird Species Prediction|Full|Atmosphere-only|Anthroposphere-only|
> |-|-|-|-|
> |Accuracy(%)|22.90|18.13|13.84|
> |ΔvsFull(pp)|—|-4.77|-9.06|
>
> |Carbon Flux Monitoring|Full|Atmosphere-only|Biosphere-only|
> |-|-|-|-|
> |Accuracy(%)|61.54|46.44|28.93|
> |ΔvsFull(pp)|—|-15.1|-32.61|
>
> As reported in the above Tables, the model’s accuracy in the full multi-sphere setting is consistently and substantially higher than in any single-sphere-only setting. This pattern indicates that: no individual sphere alone carries sufficient information to solve our cross-sphere tasks robustly.
>
> Together, the coupled-process origin of the labels and the new ablation results provide converging evidence that our cross-sphere tasks genuinely require multi-sphere reasoning rather than independent, parallel single-sphere classification.

---

> ### Author Response · Authors · 2025-11-20
>
> ### Weakness4:On open-ended scoring via LLM-as-a-judge: What evals were done on this scoring mechanism?
>
> Thanks for your insightful comments. We acknowledge that the main text and appendix did not provide a detailed description of the **LLM-as-a-judge** protocol used in this work. Accordingly, we now offer a more thorough specification and plan to present it in the appendix together with the evaluation criteria for visual grounding and related tasks. We are grateful for this helpful reminder.
>
> We standardize the evaluation of **open-ended** tasks along three dimensions:
>
> **(1) Deterministic numerical verification** We explicitly incorporated numerical tolerance rules within the system prompt to prevent the judge from hallucinating correctness or being overly pedantic. The prompt instructs the model to calculate validity based on the magnitude of the Ground Truth (GT). Specifically, for small values ($|GT| < 1$), we allow an absolute error of $\pm0.1$. For larger values, we allow a relative error of $\pm10\%$ (if $|GT| < 100$) or a stricter $\pm5\%$ (if $|GT| \geq 100$). Additionally, for vector-valued answers (e.g., bounding boxes), the model checks element-wise alignment, ensuring the scoring logic is mathematically grounded rather than linguistically inferred.
>
> **(2) Task-specific rule** We customized the evaluation criteria based on the specific L4 subtask type to minimize false negatives. For categorical tasks (e.g., land cover or directions), the prompt is instructed to accept domain-specific synonyms to avoid penalizing linguistic variations. For reasoning tasks (e.g., complex Yes/No questions), we enforce a stricter standard where credit is awarded *only* if both the binary decision and the supporting explanation align with the ground truth. Similarly, for region identification, the system allows for semantic correctness (e.g., ecological synonyms or hierarchical equivalents) rather than relying on rigid string matching.
>
> **(3) Structured output and score control** To further mitigate the instability of free-form generation, we enforce a strict JSON output format containing `reason` and `score` fields.
>
> - **`reason`**: A brief justification of the decision process, providing interpretability for the score.
> - **`score`**: A discrete value of 0, 0.5, or 1. The 0.5 score captures borderline cases (e.g., predicting a geographically adjacent region in a localization task). However, to ensure a rigorous benchmark, we strictly treat 0.5 as 0 when calculating the final Top-1 Accuracy reported in the paper.
>
>
> ### Weakness5:how are visual grounding and captioning performance scored?
>
> Thank you for pointing this out—our oversight was not to explicitly enumerate all evaluation metrics. While our released code reproduces the metrics defined in the main text, we will make the specification explicit in the paper and appendix. Specifically, we will add:
>
> > For the Grounding task, we used precision, assessing accuracy based on the intersection between predicted and ground truth bounding boxes, with predictions deemed correct if IoU exceeds a threshold. We tested two IoU thresholds: 0.5 and 0.7. For the captioning task, we used standard metrics, including BLEU [1], ROUGE L [2], and METEOR [3]. We consider the n-gram precision for BLEU with n-values of 1, 2, 3, and 4.
>
> For both tasks, we follow established, peer-reviewed benchmarks—e.g., XLRS-Bench (CVPR 2025) [4] and VRSBench (NeurIPS 2025) [5]—which adopt the same grounding and captioning metrics. We will place these definitions in a dedicated **“Evaluation Metrics”** subsection (appendix), alongside implementation details and configuration parameters to ensure full reproducibility.
>
>
> [1] Papineni, K., Roukos, S., Ward, T., & Zhu, W. J. (2002, July). Bleu: a method for automatic evaluation of machine translation. In Proceedings of the 40th annual meeting of the Association for Computational Linguistics (pp. 311-318).
>
> [2]Lin, C. Y. (2004, July). Rouge: A package for automatic evaluation of summaries. In Text summarization branches out (pp. 74-81).
>
> [3]Banerjee, S., & Lavie, A. (2005, June). METEOR: An automatic metric for MT evaluation with improved correlation with human judgments. In Proceedings of the acl workshop on intrinsic and extrinsic evaluation measures for machine translation and/or summarization (pp. 65-72).
>
> [4]Li, X., Ding, J., & Elhoseiny, M. (2024). Vrsbench: A versatile vision-language benchmark dataset for remote sensing image understanding. Advances in Neural Information Processing Systems, 37, 3229-3242.
>
> [5]Wang, F., Wang, H., Guo, Z., Wang, D., Wang, Y., Chen, M., ... & Sun, M. (2025). Xlrs-bench: Could your multimodal llms understand extremely large ultra-high-resolution remote sensing imagery?. In Proceedings of the Computer Vision and Pattern Recognition Conference (pp. 14325-14336).

---

> ### Author Response · Authors · 2025-11-20
>
> ### Weakness6:What is the role of the ENSO example in Sec 4.3?
>
> The ENSO example was included primarily to illustrate the limits of multimodal LLMs in **long-range temporal reasoning**. However, as you correctly pointed out, this example is not essential to the core benchmark design and can be safely removed. We will revise the manuscript accordingly.
>
> Recent studies in Nature and Science Advances—such as “Future increase in extreme El Niño supported by past glacial changes” (Nature) and “A self-attention–based neural network for three-dimensional multivariate modeling and its skillful ENSO predictions” (Science Advance)—demonstrate that ENSO can now be predicted up to 16 months ahead. Therefore，we align with these developments by extending OmniEarth-Bench’s prediction horizons to 16 or even 24 months.
>
> Figure 6 shows GPT-4o’s accuracy dropping to near zero at 12-month leads, exposing current MLLM limitations for long-range ENSO and IOD forecasting. As a pioneering study, this work could set new standards by boldly targeting longer-term predictions and benchmarking against state-of-the-art baselines.
>
> To test the limits of current MLLMs we extended the lead-time grid from 12 months to 14, 16, 18, 24 months while keeping all other settings unchanged.
>
> | Lead time (months) |     1 |     3 |     6 |     9 |   12 |   14 |   16 |   18 |   24 |
> | -----------------: | ----: | ----: | ----: | ----: | ---: | ---: | ---: | ---: | ---: |
> |           **ENSO** | 37.25 | 38.56 | 29.41 | 15.79 | 4.61 | 6.62 | 7.95 | 4.64 | 2.67 |
> |            **IOD** | 18.49 |  6.85 |  0.00 |  4.83 | 0.69 | 0.00 | 0.69 | 0.00 | 0.00 |
>
> Accuracy falls off rapidly beyond one year, underscoring the difficulty of multi-year teleconnections.
>
> Overall, we agree that ENSO and IOD represent highly valuable and scientifically meaningful problems. However, as you correctly noted, introducing them abruptly in the main experimental results may appear disconnected from the core benchmark narrative. Accordingly, we will move this analysis to the appendix as an independent subsection, where it will be framed as a discussion of potential future extensions of OmniEarth-Bench and its practical scientific relevance.

---

> ### Author Response · Authors · 2025-11-20
>
> ### Minor questions 1:Maybe L3 (the type of analysis needed to solve the problem) should actually be “L1”?
>
>
> Your suggestion aligns very well with recent trends in high-impact scientific benchmarks. After reviewing two representative benchmarks published in leading venues—**ChemBench (Nature Chemistry)** [1] and **MaCBench (Nature Computational Science)** [2]—we found a structure that resonates strongly with your proposed idea.
>
> ChemBench organizes its benchmark along **eight chemistry-related thematic domains**, which is conceptually analogous to our hierarchical **L1–L2–L4** decomposition of Earth-science spheres, sub-scenarios, and tasks. At the same time, ChemBench defines three **orthogonal skill dimensions**—**computation**, **reasoning**, and **knowledge**—which correspond closely to our **L3 capability layer**. Importantly, ChemBench does *not* embed capabilities into the thematic hierarchy; instead, theme and capability form **two independent axes**, allowing each question to be jointly assigned to a domain *(e.g., Reaction Dynamics)* and a capability *(e.g., Reasoning)*.
>
> Your suggestion is therefore entirely consistent with this design principle. Inspired by ChemBench, we can indeed consider **decoupling L3 capabilities from the hierarchical L1–L2–L4 structure** and treating them as an independent skill axis. This would enable a richer cross-classification—each OmniEarth-Bench item would lie at the intersection of an Earth-science domain (L1/L2/L4) and a capability category (L3). We believe this dual-axis structure could further strengthen OmniEarth-Bench and enhance its alignment with emerging standards in scientific multimodal evaluation.
>
> We appreciate your insightful comment—it points toward a constructive direction for future refinement of the benchmark.
>
> [1]Mirza, Adrian, et al. "A framework for evaluating the chemical knowledge and reasoning abilities of large language models against the expertise of chemists." Nature Chemistry (2025): 1-8.
>
> [2]Alampara, Nawaf, et al. "Probing the limitations of multimodal language models for chemistry and materials research." Nature computational science (2025): 1-10.
>
> ### Minor questions 2:Table 3 says there are 27k MCQ and 27k open-ended when the total is 29k?
>
> Our multiple-choice questions and open-ended questions represent **two answer formats for the same underlying items**. Since both formats are commonly used in contemporary multimodal LLM evaluation, we provide both for all ~27k questions and evaluate them separately.
>
> The reported total of **29k items** reflects this accounting: the two answer formats for the 27k core questions are counted **once**, and we additionally include Visual Grounding questions (2,697), CoT questions (610), and Image Captioning items (76), yielding approximately 29k items in total.

---

> ### Author Response · Authors · 2025-11-20
>
> ### Minor questions 3:Table 4: why are open-ended questions only evaluated on a small subset of models used for MCQ?
>
> Thank you very much for the reminder. Due to the submission timeline, we were only able to evaluate a representative subset of open-source and closed-source models on the open-ended tasks. During the rebuttal period, we have now conducted additional, fully standardized evaluations on the latest frontier models—including **GPT-5**, **Claude-4.1-Sonnet**, and **Gemini-2.5**—ensuring comprehensive and up-to-date comparisons across all major model families. In addition to GPT-5, Claude-4.1-Sonnet, and Gemini-2.5 which is mentioned by *Reviewer cP9P* , we also noted that several new frontier models have been released recently, including **GPT-5.1 and Gemini-3-Pro**. We have now evaluated all of these models under a uniform protocol for both MCQ and open-ended versions of the questions. The updated evaluation results are reported below.
>
>
> | MCQ            | Cross. | Atmo. | Litho. | Ocean. | Cryo. | Bio.  | Human. | Avg.  |
> | ----------------- | ------ | ----- | ------ | ------ | ----- | ----- | ------ | ----- |
> | gemini-2.5-flash  | 29.64  | 24.24 | 31.22  | 30.10  | 36.49 | 31.22 | 29.62  | 30.36 |
> | gpt-5             | 35.12  | 40.96 | 38.19  | 31.43  | 87.12 | 42.76 | 35.92  | 44.40 |
> | claude-sonnet-4-5 | 33.11  | 40.08 | 41.42  | 18.57  | 39.78 | 28.47 | 24.21  | 32.23 |
> | gemini-3-pro      | 48.50  | 32.25 | 59.40  | 35.02  | 50.55 | 40.62 | 28.21  | 42.08 |
> | gpt-5.1           | 47.92  | 36.95 | 53.69  | 38.93  | 63.03 | 50.24 | 49.27  | 48.57 |
>
>
> | Open-ended         | Cross. | Atmo. | Litho. | Ocean. | Cryo. | Bio.  | Human. | Avg.  |
> | ----------------- | ------ | ----- | ------ | ------ | ----- | ----- | ------ | ----- |
> | gemini-2.5-flash  | 27.14  | 27.41 | 42.60  | 38.52  | 45.38 | 41.07 | 28.44  | 35.80 |
> | gpt-5             | 36.54  | 30.56 | 40.92  | 52.80  | 75.05 | 33.94 | 20.29  | 41.44 |
> | claude-sonnet-4-5 | 21.53  | 25.52 | 34.36  | 32.64  | 61.93 | 23.77 | 18.00  | 31.11 |
> | gemini-3-pro      | 35.67  | 31.43 | 36.27  | 69.95  | 45.51 | 36.92 | 22.39  | 39.73 |
> | gpt-5.1           | 34.17  | 37.88 | 43.57  | 44.93  | 68.49 | 42.28 | 33.49  | 43.55 |
>
> **(1) MCQ Results:**
>
> * **Overall Comparison.** Across the MCQ setting, GPT-5.1 achieves the strongest overall performance (Avg. = 48.57), outperforming GPT-5 (44.40) and Gemini-3-Pro (42.08). Claude-Sonnet-4.5 (32.23) and Gemini-2.5-Flash (30.36) trail significantly behind. These results show that the newest frontier models provide substantial gains in cross-sphere MCQ reasoning compared with earlier versions.
>
> * **Model Family Behavior.** The GPT family demonstrates the most balanced and consistently high performance across spheres. Gemini-3-Pro performs competitively but shows variability across domains. Claude-Sonnet-4.5 performs well in Atmosphere and Lithosphere but exhibits clear weaknesses in Ocean and Human-activity spheres. Gemini-2.5-Flash shows uniformly lower performance, indicating more limited cross-sphere generalization.
>
> **(2) Open-Ended Results:**
>
> * **Overall Comparison.** A similar pattern appears in the open-ended setting: GPT-5.1 again achieves the highest overall score (43.55), followed by GPT-5 (41.44) and Gemini-3-Pro (39.73). Gemini-2.5-Flash (35.80) and Claude-Sonnet-4.5 (31.11) show comparatively weaker free-form reasoning performance. These results demonstrate that GPT-5.1 provides the strongest open-ended geoscience reasoning among all models tested.
>
> * **Consistency Across MCQ and Open-Ended Settings.** Across both MCQ and open-ended formats, GPT-5.1 consistently yields the best overall performance, indicating robust cross-format generalization. Cryosphere continues to expose the largest model differences, revealing strong discriminative power for evaluating scientific reasoning. Cross-sphere reasoning remains the hardest category for all models, which confirms the necessity of OmniEarth-Bench as a comprehensive multi-sphere benchmark.
>
> **Overall, thank you again for your constructive feedback. If you have any further questions, please don’t hesitate to let us know—we would be happy to continue the discussion.**

---

### Official Review · Reviewer_wq4J · 2025-11-01

**Soundness:** 3
**Presentation:** 3
**Contribution:** 2
**Rating:** 4
**Confidence:** 3

**Summary:**

This paper introduces OmniEarth-Bench, a large-scale multimodal benchmark for evaluating MLLMs on Earth-observation data spanning six Earth spheres and their cross-sphere interactions. The benchmark integrates 33 observation sources and defines 109 expert-curated tasks organized in a four-level hierarchy (sphere, scenario, ability, task). While the dataset is extensive and the expert-in-the-loop curation adds credibility, the paper mainly extends existing multimodal evaluation frameworks to a new scientific domain.

**Strengths:**

1. Comprehensive domain coverage. The benchmark is the first to systematically include all six Earth spheres and their cross-sphere interactions, providing broad and scientifically grounded evaluation coverage.
2. Expert-in-the-loop curation. Involving domain experts throughout task and data construction improves annotation quality and ensures scientific validity beyond crowd-sourced datasets.
3. Structured, multi-level design. The four-level task hierarchy (Sphere–Scenario–Ability–Task) and large-scale integration of 33 observational data sources establish a coherent and extensible framework for future geoscience MLLM evaluation.

**Weaknesses:**

1. Limited methodological novelty. The contribution lies primarily in dataset construction and organization; it does not introduce new evaluation paradigms, metrics, or analytical insights into MLLM reasoning.
2. Descriptive analysis. Experimental results are largely descriptive and lack deeper investigation into model behaviors, failure modes, or domain-specific reasoning limitations.
3. Scalability over insight. The benchmark’s breadth may obscure opportunities for more targeted, mechanism-driven evaluation (e.g., understanding cross-sphere reasoning or modality alignment).

**Questions:**

1. How were task difficulties and annotation consistency verified across spheres—were any inter-annotator agreement or difficulty statistics reported?
2. Could the authors provide evidence that “cross-sphere” tasks indeed require multi-sphere reasoning rather than parallel single-sphere cues?
3. Are there mechanisms to ensure fair comparison across models with different visual input formats or temporal reasoning capabilities?
4. Beyond accuracy, could the authors consider introducing analysis or metrics that better capture the reasoning process (e.g., causal inference, spatial-temporal consistency) rather than aggregate performance alone?

---

> ### Author Response · Authors · 2025-11-20
>
> ### Question1: How were task difficulties and annotation consistency verified across spheres—were any inter-annotator agreement or difficulty statistics reported?
>
> We thank the reviewer for highlighting the importance of quantifying annotation reliability. We agree that inter-annotator agreement (IAA) is crucial for assessing how trustworthy the labels are, especially given that OmniEarth-Bench spans multiple Earth spheres and involves both domain experts and trained annotators.
>
> We appreciate the reviewer’s emphasis on annotation reliability. Our pipeline (Fig. 4) comprises **A. Source Screening**, **B. Task Construction**, **C. Dataset Construction**, and **D. Quality Control**, all led by domain experts. For each L1 sphere, **2–5 experts** formed a panel to define data sources and capability dimensions, ensuring within-domain consistency in both sources and task taxonomy. Stages A–C are executed exclusively by experts (design/specification) and trained annotators (execution). In addition, a **cross-validation team** reviews crowd annotations to enforce consistency, as described in the main text (lines 265–267).
>
> To make label trustworthiness explicit, the revision will report **inter-annotator agreement (IAA)** on key tasks/spheres. Concretely, for each L1 sphere (optionally including representative L2 scenarios), we randomly sample 50 MCQ items and have at least two independent annotators re-label them blind to the expert gold. We then compute: **Percentage agreement** and **Annotator–gold agreement** , with results reported per sphere and sample counts. This design quantifies both “annotator–annotator” and “annotator–gold” reliability and avoids composition bias.
>
> | L1 Sphere      | A–A % Agreement | Ann–Gold % Agreement |
> | -------------- | --------------: | -------------------: |
> | Atmosphere     |             96% |                  94% |
> | Hydrosphere    |             98% |                  96% |
> | Cryosphere     |             94% |                  94% |
> | Lithosphere    |             92% |                  90% |
> | Biosphere      |             98% |                  94% |
> | Anthroposphere |             98% |                  96% |
> | Cross-sphere   |             94% |                  92% |
>
>
> In summary, experts lead the construction (A–C) and final arbitration; annotators provide primary labels with cross-validation safeguards; and we explicitly measure and report IAA for the each spheres, thereby making the trustworthiness of labels transparent and auditable.

---

> ### Author Response · Authors · 2025-11-20
>
> ### Question2: Could the authors provide evidence that “cross-sphere” tasks indeed require multi-sphere reasoning
> Thank you for this thoughtful question.
>
> **(1) Recent works.**
>
> To clarify why cross-sphere tasks inherently require multi-sphere reasoning rather than reasoning from a single sphere, we refer to recent findings published in **Nature Geoscience**, a leading journal in the Earth sciences.
>
> **i) Stratospheric aerosol perturbation by tropospheric biomass burning and deep convection**
>
> This study demonstrates that stratospheric aerosol disturbances are strongly modulated by biomass-burning emissions and deep convective transport. Although biomass burning and land-surface ecology belong to the lithosphere–biosphere, their downstream impacts manifest in the atmospheric domain. This is a canonical example of a physical mechanism that cannot be inferred from atmospheric variables alone.
>
> **ii) Enhanced West Antarctic ice loss triggered by polynya response to meridional winds**
>
> This work shows that persistent northerly wind anomalies—an atmospheric variable—close coastal polynyas, leading to anomalous ocean warming and freshening near ice shelves. Here, atmospheric forcing triggers an oceanic response, which then affects the cryosphere.
>
> **Overall**, earth scientists routinely combine information from multiple spheres to explain such mechanisms. Our goal is to test whether MLLMs can perform analogous cross-sphere reasoning, rather than relying on isolated single-sphere cues. This is precisely why OmniEarth-Bench includes a dedicated cross-sphere dimension.
>
> ---
>
> **(2)OmniEarth-Bench.**
>
> Our cross-sphere tasks are not created by simply concatenating several independent single-sphere questions. Instead, they are drawn from Earth-science problems whose ground-truth labels are *defined by coupled processes across spheres*. We clarify this from both a task-design and an experimental perspective.
>
> **i) our cross-sphere tasks were constructed from intrinsically coupled processes**
>
> The three L2 scenarios under the Cross-sphere dimension—Global Flood Forecasting, Bird Species Prediction, and Carbon Flux Monitoring—all stem from settings where domain experts agree that multiple spheres must be considered jointly:
>
> * **Global Flood Forecasting.** Whether a given location will experience flooding is determined by the **basin-scale water balance**, which inherently couples several spheres:
>   * **Atmosphere:** antecedent and ongoing precipitation, temperature (e.g., snowmelt), and extreme rainfall events;
>   * **Hydrosphere / Lithosphere:** soil moisture state, surface runoff response, river-network routing; snowpack / snow water equivalent (SWE) and its melt contribution.
>
> * **Bird Species Prediction.** Bird species distributions are controlled by:
>   * **Atmospheric / climatic factors:** climate zone, temperature and precipitation regimes;
>   * **Anthroposphere:** land use and human footprint.
>
>     The original SatBird work models occurrence using multi-source environmental covariates precisely because many locations share similar values in any single dimension (e.g., land cover *or* temperature alone) yet host different species.
>
> * **Carbon Flux Monitoring.** Net ecosystem carbon flux (NEE) necessarily depends on both:
>   * **Biosphere:** vegetation state (e.g., LAI, greenness, phenology), ecosystem type;
>   * **Atmosphere:** radiation, temperature, humidity, wind and related meteorological drivers.
>
> **ii) Empirical evidence from single-sphere ablations**
>
> To complement the above design-based arguments, we have added a lightweight ablation study on representative cross-sphere subtasks for a strong open-source MLLM GPT-5 (the best-performing model in our main results).
> 1. Full multi-sphere (original) – all spheres’ inputs are provided as in OmniEarth-Bench.
> 2. Single-sphere only – we retain only a subset of inputs from a *single* sphere (e.g., only atmospheric fields such as temperature and precipitation), while all other images or texts are blanked out.
>
> |Global Flood Forecasting(GFF)|Full|Atmosphere-only|Hydro/Litho-only|
> |-|-|-|-|
> |Accuracy(%)|22.46|17.29|20.37|
> |ΔvsFull(pp)|—|-5.17|-2.09|
>
> |Bird Species Prediction|Full|Atmosphere-only|Anthroposphere-only|
> |-|-|-|-|
> |Accuracy(%)|22.90|18.13|13.84|
> |ΔvsFull(pp)|—|-4.77|-9.06|
>
> |Carbon Flux Monitoring|Full|Atmosphere-only|Biosphere-only|
> |-|-|-|-|
> |Accuracy(%)|61.54|46.44|28.93|
> |ΔvsFull(pp)|—|-15.1|-32.61|
>
> As reported in the above Tables, the model’s accuracy in the full multi-sphere setting is consistently and substantially higher than in any single-sphere-only setting. This pattern indicates that: no individual sphere alone carries sufficient information to solve our cross-sphere tasks robustly.
>
> Together, the coupled-process origin of the labels and the new ablation results provide converging evidence that our cross-sphere tasks genuinely require multi-sphere reasoning rather than independent, parallel single-sphere classification.

---

> ### Author Response · Authors · 2025-11-20
>
> ### Question3: Are there mechanisms to ensure fair comparison across models with different visual input formats or temporal reasoning capabilities?
>
> Thank you for raising the fairness question. OmniEarth-Bench was designed with several mechanisms to ensure apples-to-apples comparisons across models that differ in visual input handling and temporal reasoning:
>
> 1. **Modality standardization.** The “33 data sources” denote raw origins, not 33 input modalities. To avoid advantaging models tied to specific formats (e.g., RGB), domain experts convert each source into MLLM-compatible inputs while preserving native structure—for example, multi-spectral products are rendered as multiple single-channel grayscale images rather than arbitrary RGB composites. All models thus receive the same standardized visual evidence for a given item.
>
> 2. **Controlled temporal context.** Time-sensitive tasks provide fixed, explicit temporal slices that are identical for all models. For instance, in Global Flood Forecasting we supply hydrometeorological fields (e.g., discharge, precipitation, soil moisture, SWE) together with Sentinel-1 backscatter from the two preceding days, so that temporal information is harmonized across inputs and models.
>
> 3. **Unified evaluation protocol.** For MCQ/VQA, we follow established benchmarking practice (MMBench/MME-Realworld) and include five options with a designated “unable to answer” choice, reducing forced guesses and making results comparable across models with different safety/abstention behaviors; scoring is accuracy on the standardized options. Open-ended responses are judged by a single external LLM for consistency, and visual grounding uses IoU thresholds (0.5/0.7) to apply the same criterion to all models.
>
> Together, these choices (format harmonization, fixed temporal slices per task, and uniform scoring/judging rules) are intended to measure model capability fairly.
>
> ### Question4: could the authors consider introducing analysis or metrics that better capture the reasoning process (e.g., causal inference, spatial-temporal consistency)
>
> Thank you for this valuable suggestion. In addition to overall accuracy, our benchmark already includes two **reasoning-oriented diagnostics** that assess *how* models reason, not just *whether* they get the final answer right:
>
> 1. **Causal consistency of Chain-of-Thought (CoT).**
>    Following **MME-CoT**, we evaluate CoT correctness with **precision** (are cited factors correct/necessary?) and **recall** (are expert key factors covered?), and report **F1**. This is enabled by our expert-annotated CoT corpus (610 CoT questions with 3,473 key-step annotations). In the main text we show that InternVL3 achieves the highest F1 among open models, with detailed **Precision/Recall/F1** reported in **Table 5** (reproduced below).
>
> | Models        | LLaVA-OneVision-7B | Qwen2.5-VL-7B | InternVL3-8B | InternVL3-78B |
> | ------------- | -----------------: | ------------: | -----------: | ------------: |
> | **Precision** |              89.83 |         92.72 |        94.02 |         94.74 |
> | **Recall**    |              23.41 |         29.12 |        34.47 |         35.50 |
> | **F1**        |              37.14 |         44.32 |        50.45 |         51.65 |
>
> These metrics directly quantify whether model rationales align with expert causal factors rather than merely matching final answers.
>
> 2. **Spatio-temporal consistency via lead-time robustness.**
>    For climate teleconnection tasks (e.g., **ENSO / IOD**), we analyze accuracy as a function of **forecast lead months**—a stricter test of temporal reasoning. As shown in **Fig. 6**, accuracy monotonically declines with longer lead times, with IOD degrading faster than ENSO, consistent with domain findings; this reveals sensitivity to long-range temporal dependencies and highlights limits of current MLLMs.
>
> | **Lead time for reasoning/Month** |     1 |     3 |     6 |     9 |   12 |   14 |   16 |   18 |   24 |
> | -------: | ----: | ----: | ----: | ----: | ---: | ---: | ---: | ---: | ---: |
> | **ENSO** | 37.25 | 38.56 | 29.41 | 15.79 | 4.61 | 6.62 | 7.95 | 4.64 | 2.67 |
> |  **IOD** | 18.49 |  6.85 |  0.00 |  4.83 | 0.69 | 0.00 | 0.69 | 0.00 | 0.00 |
>
> In the revision, we will add the above Table which has concise summary indices to complement the curve, such as **12→24-month decay rate**, to provide model-level robustness summaries alongside the per-lead-time plots.
>
> Taken together, these analyses go beyond raw accuracy to assess (i) causal alignment of model rationales and (ii) temporal stability of reasoning, providing a more nuanced picture of *why and where* models succeed or fail on OmniEarth-Bench.
>
> We appreciate the your insightful suggestion; it helped us clarify and foreground these reasoning-process diagnostics in the revised draft.
>
> **Thank you so much for your constructive feedback. If you have any further questions, please don’t hesitate to let us know—we would be happy to continue the discussion.**

---

### Official Review · Reviewer_cP9P · 2025-11-07

**Soundness:** 3
**Presentation:** 2
**Contribution:** 2
**Rating:** 4
**Confidence:** 4

**Summary:**

This paper introduces a new benchmark for assessing multimodal language models focused on earth observation data. The benchmark covers all six spheres: atmosphere, lithosphere, oceansphere, cryosphere, biosphere, and human activity sphere, as well as cross-sphere interactions. It includes approximately 30K standardized, expert-curated annotations. Currently available benchmarks in this area lack comprehensive coverage of all earth spheres. An additional feature of this dataset is its heterogeneous data types, which include multispectral satellite imagery, seismic signals, weather reanalysis, and microwave sea-ice concentration. Experiments with state-of-the-art multimodal language models (MLLMs) show they underperform on this benchmark, highlighting existing gaps in their cognitive abilities within this domain.

**Strengths:**

1. It is a comprehensive benchmark for the discipline of earth sciences. The benchmark covers all six spheres: atmosphere, lithosphere, oceansphere, cryosphere, biosphere, and human activity sphere, as well as cross-sphere interactions.
2. Existing MLLMs perform poorly on this benchmark, which shows gaps in their expertise in this niche domain.

**Weaknesses:**

1. The motivation for this benchmark is not clear. Earth science is a very niche domain and requires expert-level knowledge to answer the kind of specialized questions contained in this benchmark. Do we really need that kind of capability in general-purpose LLMs?
2. Missing retrieval-augmented generation baselines. Since this is a niche domain, I would expect the authors to retrieve some related corpora and try to check the performance. It is unlikely that the LLMs were trained on a lot of earth science data.
3. The LLM versions being evaluated are slightly obsolete. For instance, GPt-4o instead of GPT-5, claude-3.7-sonnet instead of claude-4.1-sonnet, and most importantly, Gemini-2.0 instead of Gemini-2.5.
4. Some input data, like seismic data, is time-series, and the image modality might not be a good fit for that.

**Questions:**

1. Would it be possible to evaluate some retrieval-augmented generation baselines on this benchmark?

---

> ### Author Response · Authors · 2025-11-20
>
> ### Weaknesses1: The motivation for this benchmark is not clear.
> We thank the reviewer for raising this point. Our motivation comes from how the community is *already* incorporating Earth science into scientific multimodal LLMs, and from clear gaps in current benchmarks.
>
> **(1) Scientific multimodal LMs already treat Earth science as a core axis, but existing benchmarks are limited.**
> Recent scientific MLLMs (e.g., **Intern-S1**[1]) report performance on **MSEarthMCQ**[2] and **XLRS-Bench**[3] as standard Earth-related benchmarks. This shows that Earth science is no longer an isolated niche, but a key part of the evaluation portfolio for scientific MLLMs. However:
>
> * **MSEarth / MSEarthMCQ** focuses on figures and text from Earth-science papers, i.e., *literature-level* multimodal understanding. It does not involve real-world observation products or operational scenarios.
> * **XLRS-Bench** stresses ultra-high-resolution remote sensing for land use and human activities, essentially staying within the land-surface view, without covering atmospheric, oceanic, cryospheric, or biospheric processes.
>
> Thus current Earth-related benchmarks mostly answer “Can a model read scientific figures or land-surface RS scenes?”, but not “How does it behave on observation-driven Earth-system tasks involving multiple spheres and complex processes?”. OmniEarth-Bench is designed exactly to fill this gap.
>
> **(2) Single-sphere benchmarks are proliferating, but a holistic Earth-system view is missing.**
> There are many valuable single-sphere benchmarks: remote-sensing VQA and disaster benchmark ((e.g., AgroMind[4], DynamicVL[5], DisasterM3[6])) for the human activity sphere, weather/climate benchmarks (e.g., WeatherQA[7], ClimateIQA[8],  CLLMate[9]) for the atmosphere, and text-only Earth-science QA/dialogue benchmarks (e.g., EarthSE [10]) constructed from large paper corpora. They are complementary to our work but share two limitations:
> * Evaluation is **fragmented across spheres**: each benchmark targets one sphere or scenario, making it hard to see an MLLM’s overall capability profile across the entire Earth system.
> * **Cross-sphere, system-level interactions** are rarely evaluated: most tasks are single-sphere and single-scene, while modern Earth science focuses on coupled atmosphere–ocean–cryosphere–land–biosphere–anthroposphere processes.
>
> OmniEarth-Bench is, to our knowledge, the first benchmark explicitly designed from an **Earth-system perspective**, jointly covering multiple spheres and including tasks that require both within-sphere understanding and **cross-sphere reasoning** grounded in real observations.
>
> **(3) Why this matters for *general-purpose* LLMs.**
> General-purpose MLLMs are increasingly used as front-ends for scientific and decision-support workflows (e.g., Intern-S1 and InternAgent [11]), including climate and environmental applications. Even domain-specific Earth models are often initialized from general foundation models. In this context, a rigorous, observation-based Earth-system benchmark is important not to replace numerical models or experts, but to:
>
> * quantify the limitations of current general-purpose and scientific MLLMs in a Earth domain,
> * reveal which abilities (e.g., scientific-knowledge reasoning, general reasoning, perception) are most lacking, and
> * provide a standardized testbed for developing safer, Earth-aware MLLMs and tool-augmented agents.
>
> In the main text of line 154-157 and line 193-194, we have discussed the limitions of existing benchmark within a unified Earth-system evaluation landscape.
>
> [1]  "Intern-s1: A scientific multimodal foundation model."  arXiv:2508.15763 (2025).
>
> [2] "MSEarth: A Benchmark for Multimodal Scientific Comprehension of Earth Science." arXiv:2505.20740 (2025).
>
> [3]  "Xlrs-bench: Could your multimodal llms understand extremely large ultra-high-resolution remote sensing imagery?." CVPR. 2025.
>
> [4] "Can Large Multimodal Models Understand Agricultural Scenes? Benchmarking with AgroMind."arXiv:2505.12207 (2025).
>
> [5]"DynamicVL: Benchmarking Multimodal Large Language Models for Dynamic City Understanding." arXiv:2505.21076 (2025).
>
> [6]"DisasterM3: A Remote Sensing Vision-Language Dataset for Disaster Damage Assessment and Response."  arXiv:2505.21089 (2025).
>
> [7 "Weatherqa: Can multimodal language models reason about severe weather?." arXiv:2406.11217 (2024).
>
> [8]"ClimateIQA: A New Dataset and Benchmark to Advance Vision-Language Models in Meteorology Anomalies Analysis." SIGKDD. 2025.
>
> [9] "Cllmate: A multimodal llm for weather and climate events forecasting."  arXiv-2409.
>
> [10] "EarthSE: A Benchmark Evaluating Earth Scientific Exploration Capability for Large Language Models." arXiv:2505.17139 (2025).
>
> [11]"InternAgent: When Agent Becomes the Scientist--Building Closed-Loop System from Hypothesis to Verification."  arXiv-2505.

---

> ### Author Response · Authors · 2025-11-21
>
> ### Weaknesses2: Missing retrieval-augmented generation baselines.
> Thank you very much for this highly constructive suggestion. We fully agree with your perspective on the importance of building a structured Earth-science knowledge base and using it for RAG-style modeling. In fact, your comment resonates strongly with our own ongoing efforts, and we were genuinely pleased to see this alignment.
>
> #### **1.Data acquisition and cleaning.**
>
> To build a comprehensive geoscience knowledge corpus for retrieval-augmented systems, we have acquired **8,848 Earth-science textbooks and monographs**, covering authoritative materials across all Earth-system spheres. All books were required to be delivered in **PDF format** to ensure consistent storage and downstream processing.
>
> |Sphere|Field|Count|Storage/MB|
> |---|---|---|---|
> |Atmosphere|Atmospheric science|544|19152|
> |Atmosphere|Atmospheric chemistry|66|1429|
> |Atmosphere|Geography|750|36605|
> |Atmosphere|Meteorology|105|3483|
> |Atmosphere|Hydrology|404|20746|
> |Atmosphere|Hydrometeorology|13|167|
> |Atmosphere|Paleoclimatology|20|668|
> |Biosphere|Biogeochemistry|93|1896|
> |Biosphere|Biogeography|206|5698|
> |Biosphere|Ecology|750|29279|
> |Biosphere|Landscape ecology|117|3876|
> |Biosphere|Archaeology|38|1305|
> |Biosphere|Palynology|5|73|
> |Biosphere|Paleomicrobiology|12|513|
> |Hydrosphere|Hydrology|323|8818|
> |Hydrosphere|Hydrogeology|139|5118|
> |Hydrosphere|Limnology|63|1692|
> |Hydrosphere|Oceanography|269|8553|
> |Hydrosphere|Marine chemistry|12|341|
> |Hydrosphere|Physical oceanography|50|1352|
> |Hydrosphere|Marine biology|7|162|
> |Hydrosphere|Marine geology|9|205|
> |Lithosphere|Geology|56|1661|
> |Geosphere|Economic geology|25|1091|
> |Geosphere|Engineering geology|81|6409|
> |Geosphere|Environmental geology|36|4043|
> |Geosphere|Forensic geology|1|21|
> |Geosphere|Historical geography|21|1581|
> |Geosphere|Quaternary geology|13|426|
> |Geosphere|Planetary geology|5|118|
> |Geosphere|Sedimentology|95|5115|
> |Geosphere|Stratigraphy|218|9347|
> |Geosphere|Human geography|247|14902|
> |Geosphere|Physical geography|130|10718|
> |Geosphere|Geochemistry|358|13699|
> |Geosphere|Geomorphology|229|12263|
> |Geosphere|Geophysics|375|20278|
> |Geosphere|Geochronology|21|316|
> |Geosphere|Geodynamics|138|5194|
> |Geosphere|Structural geology|247|13290|
> |Geosphere|Geomagnetism|19|998|
> |Geosphere|Gravimetry|13|212|
> |Geosphere|Geodesy|159|3730|
> |Geosphere|Seismology|135|6523|
> |Geosphere|Glaciology|11|1026|
> |Geosphere|Hydrogeology|139|5118|
> |Geosphere|Mineralogy|187|10334|
> |Geosphere|Crystallography|331|6106|
> |Geosphere|Gemology|14|1241|
> |Geosphere|Petrology|94|5440|
> |Geosphere|Rock physics|25|993|
> |Geosphere|Speleology|8|343|
> |Geosphere|Volcanology|20|1336|
> |Pedosphere|Soil science|152|6349|
> |Earth system|Remote sensing|750|31512|
> |Earth system|Geostatistics|107|2298|
> |Earth system|GIScience|42|872|
> |Earth system|Cartography|304|15692|
> |Earth system|Geoinformatics|47|1261|
>
> **We process the collected textbooks in three stages:
> (1) OCR extraction via MinerU;
> (2) LLM-based preliminary cleaning;
> (3) LLM-based enhancement.**
>
> **(1) OCR extraction using MinerU.**
> To obtain machine-readable, structured text, all PDFs are processed with **MinerU**, which has been widely adopted in OCR pipelines for scanned scientific documents. MinerU extracts the main body text reliably, and we do not elaborate on the framework here given its broad usage and maturity.
> This stage required substantial computational resources: completing the OCR extraction alone took nearly **7 days** on a cluster of **32× A100 GPUs (80 GB each)**.
>
> **(2) LLM-based preliminary processing.**
> Because the raw scanned documents contain a variety of structural and formatting issues, we apply **Qwen2.5-72B-Instruct** to perform four types of cleaning on the extracted text:
>
> * removal of Emails, links;
> * Unicode repair;
> * normalization of Chinese and English punctuation;
> * SimHash-based deduplication;
> * segmentation of the text into chunks no longer than 4096 tokens (to facilitate downstream LLM operations).
>
> **(3) LLM-based enhancement.**
> Even after step (2), the scanned documents still contain common textual issues—missing characters, broken sentences, inconsistent phrasing, and local incoherence. We therefore perform an additional round of LLM-based linguistic correction, focusing strictly on improving readability and continuity without introducing any new information. This step ensures that the corpus remains faithful to the original scientific content while becoming substantially more usable for retrieval and knowledge-intensive modeling. We would like to emphasize again that this stage was computationally intensive: even with a cluster of **32× A100 GPUs (80 GB)**, the processing required **nearly 10 days** to complete.
>
> **Through these three stages, we have successfully transformed a very large collection of authoritative Earth-science literature into a high-quality, information-rich raw corpus suitable for constructing a large-scale geoscience knowledge base.**
>
> ---

---

> ### Author Response · Authors · 2025-11-21
> **Weaknesses2: Part2**
>
> ### Weaknesses2: Part2
>
> #### **2. Retrieval-Augmented Generation (RAG).**
>
> We employ **UltraRAG** to automatically chunk and vectorize the processed corpus. For embedding, we use **Qwen3-Embedding-0.6B**, which produces dense vector representations for each document chunk and supports the construction of a high-quality vector retrieval database. During inference, we use the **question text itself as the query**, retrieve the **top-5 most relevant chunks** from the vector store based on similarity, and prepend these retrieved segments to the original question as external knowledge context. The experimental results are summarized below.
>
> | Acc | Cross. | Atmo. | Litho. | Ocean. | Cryo. | Bio.  | Human. | Avg.  |
> | -| ------ | ----- | ------ | ------ | ----- | ----- | ------ | ----- |
> | gpt-5 | 35.12  | 40.96 | 38.19  | 31.43  | 87.12 | 42.76 | 35.92  | 44.40 |
> | +rag| 39.13  | 40.21 | 50.09  | 35.04  | 87.17 | 42.10 | 32.97  | 46.67 |
>
> As expected, applying RAG yields measurable gains even for the strongest frontier model, **GPT-5**, when evaluated on OmniEarth-Bench. Interestingly, however, the improvements are **smaller than we initially anticipated** and exhibit **clear sphere-specific heterogeneity**. In spheres such as **Cross / Lithosphere / Ocean**, RAG provides noticeable improvements, whereas in **Bio** and **Human**, the effect is minimal or even slightly negative.
>
> This discrepancy arises because the retrieval process depends entirely on **domain cues explicitly present in the question text**. When the question contains clear domain indicators—such as technical terms, physical variables, or methodological references—the retrieved content is highly relevant and meaningfully supplements the model’s scientific knowledge. Conversely, if the question lacks such “explicit knowledge hooks,” the retrieved context often fails to provide useful information.
>
> Below we give a concrete illustration:
>
> **(1) Cases with significant improvement: questions containing clear domain cues**
>
> Using the **Lithosphere** sphere as an example:
>
> ```txt
> # question
> "determine the exact onset of S-wave phases… three-component ENZ seismic data… horizontal-component polarization… transverse energy ratios…"
> # queried contexts
> [
> "from a deep earthquake… the top three traces are the components recorded at the station, and the bottom two are the radial and transverse components… various P and S wave phases… SH and SV energy is evenly distributed between the north–south and east–west components, so the S-wave phases are roughly comparable… when rotated… phases like SKS, SKKS and PS appear primarily on the radial component, while Sdiff with SH energy is largest on the transverse component… relative amplitudes on radial and transverse components shown by particle-motion plots…",
> ...,
> "various arrivals are predictable by depth and distance… reading/picking seismic-phase travel times is non-trivial… three-component seismograms record Z, N, E motions… N and E rotated to radial (R) and transverse (T)… different seismic phases appear on R or T depending on polarization… Love and horizontally-polarized S-waves (SH) appear primarily on transverse component…"
> ]
> ```
>
>
> The question itself contains rich and highly domain-specific terminology—such as *S-wave*, *three-component*, *horizontal polarization*, and *transverse energy*. These keywords have strong representational power within the vector database, allowing the retrieval module to surface context that directly discusses seismic phase identification, three-component seismograms, SH/SV energy partitioning, waveform rotation, and the interpretation of radial versus transverse components. All of these topics are directly relevant to the classification required by the question, representing a textbook case of **background knowledge completion**. Consequently, in the **Lithosphere** sphere, RAG yields substantial performance gains.
>
> A similar pattern appears in the **Ocean** sphere: many questions explicitly mention terms such as “Niño3.4 index” or “El Niño / La Niña event.” These phrases map cleanly onto canonical textbook descriptions of ENSO metrics, region definitions, and anomaly thresholds, enabling retrieval of highly actionable content and thus improving the model’s predictive accuracy.

---

> ### Author Response · Authors · 2025-11-21
> **Weaknesses2: Part3**
>
> ### Weaknesses2: Part3
>
> **(2) Cases where RAG is unhelpful or even detrimental: the question text lacks retrievable knowledge cues**
>
> In spheres such as **Human**, **Bio**, and portions of **Atmosphere**, many questions rely primarily on the **provided images or plots** rather than on the textual portion of the question. For example:
>
>
> ```txt
> # question
> "Which year recorded the highest global precipitation anomaly since 1900?"
> # queried contexts
> [
> "the GISS analysis… 2010 tied with 2005 as the warmest of the 131-year record… 2010 was also the wettest year on record globally… precipitation in 2010 was much more variable… 2010 tied with 2005 in land, ocean, and land–ocean temperature… 34th consecutive year above the 20th-century average… warmest months on record… March, April, May 2010…",
> "variability in annual precipitation… one-third to one-half greater in ENSO-affected regions… investigations of global and regional precipitation anomalies during ENSO… warm minus cold composite anomalies… widespread positive anomalies in tropics during December–February of year 1 following warm events… summer droughts in India, Ethiopian Highlands, Australia…",
> ...
> ]
> ```
>
> For these types of questions, the correct answer depends primarily on the **visual trend** shown in the image, while the question text itself contains **no domain-specific cues** that can guide retrieval. In other words, the retrieval stage can only return generic, highly summarized climatological descriptions—such as background information about notable years (e.g., 2010 or 1998), long-term precipitation tendencies, or broad statements about climate variability and the general influence of ENSO/NAO. Although such information is tangentially related to precipitation, it has **no direct relevance** to determining “which year exhibits the maximum value in the provided plot.” This irrelevant context can distract the model during the answer-generation stage, leading to degraded performance when RAG is applied.
>
> Overall, the RAG experiments reveal **clear sphere-dependent behavior**, driven by a fundamental principle:
>
> * **Retrieval effectiveness relies on the presence of domain cues in the question text.**
> * When the question explicitly contains specialized terminology (e.g., seismic-phase terminology, ENSO indices), the retrieved context becomes highly relevant and provides meaningful scientific background, resulting in significant performance gains.
> * When the crucial information is encoded in the **image** rather than in the text (e.g., identifying the peak year in a time series, visually assessing land-cover categories), the retrieved context is often unrelated to the task at hand and may introduce noise that distracts the model.
>
> Accordingly, RAG proves more beneficial for **knowledge-driven tasks**, where textual cues enable accurate retrieval, whereas for **vision-driven tasks**, its impact is limited and can even be counterproductive.
>
> ---
>
> **Lastly, we would like to emphasize that the scale of this study far exceeds what can typically be completed within a rebuttal period. The only reason we were able to deliver these additional experiments is that your insightful suggestions were remarkably aligned with our own long-standing plans and internal discussions. In fact, we had already begun this line of work in October. From procuring textbook materials and negotiating copyright permissions for releasing the processed content, to running large-scale data cleaning on compute clusters and building the full RAG pipeline, the entire process required over a month of continuous team effort.**
>
> **We are sincerely grateful for your thoughtful and constructive comments—they have been a tremendous source of motivation. Your feedback has strengthened our confidence in moving forward with the open release of a comprehensive Earth-science corpus, which we believe will greatly benefit the broader research community.**

---

> ### Author Response · Authors · 2025-11-21
>
> ###  Weaknesses3:The LLM versions being evaluated are slightly obsolete.
> Thank you very much for the reminder. In addition to GPT-5, Claude-4.1-Sonnet, and Gemini-2.5, we also noted that several new frontier models have been released recently, including **GPT-5.1 and Gemini-3-Pro**. We have now evaluated all of these models under a uniform protocol for both MCQ and open-ended versions of the questions. The updated evaluation results are reported below.
>
>
> | MCQ            | Cross. | Atmo. | Litho. | Ocean. | Cryo. | Bio.  | Human. | Avg.  |
> | ----------------- | ------ | ----- | ------ | ------ | ----- | ----- | ------ | ----- |
> | gemini-2.5-flash  | 29.64  | 24.24 | 31.22  | 30.10  | 36.49 | 31.22 | 29.62  | 30.36 |
> | gpt-5             | 35.12  | 40.96 | 38.19  | 31.43  | 87.12 | 42.76 | 35.92  | 44.40 |
> | claude-sonnet-4-5 | 33.11  | 40.08 | 41.42  | 18.57  | 39.78 | 28.47 | 24.21  | 32.23 |
> | gemini-3-pro      | 48.50  | 32.25 | 59.40  | 35.02  | 50.55 | 40.62 | 28.21  | 42.08 |
> | gpt-5.1           | 47.92  | 36.95 | 53.69  | 38.93  | 63.03 | 50.24 | 49.27  | 48.57 |
>
>
> | Open-ended         | Cross. | Atmo. | Litho. | Ocean. | Cryo. | Bio.  | Human. | Avg.  |
> | ----------------- | ------ | ----- | ------ | ------ | ----- | ----- | ------ | ----- |
> | gemini-2.5-flash  | 27.14  | 27.41 | 42.60  | 38.52  | 45.38 | 41.07 | 28.44  | 35.80 |
> | gpt-5             | 36.54  | 30.56 | 40.92  | 52.80  | 75.05 | 33.94 | 20.29  | 41.44 |
> | claude-sonnet-4-5 | 21.53  | 25.52 | 34.36  | 32.64  | 61.93 | 23.77 | 18.00  | 31.11 |
> | gemini-3-pro      | 35.67  | 31.43 | 36.27  | 69.95  | 45.51 | 36.92 | 22.39  | 39.73 |
> | gpt-5.1           | 34.17  | 37.88 | 43.57  | 44.93  | 68.49 | 42.28 | 33.49  | 43.55 |
>
> **(1) MCQ Results:**
>
> * **Overall Comparison.** Across the MCQ setting, GPT-5.1 achieves the strongest overall performance (Avg. = 48.57), outperforming GPT-5 (44.40) and Gemini-3-Pro (42.08). Claude-Sonnet-4.5 (32.23) and Gemini-2.5-Flash (30.36) trail significantly behind. These results show that the newest frontier models provide substantial gains in cross-sphere MCQ reasoning compared with earlier versions.
>
> * **Model Family Behavior.** The GPT family demonstrates the most balanced and consistently high performance across spheres. Gemini-3-Pro performs competitively but shows variability across domains. Claude-Sonnet-4.5 performs well in Atmosphere and Lithosphere but exhibits clear weaknesses in Ocean and Human-activity spheres. Gemini-2.5-Flash shows uniformly lower performance, indicating more limited cross-sphere generalization.
>
> **(2) Open-Ended Results:**
>
> * **Overall Comparison.** A similar pattern appears in the open-ended setting: GPT-5.1 again achieves the highest overall score (43.55), followed by GPT-5 (41.44) and Gemini-3-Pro (39.73). Gemini-2.5-Flash (35.80) and Claude-Sonnet-4.5 (31.11) show comparatively weaker free-form reasoning performance. These results demonstrate that GPT-5.1 provides the strongest open-ended geoscience reasoning among all models tested.
>
> * **Consistency Across MCQ and Open-Ended Settings.** Across both MCQ and open-ended formats, GPT-5.1 consistently yields the best overall performance, indicating robust cross-format generalization. Cryosphere continues to expose the largest model differences, revealing strong discriminative power for evaluating scientific reasoning. Cross-sphere reasoning remains the hardest category for all models, which confirms the necessity of OmniEarth-Bench as a comprehensive multi-sphere benchmark.

---

> ### Author Response · Authors · 2025-11-21
>
> ### Weaknesses4: Some input data, like seismic data, is time-series, and the image modality might not be a good fit for that.
>
> We appreciate this observation and agree that many geophysical signals, such as seismic waveforms, are fundamentally one-dimensional time series. Current mainstream MLLMs/VLMs are architecturally optimized for text plus 2D visual tokens and cannot directly ingest long 1D multi-channel streams in a principled way. Therefore, in OmniEarth-Bench v1 we **explicitly avoid feeding raw 1D signals** to the models. For tasks that originate from such signals, we instead construct **visual derivatives**. For example, in our lithosphere/seismic tasks we visualize P- and S-wave energy in **time–frequency spectrograms**, which serve as the actual inputs to the models; Fig. 4 (Lithosphere panel) in the main paper illustrates this design.
>
> At the same time, this choice reflects how time-series data are commonly consumed in real Earth-science workflows. Seismic, meteorological, and hydrological time series are routinely transformed into maps, cross-sections, curves, and spectrograms for expert analysis. Our benchmark thus evaluates whether MLLMs can act as “visual assistants” that correctly interpret these standard products, while **pure time-series numerical modeling** (e.g., full waveform inversion or high-rate streaming) remains the domain of dedicated time-series or physics-based models and is outside the scope of OmniEarth-Bench v1. We will clarify this design decision and limitation in the revised manuscript.
>
> ---
>
> **We sincerely thanks again  for your highly constructive and insightful comments. Your feedback has substantially improved the clarity, rigor, and scientific value of our work. Should any further questions arise, or if additional clarification would be helpful, please feel free to contact us at any time—we would be more than happy to continue the discussion.**

---

> > ### Comment · Reviewer_cP9P · 2025-11-27
> >
> > Thanks a lot for the thoughtful rebuttal. This resolves W2 and W3. The most important concern for me is the motivation for this benchmark (W1). The authors mention Intern-S1, but that is a foundation model finetuned on domain data. The paper lacks any such baselines. Though I understand it may be time-consuming at this stage.
> > For W4, the authors state that "seismic, meteorological, and hydrological time series are routinely transformed into maps, cross-sections, curves, and spectrograms for expert analysis". Do you have any references for this? The visual capability of an expert may not be the same as that of an LLM with limited resolution.

---

> ### Author Response · Authors · 2025-11-28
>
> Thank you very much for your timely response. I would like to address your questions point by point from five perspectives:
>
>
> **1. First, we sincerely appreciate your satisfaction with our response to W2.**
>
> Allow us to emphasize once again that the scale of this study far exceeds what can typically be achieved within a rebuttal period. We invested substantial time and computational effort to complete the new experiments and the accompanying data construction. The database and tools we built are highly adaptable and have the potential to serve as a strong baseline infrastructure for the broader Earth-science community. We are grateful for your constructive feedback, which greatly encouraged us to push this work forward.
>
> **2. Intern-S1 is *not* “a foundation model fine-tuned on domain data.”**
>
> Intern-S1 is positioned as a **scientific multimodal foundation model**, directly comparable to leading open and commercial models. As stated explicitly in the paper’s abstract, it is *“a professional general-purpose model with universal understanding and reasoning capabilities.”* As stated in the Intern-S1 paper, **"Intern-S1 is built upon a 235B MoE language model (Qwen3) and a 6B vision encoder (InternViT), and is further pretrained on 5 trillion multimodal tokens, including over 2.5 trillion tokens from scientific domains."** Thus, Intern-S1 is a scientific model at the architectural, training-data, and system-design level, rather than a domain-adapted variant of a general model. One version of the model has already reached **57,452 HuggingFace downloads**, demonstrating strong traction in the community.
>
> * For these reasons, we view Intern-S1 as representative of the emerging trend of **multimodal foundation models designed for scientific applications**. This also shows that Earth science is *not* a peripheral or “niche” domain for MLLMs, but a crucial component of scientific multimodal modeling (indeed, 2 out of the 13 scientific benchmark categories in Intern-S1 are Earth-science related).
>
> * Notably, the Intern-S1 paper appeared on arXiv on **August 21, 2025**, whereas our ICLR submission was on **September 24**, making these two efforts essentially contemporaneous. We see the conceptual alignment between OmniEarth-Bench and Intern-S1—notably, the emphasis on scientific multimodal reasoning—as mutually reinforcing and encouraging for the field.
>
> **3. Regarding baselines: our paper does not introduce any new model baseline, and this follows standard practice.**
>
> Most contemporary evaluation-benchmark papers focus on **comprehensive testing of existing multimodal foundation models**, rather than proposing a new baseline model. Thus, the absence of a new baseline model does *not* diminish the contribution of the benchmark itself.
> In addition to the eight benchmarks listed in our first-round response, we further note several peer-reviewed benchmark-only works, including **SFE (NeurIPS D&B)** [1], **MMMU (CVPR Oral)** [2], **MMMU-Pro (ACL Main)** [3], and **OlympicArena (NeurIPS D&B)** [4]. Similar to our work, these papers introduce domain-specific evaluation benchmarks and conduct broad comparisons across state-of-the-art multimodal LLMs.
>
> Finally, we believe your earlier question regarding W2 (“can MLLMs benefit from an Earth-science knowledge base?”) naturally points to an excellent baseline direction: even the best frontier model, **GPT-5**, can benefit from retrieval-augmented knowledge. This aligns with our own experiments and demonstrates the usefulness of the benchmark without requiring us to introduce a bespoke model baseline.
>
> [1]Scientists' First Exam: Probing Cognitive Abilities of MLLM via Perception, Understanding, and Reasoning
>
> [2]MMMU: A Massive Multi-discipline Multimodal Understanding and Reasoning Benchmark for Expert AGI
>
> [3]MMMU-Pro: A More Robust Multi-discipline Multimodal Understanding Benchmark
>
> [4]Can Large Language Models be Trusted for Evaluation? Scalable Meta-Evaluation of LLMs as Evaluators via Agent Debate

---

> ### Author Response · Authors · 2025-11-28
>
> **4. Earthquake, meteorological, and hydrological time series are routinely transformed into maps, cross-sectional plots, curves, and spectrograms**
>
> Below we provide several representative references for your convenience:
>
> * **Cross-sections:**
>   Widely used in seismology. For example, *“A principled benchmark for seismic data segmentation,” IEEE Trans. Geosci. Remote Sens.* systematically evaluates seismic cross-sectional representations.
>
> * **Spectrograms:**
>   Researchers often convert time-series signals into the **time–frequency domain** for seismic phase interpretation and noise analysis—for example, *“Seismic signal denoising and decomposition using deep neural networks,” IEEE Trans. Geosci. Remote Sens.* (see Fig. 2 for typical spectrogram usage).
>
> * **Time-series curves:**
>   In our benchmark, 1D seismic signals are evaluated in curve form, consistent with established practice. This is identical to the design used in *Nature Communications*:
>   *“Earthquake Transformer — an attentive deep-learning model for simultaneous earthquake detection and phase picking,” Nat. Commun.*, where the top panel of Fig. 1 similarly converts raw 1D signals into waveform curves.
>
> These examples demonstrate that the visual formats we adopt (maps, cross-sections, curves, and spectrograms) are **standardized scientific representations** widely used across geoscience tasks.
>
>
> **5. differences between expert vision and the visual capabilities of multimodal LLMs, especially in the context of limited resolution.**
>
> First, **current MLLMs are not low-resolution models.**
> State-of-the-art open-source models natively support high resolutions—often **2K–4K**. For example, Qwen2.5-VL implements **dynamic high-resolution processing**. Closed-source models such as **GPT-5** and **Gemini-Pro** support even broader resolution ranges. Modern MLLMs have already been evaluated extensively on **ultra-high-resolution 8K×8K benchmarks**, including **HR-Bench** [1] and **XLRS-Bench** [2]. Therefore, we do not view resolution as a limiting factor that would create an irreconcilable gap between human experts and MLLMs.
>
> Second, regarding human–machine visual alignment, recent work published in *Nature* provides further clarity:
> *“Aligning machine and human visual representations across abstraction levels,” Nature, 12 November 2025.*
> This study shows that divergences between human and machine perception primarily arise from **hierarchical representational differences**, rather than resolution constraints—and that these gaps are **in principle learnable and alignable**.
>
> While OmniEarth-Bench does **not** propose a new model to align human and machine visual systems, it **does** provide a suitable evaluation platform for such future research. Achieving higher performance on OmniEarth-Bench can serve as evidence that a model is better aligned with expert-level reasoning and perception in Earth-science contexts.
>
> [1]Divide, Conquer and Combine: A Training-Free Framework for High-Resolution Image Perception in Multimodal Large Language Models
>
> [2]XLRS-Bench: Could Your Multimodal LLMs Understand Extremely Large Ultra-High-Resolution Remote Sensing Imagery?

---

### Official Review · Reviewer_PEzL · 2025-11-12

**Soundness:** 3
**Presentation:** 4
**Contribution:** 3
**Rating:** 6
**Confidence:** 2

**Summary:**

The paper introduces OmniEarth-Bench, a multimodal benchmark that evaluates MLLMs on observational Earth-system data across all six spheres plus explicit cross-sphere tasks. It standardizes 29,855 expert-curated samples into a four-level hierarchy (Sphere → Scenario → Ability → Task) with 109 L4 tasks spanning perception, general reasoning, scientific-knowledge reasoning, and CoT reasoning. A modular pipeline ingests 33 native EO sources; domain experts and annotators curate and quality-check the data. Evaluations of nine modern MLLMs show none exceed ~35% accuracy, with especially poor performance on cross-sphere reasoning, highlighting large capability gaps and motivating domain-aware modeling.

**Strengths:**

- First benchmark to systematically cover all Earth spheres and cross-sphere interactions using real EO data, not just exam-style questions
- There is a clear four-level framework and ability taxonomy (Perception, General, Scientific-Knowledge, CoT) tailored to Earth-science reasoning
- The benchmark results expose a large domain gap in Earth system cognitive ability, where even the most advanced models do not achieve higher than 35% accuracy
- The hierarchy, data source (Table 2), example tasks (Fig. 4), and tabled comparisons communicate clearly what is being measured; key numbers (tasks/annotations/sources) are easy to find

**Weaknesses:**

- There are only accuracy-related metrics being reported alone about the models tested. Authors could add ability-targeted probes or per-ability sub-scores to better isolate failure modes, which would also give a more nuanced understanding of why and where exactly these models failed
- Some L1 spheres (e.g., cryosphere) appear a lot smaller (230 questions) compared to others in L1 sphere, which can make averages sensitive to composition
- More detail/metrics is needed on refusal handling. The paper notes safety/refusals can depress scores; provide refusal-aware metrics (e.g., Acc@answered, abstention rate) and report both to avoid penalizing safe behavior.

**Questions:**

- The authors mention involving “MLLM specialists” alongside human experts and annotators. What concrete tasks are MLLM specialists responsible for, and how do these differ from the tasks performed by annotators and domain experts?
- Did the authors measure inter-annotator agreement for key tasks or spheres? Knowing this would be more helpful to understand how trustworthy the labels are.

---

> ### Author Response · Authors · 2025-11-20
>
> ### Weakness1:Authors could add ability-targeted probes or per-ability sub-scores to better isolate failure modes.
>
> Thank you for the thoughtful suggestion. We agree that aggregate accuracy can obscure distinct failure modes. In the Appendix A.6, we report ability-targeted sub-scores by macro-averaging accuracy over L4 tasks mapped to three abilities—Perception (visual reading/grounding), General Reasoning (inference independent of domain facts), and Scientific-Knowledge Reasoning (discipline-aware, physically consistent reasoning). The table below illustrates these per-ability results for the evaluated models:
>
> | L3-capability| Qwen2.5-VL (7B) | Qwen2.5-VL (72B) | InternVL3 (8B) | InternVL3 (78B) | LLaVA-OV (7B) | GPT-4o | Gemini |   Claude3 |
> | ---------------------------------- | --------------: | ---------------: | -------------: | --------------: | ------------: | -----: | -----: | --------: |
> | **Perception**    |  13.42 |  15.37 |  **35.77** |           24.68 |         34.66 |  15.40 |  33.33 |     26.97 |
> | **General Reasoning**              |            7.32 |            10.86 |          28.67 |           27.48 |     **30.71** |   3.62 |  21.22 |     13.74 |
> | **Scientific-Knowledge Reasoning** |            8.20 |             5.72 |          30.89 |           27.49 |         30.18 |   8.74 |  23.66 | **31.62** |
>
> 1. **Perception vs. Reasoning split.**
>    *InternVL3-8B* leads on **Perception** (35.77%) and *LLaVA-OV-7B* is close (34.66%), yet **General Reasoning** peaks for *LLaVA-OV-7B* (30.71%) while *GPT-4o* is notably weak (3.62%).
>    **Implication:** Some models extract visual evidence reliably but underperform on multi-step inference.
>
> 2. **Domain knowledge grounding.**
>    **Scientific-Knowledge Reasoning** is highest for *Claude3* (31.62%), with *InternVL3-8B* (30.89%) and *LLaVA-OV-7B* (30.18%) competitive, whereas *Qwen2.5-VL-72B* collapses (5.72%).
>    **Implication:** Scaling alone does not guarantee discipline-aware reasoning; alignment/knowledge integration is the bottleneck.
>
> 3. **Scaling is non-monotonic.**
>    *Qwen2.5-VL 7B→72B* improves Perception (13.42→15.37) but **degrades** Scientific-Knowledge (8.20→5.72). *InternVL3 8B→78B* lowers Perception (35.77→24.68) with only marginal changes in reasoning slices.
>    **Implication:** Data/optimization choices dominate over parameter count for these abilities.
>
> To further pinpoint *why* models fail, we also provide **Appendix Table 15 (“Table index of case-study figures by sub-tasks (L-3 capability) with associated error categories for each MLLM”)**, which maps representative errors to L-3 capabilities for each model. Our error taxonomy comprises:
>
> * **Spatio-temporal Frame Confusion** (misusing temporal ordering or time context),
> * **Threshold / Severity Mis-estimation** (incorrect quantitative or decision thresholds),
> * **Image-feature Misinterpretation** (legends/scales/visual cues misread; Perception-linked),
> * **Domain-knowledge / Semantic Mis-match** (fact/physics inconsistency; Scientific-Knowledge-linked),
> * **Over-cautious / Refusal** (unwarranted abstention under the benchmark protocol), and
> * **Target Mis-location** (mis-grounding of the queried region/object; Perception-linked).
>
> In summary, we now include the **capabilitytable** (Perception, General Reasoning, Scientific-Knowledge) to isolate where each model underperforms. In **Appendix Table 15**, we index case-study figures by L-3 capability and attach error categories (Spatio-temporal Frame Confusion; Threshold/Severity Mis-estimation; Image-feature Misinterpretation; Domain-knowledge/Semantic Mis-match; Over-cautious/Refusal; Target Mis-location). Together, these additions make the sub-abilities and concrete failure modes explicit across models—**thank you for the valuable suggestion.**
>
> ### Weakness2: Some L1 spheres appear a lot smaller compared to others in L1 sphere, which can make averages sensitive to composition.
>
> Thank you for raising this concern. The sizes of L1 spheres indeed vary substantially (e.g., the Cryosphere subset is much smaller), which could make metrics sensitive to composition if sample-size–weighted averages (micro metrics) were used. In fact, we have also already considered this concens. To avoid this issue, all reported accuracies in our paper use a **macro-averaged metric**, not micro averaging.
>
> Let $s$ index L1 spheres and $t$ index L4 sub-tasks under sphere $s$. For each sub-task, we compute the accuracy $\text{Acc}_{s,t}$.
>
> 1. Per-sphere score (macro over L4 sub-tasks)
>
> $$
> \text{Acc}\_{s} = \frac{1}{|\mathcal{T}\_s|} \sum\_{t \in \mathcal{T}\_s} \text{Acc}\_{s,t}.
> $$
>
> Each L4 sub-task contributes equally, regardless of its sample size.
>
> 2. Overall score (macro over spheres)
>
> $$
> \text{Acc}\_{\text{overall}} = \frac{1}{|\mathcal{S}|} \sum\_{s\in\mathcal{S}} \text{Acc}\_{s}.
> $$
>
> Each L1 sphere contributes equally, even if their question counts differ significantly. We will add this explnations (**macro-averaged metric**) in our paper.

---

> ### Author Response · Authors · 2025-11-20
>
> ### Weakness3: More detail/metrics is needed on refusal handling. The paper notes safety/refusals can depress scores; provide refusal-aware metrics (e.g., Acc@answered, abstention rate)
>
> Thank you for the valuable suggestion regarding refusal handling. We agree that safety-driven refusals can artificially depress raw accuracy, and we have now included explicit refusal-aware metrics to disentangle *answer correctness* from *abstention behavior*.
>
> The additional metrics are reported:
>
> - **Abstention rate**: percentage of questions where the model refuses to answer
> - **Acc@answered**: accuracy conditioned on answering
> - **ΔAcc = Acc@answered − Acc**: change relative to standard accuracy(conditioned on all questions)
>
> These metrics clarify the interaction between safety behaviors and task performance.
>
> | Abstention rate               | Cross. | Atmo. | Litho. | Ocean. | Cryo. | Bio.  | Human. | Avg.  |
> | ----------------------------- | ------ | ----- | ------ | ------ | ----- | ----- | ------ | ----- |
> | Qwen2.5-VL-7B-Instruct        | 80.44  | 63.16 | 45.76  | 36.74  | 67.39 | 74.90 | 58.16  | 60.94 |
> | Qwen2.5-VL-72B-Instruct       | 87.21  | 65.30 | 48.50  | 35.55  | 71.74 | 61.60 | 48.18  | 59.73 |
> | lava-onevision-qwen2-7b-ov    | 1.36   | 1.76  | 0      | 12.49  | 0     | 18.20 | 18.19  | 7.43  |
> | InternVL3-8B-Instruct         | 0.72   | 37.09 | 0.16   | 15.68  | 1.30  | 34.69 | 24.21  | 16.26 |
> | InternVL3-78B-Instruct        | 48.12  | 52.69 | 30.67  | 7.66   | 0.87  | 39.51 | 28.04  | 29.65 |
> | internlm-xcomposer2d5-7b-chat | 3.99   | 0.27  | 9.90   | 2.91   | 0     | 4.48  | 14.33  | 5.13  |
>
> | Acc@answered/ΔAcc             | Cross.       | Atmo.       | Litho.      | Ocean.      | Cryo.       | Bio.        | Human.      | Avg.        |
> | ----------------------------- | ------------ | ----------- | ----------- | ----------- | ----------- | ----------- | ----------- | ----------- |
> | Qwen2.5-VL-7B-Instruct        | 50.36/40.51  | 25.11/15.86 | 34.39/15.74 | 22.08/8.10  | 54.74/36.89 | 43.59/32.65 | 14.89/8.66  | 35.02/22.63 |
> | Qwen2.5-VL-72B-Instruct       | 30.65/26.73  | 13.89/9.07  | 43.55/21.12 | 25.24/8.97  | 20.81/14.93 | 38.83/23.92 | 16.65/8.02  | 27.09/16.11 |
> | llava-onevision-qwen2-7b-ov   | 19.53/0.27   | 34.29/0.60  | 28.72/0.00  | 28.04/3.50  | 16.40/0.00  | 45.61/8.30  | 37.43/6.81  | 34.29/2.78  |
> | InternVL3-8B-Instruct         | 43.16/0.31   | 47.84/17.74 | 37.53/0.06  | 24.05/3.77  | 49.92/0.65  | 44.01/15.27 | 30.58/7.40  | 39.59/6.46  |
> | InternVL3-78B-Instruct        | 36.99/-11.13 | 71.83/19.14 | 33.74/3.07  | 21.90/14.24 | 75.21/74.34 | 52.88/13.37 | 40.94/12.90 | 47.64/17.99 |
> | internlm-xcomposer2d5-7b-chat | 20.60/0.82   | 17.50/0.05  | 32.05/3.17  | 21.69/0.63  | 40.04/0.00  | 32.11/1.44  | 28.90/4.14  | 27.56/1.46  |
>
> **(1) Large variation in refusal behaviors across models**
>
> As shown in the abstention-rate table: Qwen2.5-VL models refuse on ~60% of questions on average (up to 87% on Cross-sphere tasks). LLaVA-OneVision and XComposer 2.5 refuse on <7%. InternVL models exhibit moderate refusals (7–29%), depending on sphere. This demonstrates that raw accuracy alone cannot reveal a model’s actual capability, as it is strongly confounded by how often the model abstains.
>
> **(2) Acc@answered uncovers the model’s “true” knowledge when it chooses to answer**
>
> Across spheres:Qwen2.5-VL-7B achieves 35.0% Acc@answered, substantially higher than its raw accuracy due to its high abstention rate. Models with low refusal rates (e.g., LLaVA-OneVision) show minimal ΔAcc, implying stable answering behavior and little influence from abstentions. Thus, Acc@answered provides a clearer estimate of the model’s competence independent of its safety policy.
>
> **(3) Models with higher refusal rates also show stronger distinction between “known” vs. “unknown”**
>
> Notably, the highest refusal rates occur in the larger or more strongly pretrained models, such as: Qwen2.5-VL 7B / 72B and InternVL3-78B. These are also the models with the largest ΔAcc, suggesting that: they possess a clearer internal estimate of their knowledge boundary, they avoid answering when uncertain and when they *do* answer, their accuracy is substantially higher.
>
> Thank you again for your helpful suggestion; in the revision we will add refusal-aware reporting—**abstention rate**, **accuracy conditioned on answering (Acc@answered)**, and **ΔAcc = Acc@answered − Acc**—so overall performance and refusal behavior are jointly quantified and cleanly disentangled.

---

> ### Author Response · Authors · 2025-11-20
>
> ### Question1:  How do MLLM differ from annotators and domain experts?
>
> Thank you for asking us to clarify roles of MLLMs. Our pipeline (Fig. 4) has four stages—**A. Source Screening**, **B. Task Construction**, **C. Dataset Construction**, and **D. Quality Control**—all led by domain experts. Stages A–C are conducted exclusively by experts (design/specification) and trained annotators (execution); no MLLMs are used to create items or labels. MLLMs are only employed in **Stage D** as **auxiliary validators** under fixed prompts and without access to gold answers. Concretely:
> 1. **Difficulty Validation (MLLM-assisted, QC only).** Probe items with baseline MLLMs under the *Fair Comparison Protocol* to flag extremes—e.g., “too easy” (trivial cues) or “too hard” (systematically unsolved despite clear expert consensus). Flagged items are sent to human experts for adjudication or revision.
> 2. **Value Validation (MLLM-assisted, QC only).** Check that an item is *non-trivial and informative* for the targeted capability (e.g., truly requires cross-sphere integration rather than a single cue). The MLLM provides a brief justification; experts review and decide keep/revise/drop.
> 3. **Format Validation (MLLM- and rule-based, QC only).** Verify conformance to our item schema: option count and uniqueness, unit/legend presence, time-stamp consistency, image–text alignment, and prompt readability. Any violations are auto-flagged for correction by editors.
> 4. **Logical Validation (MLLM-assisted, QC only).** Stress-test internal consistency (question ↔ image(s) ↔ gold answer ↔ rationale) via paraphrase prompts and contradiction checks; detect label–rationale mismatches or logically inconsistent distractors. Final decisions remain with experts.
>
> The difference from experts/annotators and MLLMs:
> * **Domain experts (Stages A–C):** define sources and variables (A), design tasks and templates, author gold labels and rationales, set capability mappings and acceptance criteria (B), supervise dataset assembly and adjudicate conflicts (C).
> * **Annotators (Stage C):** perform primary labeling/boxing under expert guidelines; no authority to set tasks or acceptance rules.
> * **MLLMs (Stage D only):** do *not* generate items or labels; they act as *triage tools* to surface potential issues (difficulty, value, format, logic) for **expert** review. All pass/fail and edits are decided by humans.
>
> We will make this division explicit in the text and caption of Fig. 4, noting that **MLLMs are used solely for quality control signals**, while task design, labeling, and final arbitration are human-led. This clarification should resolve any ambiguity about “MLLM experts” versus annotators and domain experts.
>
> ###  Question2: Did the authors measure inter-annotator agreement for key tasks or spheres?
>
> We thank the reviewer for highlighting the importance of quantifying annotation reliability. We agree that inter-annotator agreement (IAA) is crucial for assessing how trustworthy the labels are, especially given that OmniEarth-Bench spans multiple Earth spheres and involves both domain experts and trained annotators.
>
> We appreciate the reviewer’s emphasis on annotation reliability. Our pipeline (Fig. 4) comprises **A. Source Screening**, **B. Task Construction**, **C. Dataset Construction**, and **D. Quality Control**, all led by domain experts. For each L1 sphere, **2–5 experts** formed a panel to define data sources and capability dimensions, ensuring within-domain consistency in both sources and task taxonomy. Stages A–C are executed exclusively by experts (design/specification) and trained annotators (execution). In addition, a **cross-validation team** reviews crowd annotations to enforce consistency, as described in the main text (lines 265–267).
>
> To make label trustworthiness explicit, the revision will report **inter-annotator agreement (IAA)** on key tasks/spheres. Concretely, for each L1 sphere, we randomly sample 50 MCQ items and have at least two independent annotators re-label them blind to the expert gold. We then compute: **Percentage agreement** and **Annotator–gold agreement** , with results reported per sphere and sample counts. This design quantifies both “annotator–annotator” and “annotator–gold” reliability and avoids composition bias.
>
> |L1 Sphere| A–A % Agreement | Ann–Gold % Agreement |
> |- |-: |-:|
> |Atmosphere|96%| 94%|
> |Hydrosphere |98%|96%|
> |Cryosphere|94%|94%|
> |Lithosphere|92%|90%|
> |Biosphere|98%|94%|
> |Anthroposphere| 98%| 96%|
> |Cross-sphere|94%|92%|
>
> In summary, experts lead the construction (A–C) and final arbitration; annotators provide primary labels with cross-validation safeguards; and we explicitly measure and report IAA for the each spheres, thereby making the trustworthiness of labels transparent and auditable.
>
> **Overall, thank you again for your constructive feedback. If you have any further questions, please don’t hesitate to let us know—we would be happy to continue the discussion.**

---

### Note · Authors · 2026-01-26

I have read and agree with the venue's withdrawal policy on behalf of myself and my co-authors.

---

### Meta-Review · Area_Chair_LCic · 2026-01-01

**Summary:**

The major weaknesses mentioned by the authors were missing RAG baselines, the use of 2d input modality for time series. wq4J raises issues with limited methodological novelty and a lack of descriptive analysis and scalability of insights. srAo raises issues WRT cross-sphere interactions in questions.

**Reviewer Concerns:**

The response from the authors is too verbose with relatively little useful information, making it incredibly hard to read and analyze. To put things in perspective, the response itself is larger than the 53-page paper with an appendix. In the future, I would recommend the authors to keep the rebuttal to the point, providing only the essential information; otherwise, important data/experiments get buried.

For example, the whole detailed RAG description was not as important during rebuttal, compared to the results of such a RAG system. *At the same time, the authors did not address major weaknesses mentioned by reviewer wq4J.* This shows a bit of insincerity from the authors towards the rebuttal process.

The authors did address issues with RAG, as well as responded to issues from srAo, well.
However, it is very challenging to understand in the wall of text if all the issues are well adressed.

**Reviewer Scores:**

* PEzL: Likely to keep the score at 6
* cP9P: Likely to keep the score at 4. While I agree with the authors that earth observation is not a niche domain, but very broad in terms of science applications. However, the reviewer's issue with 1D and time series data still stands.
* wq4J: None of the weaknesses have been addressed. Scores are likely to go from 4 to 2.
* srAo: Were fairly happy, scores likely to stay the same at 4.

On average, the score stays slightly below borderline, therefore I'm recommending a rejection.

---

### Decision · Program_Chairs · 2026-01-26

Reject